# On Oracle-Efficient PAC RL with Rich Observations

**Christoph Dann**
Carnegie Mellon University
Pittsburgh, Pennsylvania
cdann@cdann.net

**Nan Jiang**[*]
UIUC
Urbana, Illinois
nanjiang@illinois.edu

**Akshay Krishnamurthy**
Microsoft Research
New York, New York
akshay@cs.umass.edu

**Alekh Agarwal**
Microsoft Research
Redmond, Washington
alekha@microsoft.com

**John Langford**
Microsoft Research
New York, New York
jcl@microsoft.com

**Robert E. Schapire**
Microsoft Research
New York, New York
schapire@microsoft.com

## Abstract

We study the computational tractability of PAC reinforcement learning with rich observations. We present new provably sample-efficient algorithms for environments with deterministic hidden state dynamics and stochastic rich observations. These methods operate in an oracle model of computation—accessing policy and value function classes exclusively through standard optimization primitives—and therefore represent computationally efficient alternatives to prior algorithms that require enumeration. With stochastic hidden state dynamics, we prove that the only known sample-efficient algorithm, OLIVE [1], cannot be implemented in the oracle model. We also present several examples that illustrate fundamental challenges of tractable PAC reinforcement learning in such general settings.

## 1 Introduction

We study episodic reinforcement learning (RL) in environments with realistically rich observations such as images or text, which we refer to broadly as *contextual decision processes*. We aim for methods that use function approximation in a provably effective manner to find the best possible policy through strategic exploration.

While such problems are central to empirical RL research [2], most theoretical results on strategic exploration focus on tabular MDPs with small state spaces [3–10]. Comparatively little work exists on provably effective exploration with large observation spaces that require generalization through function approximation. The few algorithms that do exist either have poor sample complexity guarantees [e.g., 11–14] or require fully deterministic environments [15, 16] and are therefore inapplicable to most real-world applications and modern empirical RL benchmarks. This scarcity of positive results on efficient exploration with function approximation can likely be attributed to the challenging nature of this problem rather than a lack of interest by the research community.

On the statistical side, recent important progress was made by showing that contextual decision processes (CDPs) with rich stochastic observations and deterministic dynamics over $M$ hidden states can be learned with a sample complexity polynomial in $M$ [17]. This was followed by an algorithm called OLIVE [1] that enjoys a polynomial sample complexity guarantee for a broader range of CDPs, including ones with stochastic hidden state transitions. While encouraging, these efforts focused exclusively on statistical issues, ignoring computation altogether. Specifically, the proposed algorithms exhaustively enumerate candidate value functions to eliminate the ones that violate Bellman equations, an approach that is computationally intractable for any function class of

---

[*]The work was done while NJ was a postdoc researcher at MSR NYC.

practical interest. Thus, while showing that RL with rich observations can be statistically tractable, these results leave open the question of computational feasibility.

In this paper, we focus on this difficult computational challenge. We work in an oracle model of computation, meaning that we aim to design sample-efficient algorithms whose computation can be reduced to common optimization primitives over function spaces, such as linear programming and cost-sensitive classification. The oracle-based approach has produced practically effective algorithms for active learning [18], contextual bandits [19], structured prediction [20, 21], and multi-class classification [22], and here, we consider oracle-based algorithms for challenging RL settings.

We begin by studying the setting of Krishnamurthy et al. [17] with deterministic dynamics over $M$ hidden states and stochastic rich observations. In Section 4, we use cost-sensitive classification and linear programming oracles to develop VALOR, the first algorithm that is both *computationally* and *statistically* efficient for this setting. While deterministic hidden-state dynamics are somewhat restrictive, the model is considerably more general than fully deterministic MDPs assumed by prior work [15, 16], and it accurately captures modern empirical benchmarks such as visual grid-worlds in Minecraft [23]. As such, this method represents a considerable advance toward provably efficient RL in practically relevant scenarios.

Nevertheless, we ultimately seek efficient algorithms for more general settings, such as those with stochastic hidden-state transitions. Working toward this goal, we study the computational aspects of the OLIVE algorithm [1], which applies to a wide range of environments. Unfortunately, in Section 5.1, we show that OLIVE *cannot* be implemented efficiently in the oracle model of computation. As OLIVE is the only known statistically efficient approach for this general setting, our result establishes a significant barrier to computational efficiency. In the appendix, we also describe several other barriers, and two other oracle-based algorithms for the deterministic-dynamics setting that are considerably different from VALOR. The negative results identify where the hardness lies while the positive results provide a suite of new algorithmic tools. Together, these results advance our understanding of efficient reinforcement learning with rich observations.

## 2 Related Work

There is abundant work on strategic exploration in the tabular setting [3–10]. The computation in these algorithms often involves planning in optimistic models and can be solved efficiently via dynamic programming. To extend the theory to the more practical settings of large state spaces, typical approaches include (1) distance-based state identity test under smoothness assumptions [e.g., 11–14], or (2) working with factored MDPs [e.g., 24]. The former approach is similar to the use of state abstractions [25], and typically incurs exponential sample complexity in state dimension. The latter approach does have sample-efficient results, but the factored representation assumes relatively disentangled state variables which cannot model rich sensory inputs (such as images).

Azizzadenesheli et al. [26] have studied regret minimization in rich observation MDPs, a special case of contextual decision processes with a small number of hidden states and reactive policies. They do not utilize function approximation, and hence incur polynomial dependence on the number of unique observations in both sample and computational complexity. Therefore, this approach, along with related works [27, 28], does not scale to the rich observation settings that we focus on here.

Wen and Van Roy [15, 16] have studied exploration with function approximation in fully deterministic MDPs, which is considerably more restrictive than our setting of deterministic hidden state dynamics with stochastic observations and rewards. Moreover, their analysis measures representation complexity using *eluder dimension* [29, 30], which is only known to be small for some simple function classes. In comparison, our bounds scale with more standard complexity measures and can easily extend to VC-type quantities, which allows our theory to apply to practical and popular function approximators including neural networks [31].

## 3 Setting and Background

We consider reinforcement learning (RL) in a common special case of contextual decision processes [17, 1], sometimes referred to as rich observation MDPs [26]. We assume an $H$-step process where in each episode, a random *trajectory* $s_1, x_1, a_1, r_1, s_2, x_2, \ldots, s_H, x_H, a_H, r_H$ is generated.

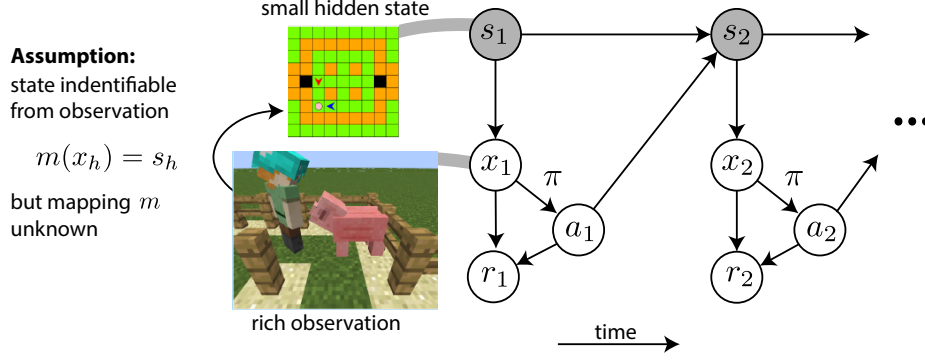

Figure 1: Graphical representation of the problem class considered by our algorithm, VALOR: The main assumptions that enable sample-efficient learning are (1) that the small hidden state $s_h$ is identifiable from the rich observation $x_h$ and (2) that the next state is a deterministic function of the previous state and action. State and observation examples are from https://github.com/Microsoft/malmo-challenge.

For each time step (or *level*) $h \in [H]$, $s_h \in \mathcal{S}$ where $\mathcal{S}$ is a finite hidden state space, $x_h \in \mathcal{X}$ where $\mathcal{X}$ is the rich observation (context) space, $a_h \in \mathcal{A}$ where $\mathcal{A}$ is a finite action space of size $K$, and $r_h \in \mathbb{R}$. Each hidden state $s \in \mathcal{S}$ is associated with an emission process $O_s \in \Delta(\mathcal{X})$, and we use $x \sim s$ as a shorthand for $x \sim O_s$. We assume that each rich observation contains enough information so that $s$ can in principle be identified just from $x \sim O_s$—hence $x$ is a Markov state and the process is in fact an MDP over $\mathcal{X}$—but the mapping $x \mapsto s$ is unavailable to the agent and $s$ is never observed. The hidden states $\mathcal{S}$ introduce structure into the problem, which is essential since we allow the observation space $\mathcal{X}$ to be infinitely large.[2] The issue of partial observability is not the focus of the paper.

Let $\Gamma : \mathcal{S} \times \mathcal{A} \to \Delta(\mathcal{S})$ define transition dynamics over the hidden states, and let $\Gamma_1 \in \Delta(\mathcal{S})$ denote an initial distribution over hidden states. $R : \mathcal{X} \times \mathcal{A} \to \Delta(\mathbb{R})$ is the reward function; this differs from partially observable MDPs where reward depends only on $s$, making the problem more challenging. With this notation, a trajectory is generated as follows: $s_1 \sim \Gamma_1$, $x_1 \sim s_1$, $r_1 \sim R(x_1, a_1)$, $s_2 \sim \Gamma(s_1, a_1)$, $x_2 \sim s_2, \ldots, s_H \sim \Gamma(s_{H-1}, a_{H-1})$, $x_H \sim s_H$, $r_H \sim R(x_H, a_H)$, with actions $a_{1:H}$ chosen by the agent. We emphasize that $s_{1:H}$ are unobservable to the agent.

To simplify notation, we assume that each observation and hidden state can only appear at a particular level. This implies that $\mathcal{S}$ is partitioned into $\{\mathcal{S}_h\}_{h=1}^H$ with size $M := \max_{h \in [H]} |\mathcal{S}_h|$. For regularity, assume $r_h \geq 0$ and $\sum_{h=1}^H r_h \leq 1$ almost surely.

In this setting, the learning goal is to find a policy $\pi : \mathcal{X} \to \mathcal{A}$ that maximizes the expected return $V^\pi := \mathbf{E}[\sum_{h=1}^H r_h \,|\, a_{1:H} \sim \pi]$. Let $\pi^\star$ denote the optimal policy, which maximizes $V^\pi$, with optimal value function $g^\star$ defined as $g^\star(x) := \mathbf{E}[\sum_{h'=h}^H r_{h'} | x_h = x, a_{h:H} \sim \pi^\star]$. As is standard, $g^\star$ satisfies the Bellman equation: $\forall x$ at level $h$,

$$g^\star(x) = \max_{a \in \mathcal{A}} \mathbf{E}[r_h + g^\star(x_{h+1}) | x_h = x, a_h = a],$$

with the understanding that $g^\star(x_{H+1}) \equiv 0$. A similar equation holds for the optimal Q-value function $Q^\star(x, a) := \mathbf{E}[\sum_{h'=h}^H r_{h'} | x_h = x, a_h = a, a_{h+1:H} \sim \pi^\star]$, and $\pi^\star = \operatorname{argmax}_{a \in \mathcal{A}} Q^\star(x, a)$.[3]

Below are two special cases of the setting described above that will be important for later discussions. **Tabular MDPs**: An MDP with a finite and small state space is a special case of this model, where $\mathcal{X} = \mathcal{S}$ and $O_s$ is the identity map for each $s$. This setting is relevant in our discussion of oracle-efficiency of the existing OLIVE algorithm in Section 5.1.
**Deterministic dynamics over hidden states**: Our algorithm, VALOR, works in this special case, which requires $\Gamma_1$ and $\Gamma(s, a)$ to be point masses. Originally proposed by Krishnamurthy et al. [17],

this setting can model some challenging benchmark environments in modern reinforcement learning, including visual grid-worlds common to the deep RL literature [e.g., 23]. In such tasks, the state records the position of each game element in a grid but the agent observes a rendered 3D view. Figure 1 shows a visual summary of this setting. We describe VALOR in detail in Section 4.

Throughout the paper, we use $\hat{\mathbf{E}}_D[\cdot]$ to denote empirical expectation over samples from a data set $D$.

## 3.1 Function Classes and Optimization Oracles

As $\mathcal{X}$ can be rich, the agent must use function approximation to generalize across observations. To that end, we assume a given value function class $\mathcal{G} \subset (\mathcal{X} \to [0, 1])$ and policy class $\Pi \subset (\mathcal{X} \to \mathcal{A})$. Our algorithm is agnostic to the specific function classes used, but for the guarantees to hold, they must be expressive enough to represent the optimal value function and policy, that is, $\pi^\star \in \Pi$ and $g^\star \in \mathcal{G}$. Prior works often use $\mathcal{F} \subset (\mathcal{X} \times \mathcal{A} \to [0, 1])$ to approximate $Q^\star$ instead, but for example Jiang et al. [1] point out that their OLIVE algorithm can equivalently work with $\mathcal{G}$ and $\Pi$. This $(\mathcal{G}, \Pi)$ representation is useful in resolving the computational difficulty in the deterministic setting, and has also been used in practice [32].

When working with large and abstract function classes as we do here, it is natural to consider an oracle model of computation and assume that these classes support various optimization primitives. We adopt this *oracle-based* approach here, and specifically use the following oracles:

**Cost-Sensitive Classification (CSC) on Policies.** A cost-sensitive classification (CSC) oracle receives as inputs a parameter $\epsilon_{\text{sub}}$ and a sequence $\{(x^{(i)}, c^{(i)})\}_{i \in [n]}$ of observations $x^{(i)} \in \mathcal{X}$ and cost vectors $c^{(i)} \in \mathbb{R}^K$, where $c^{(i)}(a)$ is the cost of predicting action $a \in \mathcal{A}$ for $x^{(i)}$. The oracle returns a policy whose average cost is within $\epsilon_{\text{sub}}$ of the minimum average cost, $\min_{\pi \in \Pi} \frac{1}{n} \sum_{i=1}^{n} c^{(i)}(\pi(x^{(i)}))$. While CSC is NP-hard in the worst case, CSC can be further reduced to binary classification [33, 34] for which many practical algorithms exist and actually form the core of empirical machine learning. As further motivation, the CSC oracle has been used in practically effective algorithms for contextual bandits [35, 19], imitation learning [20], and structured prediction [21].

**Linear Programs (LP) on Value Functions.** A linear program (LP) oracle considers an optimization problem where the objective $o : \mathcal{G} \to \mathbb{R}$ and the constraints $h_1, \ldots h_m$ are linear functionals of $\mathcal{G}$ generated by finitely many function evaluations. That is, $o$ and each $h_j$ have the form $\sum_{i=1}^{n} \alpha_i g(x_i)$ with coefficients $\{\alpha_i\}_{i \in [n]}$ and contexts $\{x_i\}_{i \in [n]}$. Formally, for a program of the form

$$\max_{g \in \mathcal{G}} o(g), \quad \text{subject to } h_j(g) \le c_j, \ \forall j \in [m],$$

with constants $\{c_j\}_{j \in [m]}$, an LP oracle with approximation parameters $\epsilon_{\text{sub}}, \epsilon_{\text{feas}}$ returns a function $\hat{g}$ that is at most $\epsilon_{\text{sub}}$-suboptimal and that violates each constraint by at most $\epsilon_{\text{feas}}$. For intuition, if the value functions $\mathcal{G}$ are linear with parameter vector $\theta \in \mathbb{R}^d$, i.e., $g(x) = \langle \theta, x \rangle$, then this reduces to a linear program in $\mathbb{R}^d$ for which a plethora of provably efficient solvers exist. Beyond the linear case, such problems can be practically solved using standard continuous optimization methods. LP oracles are also employed in prior work focusing on deterministic MDPs [15, 16].

**Least-Squares (LS) Regression on Value Functions.** We also consider a least-squares regression (LS) oracle that returns the value function which minimizes a square-loss objective. Since VALOR does not use this oracle, we defer details to the appendix.

We define the following notion of oracle-efficiency based on the optimization primitives above.

**Definition 1** (Oracle-Efficient). *An algorithm is* oracle-efficient *if it can be implemented with polynomially many basic operations and calls to CSC, LP, and LS oracles.*

Note that our algorithmic results continue to hold if we include additional oracles in the definition, while our hardness results easily extend, provided that the new oracles can be efficiently implemented in the tabular setting (i.e., they satisfy Proposition 6; see Section 5).

## 4 VALOR: An Oracle-Efficient Algorithm

In this section we propose and analyze a new algorithm, VALOR (Values stored Locally for RL) shown in Algorithm 1 (with 2 & 3 as subroutines). As we will show, this algorithm is oracle-efficient

and enjoys a polynomial sample-complexity guarantee in the deterministic hidden-state dynamics setting described earlier, which was originally introduced by Krishnamurthy et al. [17].

---

**Algorithm 1:** Main Algorithm VALOR

1 **Global**: $\mathcal{D}_1, \ldots \mathcal{D}_H$ initialized as $\emptyset$;
2 **Function** MetaAlg
3     dfslearn $(\varnothing)$ ;         // Alg.3
4     **for** $k = 1, \ldots, MH$ **do**
5        $\hat{\pi}^{(k)}, \hat{V}^{(k)} \leftarrow$ polvalfun() ;   // Alg.2
6        $T \leftarrow$ sample $n_{eval}$ trajectories with $\hat{\pi}^{(k)}$;
7        $\hat{V}^{\hat{\pi}^{(k)}} \leftarrow$ average return of $T$;
8        **if** $\hat{V}^{(k)} \leq \hat{V}^{\hat{\pi}^{(k)}} + \frac{\epsilon}{2}$ **then return** $\hat{\pi}^{(k)}$ ;
9        **for** $h = 1 \ldots H - 1$ **do**
10          **for** *all $a_{1:h}$ of $n_{expl}$ traj.* $\in T$ **do**
11            dfslearn $(a_{1:h})$ ;     // Alg.3
12     **return** failure;

---

**Algorithm 2:** Subroutine: Policy optimization with local values

1 **Function** polvalfun()
2     $\hat{V}^\star \leftarrow V$ of the only dataset in $\mathcal{D}_1$;
3     **for** $h = 1 : H$ **do**
4        // CSC-oracle
       $\hat{\pi}_h \leftarrow \underset{\pi \in \Pi_h}{\text{argmax}} \underset{(D, V, \{V_a\}) \in \mathcal{D}_h}{\sum} V_D(\pi; \{V_a\})$;
5     **return** $\hat{\pi}_{1:H}, \hat{V}^\star$;

---

**Notation:**
$V_D(\pi; \{V_a\}) := \hat{\mathbf{E}}_D[K\mathbf{1}\{\pi(x) = a\}(r + V_a)]$

---

**Algorithm 3:** Subroutine: DFS Learning of local values

1 $\epsilon_{\text{feas}} = \epsilon_{\text{sub}} = \epsilon_{\text{stat}} = \tilde{O}(\epsilon^2/MH^3)$ ;    // see exact values in Table 1 in the appendix
2 $\phi_h = (H + 1 - h)(6\epsilon_{\text{stat}} + 2\epsilon_{\text{sub}} + \epsilon_{\text{feas}})$ ;     // accuracy of learned values at level $h$
3 **Function** dfslearn(*path $p$ with length $h - 1$*)
4     **for** $a \in \mathcal{A}$ **do**
5        $D' \leftarrow$ Sample $n_{\text{test}}$ trajectories with actions $p \circ a$ ;
       // compute optimistic / pessimistic values using LP-oracle
6        $V_{opt} \leftarrow \max_{g \in \mathcal{G}_{h+1}} \hat{\mathbf{E}}_{D'}[g(x_{h+1})]$    (and $V_{pes} \leftarrow \min_{g \in \mathcal{G}_{h+1}} \hat{\mathbf{E}}_{D'}[g(x_{h+1})]$)
          s.t. $\forall (D, V, \_) \in \mathcal{D}_{h+1}: \ |V - \hat{\mathbf{E}}_D[g(x_{h+1})]| \leq \phi_{h+1}$ ;
7        **if** $|V_{opt} - V_{pes}| \leq 2\phi_{h+1} + 4\epsilon_{stat} + 2\epsilon_{feas}$ **then**
8          $V_a \leftarrow (V_{opt} + V_{pes})/2$ ;         // consensus among remaining functions
9        **else**
10          $V_a \leftarrow$ dfslearn$(p \circ a)$ ;            // no consensus, descend
11     $\tilde{D} \leftarrow$ Sample $n_{\text{train}}$ traj. with $p$ and $a_h \sim \text{Unif}(K)$;
12     $\tilde{V} \leftarrow \max_{\pi \in \Pi_h} V_{\tilde{D}}(\pi; \{V_a\})$;                      // CSC-oracle
13     Add $(\tilde{D}, \tilde{V}, \{V_a\}_{a \in \mathcal{A}})$ to $\mathcal{D}_h$;
14     **return** $\tilde{V}$;

---

Since hidden states can be deterministically reached by sequences of actions (or *paths*), from an algorithmic perspective, the process can be thought of as an exponentially large tree where each node is associated with a hidden state (such association is unknown to the agent). Similar to LSVEE [17], VALOR first explores this tree (Line 3) with a form of depth first search (Algorithm 3). To avoid visiting all of the exponentially many paths, VALOR performs a state identity test (Algorithm 3, Lines 5–8): the data collected so far is used to (virtually) eliminate functions in $\mathcal{G}$ (Algorithm 3, Line 6), and we do not descend to a child if the remaining functions agree on the value of the child node (Algorithm 3, Line 7).

The state identity test prevents exploring the same hidden state twice but might also incorrectly prune unvisited states if all functions happen to agree on the value. Unfortunately, with no data from such pruned states, we are unable to learn the optimal policy on them. To address this issue, after dfslearn returns, we first use the stored data and values (Line 5) to compute a policy (see Algorithm 2) that is near optimal on all explored states. Then, VALOR deploys the computed policy (Line 6) and only terminates if the estimated optimal value is achieved (Line 8). If not, the policy has good probability of visiting those accidentally pruned states (see Appendix B.5), so we invoke dfslearn on the generated paths to complement the data sets (Line 11).

In the rest of this section we describe VALOR in more detail, and then state its statistical and computational guarantees. VALOR follows a dynamic programming style and learns in a bottom-up fashion. As a result, even given stationary function classes $(\mathcal{G}, \Pi)$ as inputs, the algorithm can return a non-stationary policy $\hat{\pi}_{1:H} := (\hat{\pi}_1, \ldots, \hat{\pi}_H) \in \Pi^H$ that may use different policies at different time steps.[4] To avoid ambiguity, we define $\Pi_h := \Pi$ and $\mathcal{G}_h := \mathcal{G}$ for $h \in [H]$, to emphasize the time point $h$ under consideration. For convenience, we also define $\mathcal{G}_{H+1}$ to be the singleton $\{x \mapsto 0\}$. This notation also allows our algorithms to handle more general non-stationary function classes.

**Details of depth-first search exploration.** VALOR maintains many data sets collected at paths visited by `dfslearn`. Each data set $D$ is collected from some path $p$, which leads to some hidden state $s$. (Due to determinism, we will refer to $p$ and $s$ interchangeably throughout this section.) $D$ consists of tuples $(x, a, r)$ where $x \sim p$ (i.e., $x \sim O_s$), $a \sim \text{Unif}(K)$, and $r$ is the instantaneous reward. Associated with $D$, we also store a scalar $V$ which approximates $V^\star(s)$, and $\{V_a\}_{a \in \mathcal{A}}$ which approximate $\{V^\star(s \circ a)\}_{a \in \mathcal{A}}$, where $s \circ a$ denotes the state reached when taking $a$ in $s$. The estimates $\{V_a\}_{a \in \mathcal{A}}$ of the future optimal values associated with the current path $p \in \mathcal{A}^{h-1}$ are either determined through a recursive call (Line 10), or through a *state-identity test* (Lines 5–8 in `dfslearn`). To check if we already know $V^\star(p \circ a)$, we solve constrained optimization problems to compute optimistic and pessimistic estimates, using a small amount of data from $p \circ a$. The constraints eliminate all $g \in \mathcal{G}_{h+1}$ that make incorrect predictions for $V^\star(s')$ for any previously visited $s'$ at level $h + 1$. As such, if we have learned the value of $s \circ a$ on a different path, the optimistic and pessimistic values must agree ("consensus"), so we need not descend. Once we have the future values $V_a$, the value estimate $\tilde{V}$ (which approximates $V^\star(s)$) is computed (in Line 12) by maximizing the sum of immediate reward and future values, re-weighted using importance sampling to reflect the policy under consideration $\pi$:

$$V_D(\pi; \{V_a\}) := \hat{\mathbf{E}}_D[K\mathbf{1}\{\pi(x) = a\}(r + V_a)]. \tag{1}$$

**Details of policy optimization and exploration-on-demand.** `polvalfun` performs a sequence of policy optimization steps using all the data sets collected so far to find a non-stationary policy that is near-optimal at all explored states simultaneously. Note that this policy differs from that computed in (Alg. 3, Line 12) as it is common for all datasets at a level $h$. And finally using this non-stationary policy, `MetaAlg` estimates its suboptimality and either terminates successfully, or issues several other calls to `dfslearn` to gather more data sets. This so-called exploration-on-demand scheme is due to Krishnamurthy et al. [17], who describe the subroutine in more detail.

## 4.1 What is new compared to LSVEE?

The overall structure of VALOR is similar to LSVEE [17]. The main differences are in the pruning mechanism, where we use a novel state-identity test, and the policy optimization step in Algorithm 2.

LSVEE uses a $Q$-value function class $\mathcal{F} \subset (\mathcal{X} \times \mathcal{A} \to [0, 1])$ and a state identity test based on Bellman errors on data sets $D$ consisting of $(x, a, r, x')$ tuples:

$$\hat{\mathbf{E}}_D\left[\left(f(x, a) - r - \hat{\mathbf{E}}_{x' \sim a} \max_{a' \in \mathcal{A}} f(x', a')\right)^2\right].$$

This enables a conceptually simpler statistical analysis, but the coupling between value function and the policy yield challenging optimization problems that do not obviously admit efficient solutions.

In contrast, VALOR uses dynamic programming to propagate optimal value estimates from future to earlier time points. From an optimization perspective, we fix the future value and only optimize the current policy, which can be implemented by standard oracles, as we will see. However, from a statistical perspective, the inaccuracy of the future value estimates leads to bias that accumulates over levels. By a careful design of the algorithm and through an intricate and novel analysis, we show that this bias only accumulates linearly (as opposed to exponentially; see e.g., Appendix E.1), which leads to a polynomial sample complexity guarantee.

## 4.2 Computational and Sample Complexity of VALOR

VALOR requires two types of nontrivial computations over the function classes. We show that they can be reduced to CSC on $\Pi$ and LP on $\mathcal{G}$ (recall Section 3.1), respectively, and hence VALOR is oracle-efficient.

First, Lines 4 in `polvalfun` and 12 in `dfslearn` involve optimizing $V_D(\pi; \{V_a\})$ (Eq. (1)) over $\Pi$, which can be reduced to CSC as follows: We first form tuples $(x^{(i)}, a^{(i)}, y^{(i)})$ from $D$ and $\{V_a\}$ on which $V_D(\pi; \{V_a\})$ depends, where we bind $x_h$ to $x^{(i)}$, $a_h$ to $a^{(i)}$, and $r_h + V_{a_h}$ to $y^{(i)}$. From the tuples, we construct a CSC data set $(x^{(i)}, -[K\mathbf{1}\{a = a^{(i)}\}y^{(i)}]_{a\in\mathcal{A}})$. On this data set, the cost-sensitive error of any policy (interpreted as a classifier) is exactly $-V_D(\pi; \{V_a\})$, so minimizing error (which the oracle does) maximizes the original objective.

Second, the state identity test requires solving the following problem over the function class $\mathcal{G}$:

$$V_{opt} = \max_{g\in\mathcal{G}} \hat{\mathbf{E}}_{D'}[g(x_h)] \quad \text{(and min for } V_{pes}) \tag{2}$$

$$\text{s.t. } V - \phi_h \le \hat{\mathbf{E}}_D[g(x_h)] \le V + \phi_h, \forall(D, V) \in \mathcal{D}_h.$$

The objective and the constraints are linear functionals of $\mathcal{G}$, all empirical expectations involve polynomially many samples, and the number of constraints is $|\mathcal{D}_h|$ which remains polynomial throughout the execution of the algorithm, as we will show in the sample complexity analysis. Therefore, the LP oracle can directly handle this optimization problem.

We now formally state the main computational and statistical guarantees for VALOR.

**Theorem 2** (Oracle efficiency of VALOR). *Consider a contextual decision process with deterministic dynamics over $M$ hidden states as described in Section 3. Assume $\pi^\star \in \Pi$ and $g^\star \in \mathcal{G}$. Then for any $\epsilon, \delta \in (0, 1)$, with probability at least $1 - \delta$, VALOR makes $O\left(\frac{MH^2}{\epsilon}\log\frac{MH}{\delta}\right)$ CSC oracle calls and at most $O\left(\frac{MKH^2}{\epsilon}\log\frac{MH}{\delta}\right)$ LP oracle calls with required accuracy $\epsilon_{feas} = \epsilon_{sub} = \tilde{O}(\epsilon^2/MH^3)$.*

**Theorem 3** (PAC bound of VALOR). *Under the same setting and assumptions as in Theorem 2, VALOR returns a policy $\hat{\pi}$ such that $V^\star - V^{\hat{\pi}} \le \epsilon$ with probability at least $1 - \delta$, after collecting at most $\tilde{O}\left(\frac{M^3H^8K}{\epsilon^5}\log(|\mathcal{G}||\Pi|/\delta)\log^3(1/\delta)\right)$ trajectories.[5]*

Note that this bound assumes finite value function and policy classes for simplicity, but can be extended to infinite function classes with bounded statistical complexity using standard tools, as in Section 5.3 of Jiang et al. [1]. The resulting bound scales linearly with the Natarajan and Pseudo-dimension of the function classes, which are generalizations of VC-dimension. We further expect that one can generalize the theorems above to an approximate version of realizability as in Section 5.4 of Jiang et al. [1].

Compared to the guarantee for LSVEE [17], Theorem 3 is worse in the dependence on $M$, $H$, and $\epsilon$. Yet, in Appendix B.7 we show that a version of VALOR with alternative oracle assumptions enjoys a better PAC bound than LSVEE. Nevertheless, we emphasize that our main goal is to understand the interplay between statistical and computational efficiency to discover new algorithmic ideas that may lead to practical methods, rather than improve sample complexity bounds.

## 5 Toward Oracle-Efficient PAC-RL with Stochastic Hidden State Dynamics

VALOR demonstrates that provably sample- and oracle-efficient RL with rich stochastic observations is possible and, as such, makes progress toward reliable and practical RL in many applications. In this section, we discuss the natural next step of allowing stochastic hidden-state transitions.

### 5.1 OLIVE is not Oracle-Efficient

For this more general setting with stochastic hidden state dynamics, OLIVE [1] is the only known algorithm with polynomial sample complexity, but its computational properties remain underexplored.

We show here that OLIVE is in fact not oracle-efficient. A brief description of the algorithm is provided below, and in the theorem statement, we refer to a parameter $\phi$, which the algorithm uses as a tolerance on deviations of empirical expectations.

**Theorem 4.** *Assuming $P \neq NP$, even with algorithm parameter $\phi = 0$ and perfect evaluation of expectations,* OLIVE *is not oracle-efficient, that is, it cannot be implemented with polynomially many basic arithmetic operations and calls to CSC, LP, and LS oracles.*

The assumptions of perfect evaluation of expectations and $\phi = 0$ are merely to unclutter the constructions in the proofs. We show this result by proving that even in tabular MDPs, OLIVE solves an NP-hard problem to determine its next exploration policy, while all oracles we consider have polynomial runtime in the tabular setting. While we only show this for CSC, LP, and LS oracles explicitly, we expect other practically relevant oracles to also be efficient in the tabular setting, and therefore they could not help to implement OLIVE efficiently.

This theorem shows that there are no known oracle-efficient PAC-RL methods for this general setting and that simply applying clever optimization tricks to implement OLIVE is not enough to achieve a practical algorithm. Yet, this result does not preclude tractable PAC RL altogether, and we discuss plausible directions in the subsequent section. Below we highlight the main arguments of the proof.

**Proof Sketch of Theorem 4.** OLIVE is round-based and follows the *optimism in the face of uncertainty* principle. At round $k$ it selects a value function and a policy to execute $(\hat{g}_k, \hat{\pi}_k)$ that promise the highest return while satisfying all average Bellman error constraints:

$$\hat{g}_k, \hat{\pi}_k = \underset{g \in \mathcal{G}, \pi \in \Pi}{\operatorname{argmax}} \hat{\mathbf{E}}_{D_0}[g(x)] \tag{3}$$

$$\text{s.t. } |\hat{\mathbf{E}}_{D_i}[K\mathbf{1}\{a = \pi(x)\}(g(x) - r - g(x'))]| \leq \phi, \ \forall D_i \in \mathcal{D}.$$

Here $D_0$ is a data set of initial contexts $x$, $\mathcal{D}$ consists of data sets of $(x, a, r, x')$ tuples collected in the previous rounds, and $\phi$ is a statistical tolerance parameter. If this optimistic policy $\hat{\pi}_k$ is close to optimal, OLIVE returns it and terminates. Otherwise we add a constraint to (3) by (i) choosing a time point $h$, (ii) collecting trajectories with $\hat{\pi}_k$ but choosing the $h$-th action uniformly, and (iii) storing the tuples $(x_h, a_h, r_h, x_{h+1})$ in the new data set $D_k$ which is added to the constraints for the next round.

The following theorem shows that OLIVE's optimization is NP-hard even in tabular MDPs.

**Theorem 5.** *Let $\mathcal{P}_{\text{OLIVE}}$ denote the family of problems of the form (3), parameterized by $(\mathcal{X}, \mathcal{A}, \text{Env}, t)$, which describes the optimization problem induced by running* OLIVE *in the MDP Env (with states $\mathcal{X}$, actions $\mathcal{A}$, and perfect evaluation of expectations) for $t$ rounds.* OLIVE *is given tabular function classes $\mathcal{G} = (\mathcal{X} \rightarrow [0, 1])$ and $\Pi = (\mathcal{X} \rightarrow \mathcal{A})$ and uses $\phi = 0$. Then $\mathcal{P}_{\text{OLIVE}}$ is NP-hard.*

At the same time, oracles are implementable in polynomial time:

**Proposition 6.** *For tabular value functions $\mathcal{G} = (\mathcal{X} \rightarrow [0, 1])$ and policies $\Pi = (\mathcal{X} \rightarrow \mathcal{A})$, the CSC, LP, and LS oracles can be implemented in time polynomial in $|\mathcal{X}|$, $K = |\mathcal{A}|$ and the input size.*

Both proofs are in Appendix D. Proposition 6 implies that if OLIVE could be implemented with polynomially many CSC/LP/LS oracle calls, its total runtime would be polynomial for tabular MDPs. Assuming $P \neq NP$, this contradicts Theorem 5 which states that determining the exploration policy of OLIVE in tabular MDPs is NP-hard. Combining both statements therefore proves Theorem 4.

We now give brief intuition for Proposition 6. To implement the CSC oracle, for each of the polynomially many observations $x \in \mathcal{X}$, we simply add the cost vectors for that observation together and pick the action that minimizes the total cost, that is, compute the action $\hat{\pi}(x)$ as $\min_{a \in \mathcal{A}} \sum_{i \in [n]: x^{(i)} = x} c^{(i)}(a)$. Similarly, the square-loss objective of the LS-oracle decomposes and we can compute the tabular solution one entry at a time. In both cases, the oracle runtime is $O(nK|\mathcal{X}|)$. Finally, using one-hot encoding, $\mathcal{G}$ can be written as a linear function in $\mathbb{R}^{|\mathcal{X}|}$ for which the LP oracle problem reduces to an LP in $\mathbb{R}^{|\mathcal{X}|}$. The ellipsoid method [37] solves these approximately in polynomial time.

## 5.2 Computational Barriers with Decoupled Learning Rules.

One factor contributing to the computational intractability of OLIVE is that (3) involves optimizing over policies and values *jointly*. It is therefore promising to look for algorithms that separate

optimizations over policies and values, as in VALOR. In Appendix E, we provide a series of examples that illustrate some limitations of such algorithms. First, we show that methods that compute optimal values iteratively in the style of fitted value iteration [38] need additional assumptions on $\mathcal{G}$ and $\Pi$ besides realizability (Theorem 45). (Storing value estimates of states explicitly allows VALOR to only require realizability.) Second, we show that with stochastic state dynamics, average value constraints, as in Line 6 of Algorithm 3, can cause the algorithm to miss a high-value state (Proposition 46). Finally, we show that square-loss constraints suffer from similar problems (Proposition 47).

### 5.3 Alternative Algorithms.

An important element of VALOR is that it explicitly stores value estimates of the hidden states, which we call "local values." Local values lead to statistical and computational efficiency under weak realizability conditions, but this approach is unlikely to generalize to the stochastic setting where the agent may not be able to consistently visit a particular hidden state. In Appendices B.7-C.2, we therefore derive alternative algorithms which do not store local values to approximate the future value $g^\star(x_{h+1})$. Inspired by classical RL algorithms, these algorithms approximate $g^\star(x_{h+1})$ by either bootstrap targets $\hat{g}_{h+1}(x_{h+1})$ (as in TD methods) or Monte-Carlo estimates of the return using a near-optimal roll-out policy $\hat{\pi}_{h+1:H}$ (as in PSDP [39]). Using such targets can introduce additional errors, and stronger realizability-type assumptions on $\Pi, \mathcal{G}$ are necessary for polynomial sample-complexity (see Appendix C and E). Nevertheless, these algorithms are also oracle-efficient and while we only establish statistical efficiency with deterministic hidden state dynamics, we believe that they considerably expand the space of plausible algorithms for the general setting.

## 6 Conclusion

This paper describes new RL algorithms for environments with rich stochastic observations and deterministic hidden state dynamics. Unlike other existing approaches, these algorithms are computationally efficient in an oracle model, and we emphasize that the oracle-based approach has led to practical algorithms for many other settings. We believe this work represents an important step toward computationally and statistically efficient RL with rich observations.

While challenging benchmark environments in modern RL (e.g. visual grid-worlds [23]) often have the assumed deterministic hidden state dynamics, the natural goal is to develop efficient algorithms that handle stochastic hidden-state dynamics. We show that the only known approach for this setting is not implementable with standard oracles, and we also provide several constructions demonstrating other concrete challenges of RL with stochastic state dynamics. This provides insights into the key open question of whether we can design an efficient algorithm for the general setting. We hope to resolve this question in future work.

## Footnotes

[2]Indeed, the lower bound in Proposition 6 in Jiang et al. [1] show that ignoring underlying structure precludes provably-efficient RL, even with function approximation.

[3]Note that the optimal policy and value functions depend on $x$ and not just $s$ even if $s$ was known, since reward is a function of $x$.

[4]This is not rare in RL; see e.g., Chapter 3.4 of Ross [36].

[5] $\tilde{O}(\cdot)$ suppresses logarithmic dependencies on $M$, $K$, $H$, $1/\epsilon$ and doubly-logarithmic dependencies on $1/\delta$, $|\mathcal{G}|$, and $|\Pi|$.

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
