[Supplementary Material · fast_RL_supplementary.pdf]

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

[6]Note that the inputs to the oracle have polynomial length: $D$ consists of polynomially many $(x, a, r, x')$ tuples, each of which should be assumed to have polynomial description length, and $\{V_a\}$ similarly.

[7]This family algorithms are prevalent in empirical RL research today: the popular Q-learning algorithm with function approximation can be viewed as a stochastic variant of its batch version known as Fitted Q-iteration [46], which fits into the FVI framework.

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

# Appendices

## Contents

# A   Additional Notation and Definitions

In the next few sections we analyze the new algorithms for the deterministic setting. We will adopt the following conventions:

- In the deterministic setting (which we focus on here), a path $p$ always deterministically leads to some state $s$, so we use them interchangeably, e.g., $V^\star(p) \equiv V^\star(s)$, $x \sim p \Leftrightarrow x \sim s$.

- It will be convenient to define $V^\pi(s) := \mathbf{E}[\sum_{h'=h}^{H} r_{h'} \mid s_h = s, a_{h:H} \sim \pi]$ for $s$ at level $h$, which is the analogy of $V^\star(s)$ for $\pi$. Recall that $V^\pi \equiv V^\pi(\varnothing)$ and $V^\star \equiv V^\star(\varnothing)$. Also define $Q^\star(s, \pi) := \mathbf{E}_{x \sim s}[Q^\star(x, \pi(x))]$.

- We use $\hat{\mathbf{E}}_D[\cdot]$ to denote empirical expectation over samples drawn from data set $D$, and we use $\mathbf{E}_p[\cdot]$ to denote population averages where data is drawn from path $p$. Often for this latter expectation, we will draw $(x, a, r, x')$ where $x \sim p, a \sim \mathrm{Unif}(\mathcal{A})$ and $r, x'$ are sampled according to the appropriate conditional distributions. In the notation $\mathbf{E}_p$ we default to the uniform action distribution unless otherwise specified.

## A.1   Additional Oracles

**Least-Squares (LS) Oracle**   The least-squares oracle takes as inputs a parameter $\epsilon_{\mathrm{sub}}$ and a sequence $\{(x^{(i)}, v^{(i)})\}_{i \in [n]}$ of observations $x^{(i)} \in \mathcal{X}$ and values $v^{(i)} \in \mathbb{R}$. It outputs a value function $\hat{g} \in \mathcal{G}$ whose squared error is $\epsilon_{\mathrm{sub}}$ close to the least-squares fit

$$\min_{g \in \mathcal{G}} \frac{1}{n} \sum_{i=1}^{n} (v^{(i)} - g(x^{(i)}))^2. \tag{4}$$

**Multi Data Set Classification Oracle**   The multi data set classification oracle receives as inputs a parameter $\epsilon_{\mathrm{feas}}$, $m$ scalars that are upper bounds on the allowed cost $\{U_j\}_{j \in [m]} \in \mathbb{R}^m$, and $m$ cost-sensitive classification data sets $D_1, \ldots D_m$, each of which consists of a sequence of observations $\{x_j^{(i)}\}_{i \in [n]} \in \mathcal{X}^n$ and a sequence of cost vectors $\{c_j^{(i)}\}_{i \in [n]} \in \mathbb{R}^{K \times n}$, where $c_j^{(i)}(a)$ is the cost of predicting action $a \in \mathcal{A}$ for $x_j^{(i)}$. The oracle returns a policy that achieves on each data set $D_j$ at most an average cost of $U_j + \epsilon_{\mathrm{feas}}$, if a policy exists in $\Pi$ that achieves costs at most $U_j$ on each dataset. Formally, the oracle returns a policy in

$$\left\{ \pi \in \Pi \; \middle| \; \forall j \in [m] : \; \frac{1}{n} \sum_{i=1}^{n} c_j^{(i)}(\pi(x_j^{(i)})) \leq U_j + \epsilon_{\mathrm{feas}} \right\}. \tag{5}$$

This oracle generalizes the CSC oracle by requiring the same policy to achieve low cost on multiple CSC data sets simultaneously. Nonetheless, it can be implemented with a CSC oracle as follows: We associate a Lagrange parameter with each constraint, and optimize the Lagrange parameters using multiplicative weights. In each iteration, we use the multiplicative weights to combine the $m$ constraints into a single one, and then solve the resulting cost-sensitive problem with the CSC oracle. The slack in the constraint as witnessed by the resulting policy is used as the loss to update the multiplicative weights parameters. See [40] for more details.

## A.2   Assumptions on the Function Classes

While VALOR only requires realizability of the policy and the value function classes, our other algorithms require stronger assumptions which we introduce below.

**Assumption 7** (Policy realizability). *$\pi^\star \in \Pi$.*

**Assumption 8** (Value realizability). *$g^\star \in \mathcal{G}$.*

**Assumption 9** (Policy-value completeness). *At each level $h$, $\forall g' \in \mathcal{G}_{h+1}$, there exists $\pi_{g'}^\star \in \Pi_h$ such that $\forall x \in \mathcal{X}$,*

$$\pi_{g'}^\star(x) = \underset{a \in \mathcal{A}}{\mathrm{argmax}} \, \mathbf{E}[r + g'(x_{h+1}) | x_h = x, a_h = a].$$

*In addition, $\forall g' \in \mathcal{G}_{h+1}, \exists g_{\star,g'} \in \mathcal{G}_h$ s.t. $\forall x \in \mathcal{X}$,*

$$g_{\star,g'}(x) = \mathbf{E}[r + g'(x') | x_h = x, a_h = \pi_{g'}^\star(x)].$$

$$\epsilon_{\text{stat}} = \epsilon_{\text{sub}} = \epsilon_{\text{feas}} = \frac{\epsilon}{2^6 7 H^2 T_{\max}}$$

$$\phi_h = (H - h + 1)(6\epsilon_{\text{stat}} + 2\epsilon_{\text{sub}} + \epsilon_{\text{feas}})$$

$$T_{\max} = MHn_{\text{exp}} + M$$

$$n_{\text{test}} = \frac{\log\left(12KHT_{\max}|\mathcal{G}|/\delta\right)}{2\epsilon_{\text{stat}}^2},$$

$$n_{\text{train}} = \frac{16K \log(12HT_{\max}|\mathcal{G}||\Pi|/\delta)}{\epsilon_{\text{stat}}^2},$$

$$n_{\text{exp}} = \frac{8\log(4MH/\delta)}{\epsilon},$$

$$n_{\text{eval}} = \frac{32\log(8MH/\delta)}{\epsilon^2}$$

Table 1: Exact values of parameters of VALOR run with inputs $\epsilon, \delta \in (0,1)$ and $M, K \in \mathbb{N}$.

**Assumption 10** (Policy completeness). *For every $h$, and every non-stationary policy $\pi_{h+1:H}$, there exists a policy $\pi \in \Pi_h$ such that, for all $x \in \mathcal{X}_h$, we have*

$$\pi(x) = \text{argmax}_a \mathbf{E}[\sum_{h'=h}^H r_{h'}|x, a, a_{h+1:H} \sim \pi_{h+1:H}].$$

**Fact 11** (Relationship between the assumptions).
*Assum.9 $\Rightarrow$ Assum.10 $\Rightarrow$ Assum.7.    Assum.9 $\Rightarrow$ Assum.8.*

In words, these assumptions ask that for any possible approximation of the future value that we might use, the induced square loss or cost-sensitive problems are realizable using $\mathcal{G}, \Pi$, which is a much stronger notion of realizability than Assumptions 7 and 8. Such assumptions are closely related to the conditions needed to analyze Fitted Value/Policy Iteration methods [see e.g., 41, 42], and are further justified by Theorem 45 in Appendix E.

# B  Analysis of VALOR

**Definition 12.** *A state $s \in \mathcal{S}_h$ is called* learned *if there is a data set in $\mathcal{D}_h$ that is sampled from a path leading to that state. The set of all learned states at level $h$ is $\mathcal{S}_h^{learned}$ and $\mathcal{S}^{learned} := \bigcup_{h \in [H]} \mathcal{S}_h^{learned}$.*

## B.1  Concentration Results

We now define an event $\mathcal{E}$ that holds with high probability and will be the main concentration argument in the proof. This event uses a parameter $\epsilon_{\text{stat}}$ whose value we will set later.

**Definition 13** (Deviation Bounds). *Let $\mathcal{E}$ denote the event that for all $h \in [H]$ the total number of calls to* dfslearn($p$) *at level $h$ is at most $T_{\max} = MHn_{exp} + M$ during the execution of* MetaAlg *and that for all these calls to* dfslearn($p$) *the following deviation bounds hold for all $g \in \mathcal{G}_h$ and $\pi \in \Pi_h$ (where $D_a'$ is a data set of $n_{test}$ observations sampled from $p \circ a$ in Line 5, and $\tilde{D}$ is the data set of $n_{train}$ samples from Line 11 with stored values $\{V_a\}_{a \in \mathcal{A}}$):*

$$\left|\hat{\mathbf{E}}_{D_a'}[g(x_{h+1})] - \mathbf{E}_{p \circ a}[g(x_{h+1})]\right| \le \epsilon_{stat}, \quad \forall a \in \mathcal{A} \tag{6}$$

$$\left|\hat{\mathbf{E}}_{\tilde{D}}[g(x_h)] - \mathbf{E}_p[g(x_h)]\right| \le \epsilon_{stat} \tag{7}$$

$$\left|\hat{\mathbf{E}}_{\tilde{D}}[K\mathbf{1}\{\pi(x_h) = a_h\}(r_h + V_a)] - \mathbf{E}_p[K\mathbf{1}\{\pi(x_h) = a_h\}(r_h + V_a)]\right| \le \epsilon_{stat}. \tag{8}$$

In the next Lemma, we bound $\mathbf{P}[\mathcal{E}]$, which is the main concentration argument in the proof. The bound involves a new quantity $T_{\max}$ which is the maximum number of calls to `dfslearn`. We will control this quantity later.

**Lemma 14.** *Set*

$$n_{test} \geq \frac{1}{2\epsilon_{stat}^2} \ln\left(\frac{12KHT_{\max}|\mathcal{G}|}{\delta}\right), \qquad n_{train} \geq \frac{16K}{\epsilon_{stat}^2} \ln\left(\frac{12HT_{\max}|\mathcal{G}||\Pi|}{\delta}\right).$$

*Then* $\mathbf{P}[\mathcal{E}] \geq 1 - \delta/2$.

*Proof.* Let us denote the total number of calls to `dfslearn` before the algorithm stops by $N_{\text{dfs}}$ (which is random) and first focus on the $j$-th call to `dfslearn`. Let $\mathcal{B}_j$ be the sigma-field of all samples collected before the $j$th call to `dfslearn` (if it exists, or otherwise the last call to `dfslearn`) and all intrinsic randomness of the algorithm. The current path is denoted by $p_j$ at level $h_j$ and data sets $D'_a$, $\tilde{D}$ collected are denoted by $D'_{j,a}$ and $\tilde{D}_j$ respectively. Consider a fix $a \in \mathcal{A}$ and $g \in \mathcal{G}$ and define

$$Y_{i,j} = \begin{cases} 0 & \text{if } j > N_{dfs} \\ g(x_{h+1}^{(i,j)}) - \mathbf{E}_{p_j \circ a}[g(x_{h+1})] & \text{otherwise} \end{cases} \tag{9}$$

which is well-defined even if $j > N_{dfs}$ and where $x_{h+1}^{(i,j)}$ is the $i$-th sample of $x_{h+1}$ in $D'_{j,a}$. Since $|Y_{i,j}| \leq 1$ and since contexts $x_{h+1}$ are sampled i.i.d. from $p_j \circ a$ conditioned on $p_j$ which is measurable in $\mathcal{B}_j$, we get by Hoeffding's lemma that $\mathbf{E}[\exp(\lambda Y_{i,j})|Y_{1:i-1,j}, \mathcal{B}_j] = \mathbf{E}[\exp(\lambda Y_{i,j})|\mathcal{B}_j] \leq \exp(\lambda^2/2)$ for $\lambda \in \mathbb{R}$. As a result, we have $\mathbf{E}[\exp(\lambda \sum_{i=1}^{n_{test}} Y_{i,j})] = \mathbf{E}[\mathbf{E}[\exp(\lambda \sum_{i=1}^{n_{test}} Y_{i,j})|\mathcal{B}_j]] \leq \exp(n_{test}\lambda^2/2)$ and by Chernoff's bound the following concentration result holds

$$\left|\hat{\mathbf{E}}_{D'_{j,a}}[g(x_{h+1})] - \mathbf{E}_{p_j \circ a}[g(x_{h+1})]\right| \leq \sqrt{\frac{\log(2K|\mathcal{G}|/\delta')}{2n_{test}}}$$

with probability at least $1 - \frac{\delta'}{K|\mathcal{G}|}$ for a fixed $a$ and $g$ and $j$ as long as $j \leq N_{dfs}$. With a union bound over $\mathcal{A}$ and $\mathcal{G}$, the following statement holds: Given a fix $j \in \mathbb{N}$, with probability at least $1 - \delta'$, if $j \leq N_{dfs}$ then for all $g \in \mathcal{G}_{h+1}$ and $a \in \mathcal{A}$

$$\left|\hat{\mathbf{E}}_{D'_{j,a}}[g(x_{h+1})] - \mathbf{E}_{p_j \circ a}[g(x_{h+1})]\right| \leq \sqrt{\frac{\log(2K|\mathcal{G}|/\delta')}{2n_{test}}}.$$

Choosing $n_{test} \geq \frac{1}{2\epsilon_{stat}^2} \ln\left(\frac{12KHT_{\max}|\mathcal{G}|}{\delta}\right)$ and $\delta' = \frac{\delta}{6HT_{\max}}$ allows us to bound the LHS by $\epsilon_{stat}$. In exactly the same way since the data set $\tilde{D}_j$ consists of $n_{train}$ samples that, given $\mathcal{B}_j$, are sampled i.i.d. from $p_j$, we have for all $g \in \mathcal{G}_h$

$$\left|\hat{\mathbf{E}}_{\tilde{D}_j}[g(x_h)] - \mathbf{E}_{p_j}[g(x_h)]\right| \leq \sqrt{\frac{\log(2|\mathcal{G}|/\delta')}{2n_{train}}},$$

with probability $1 - \delta'$ as long as $j \leq N_{dfs}$. As above, our choice of $n_{train}$ ensures that this deviation is bound by $\epsilon_{stat}$.

Finally, for the third inequality we must use Bernstein's inequality. For the random variable $K\mathbf{1}\{\pi(x_h) = a_h\}(r_h + V_{a_h})$, since $a_h$ is chosen uniformly at random, it is not hard to see that both the variance and the range are at most $2K$ (see for example Lemma 14 by Jiang et al. [1]). As such, Bernstein's inequality with a union bound over $\pi \in \Pi$ gives that with probability $1 - \delta'$,

$$\left|(\hat{\mathbf{E}}_{\tilde{D}_j} - \mathbf{E}_{p_j})[K\mathbf{1}\{\pi(x_h) = a_h\}(r_h + V_{a_h})]\right| \leq \sqrt{\frac{4K\log(2|\Pi|/\delta')}{n_{train}}} + \frac{4K}{3n_{train}}\log(2|\Pi|/\delta') \leq \epsilon_{stat},$$

since $\{V_a\}$ and $p_j$ can essentially be considered fixed at the time when $\tilde{D}_j$ is collected (a more formal treatment is analogous to the proof of the first two inequalities). Using a union bound, the deviation bounds (6)–(8) hold for a single call to `dfslearn` with probability $1 - 3\delta'$.

Consider now the event $\mathcal{E}'$ that these bounds hold for the first $T_{\max}$ calls at each level $h$. Applying a union bound let us bound $\mathbf{P}(\mathcal{E}') \geq 1 - 3HT_{\max}\delta' = 1 - \frac{\delta}{2}$. It remains to show that $\mathcal{E}' \subseteq \mathcal{E}$.

First note that in event $\mathcal{E}'$ in the first $T_{\max}$ calls to dfslearn, the algorithm does not call itself recursively if $p \circ a$ leads to a learned state. To see this assume $p \circ a$ leads to a state $s \in \mathcal{S}^{\text{learned}}$. Let $D'_a$ be the data set collected in Line 5 for this action $a$. Since the subsequent state $s \in \mathcal{S}^{\text{learned}}$, then there is a data set $(D, V, \{V_b\}) \in \mathcal{D}_{h+1}$ sampled from this state (we will only use the first two items in the tuple). This means that $D'_a$ and $D$ are two data sets sampled from the same distribution, and as such, we have

$$
\begin{aligned}
V_{opt} - V_{pes} = \hat{\mathbf{E}}_{D'_a}[g_{opt}(x_{h+1}) - g_{pes}(x_{h+1})] &\leq \mathbf{E}_s[g_{opt}(x_{h+1}) - g_{pes}(x_{h+1})] + 2\epsilon_{\text{stat}} \\
&\leq \hat{\mathbf{E}}_D[g_{opt}(x_{h+1}) - g_{pes}(x_{h+1})] + 4\epsilon_{\text{stat}} \\
&\leq V + \phi_{h+1} + \epsilon_{\text{feas}} - V + \phi_{h+1} + \epsilon_{\text{feas}} + 4\epsilon_{\text{stat}} = 2\phi_{h+1} + 4\epsilon_{\text{stat}} + 2\epsilon_{\text{feas}}.
\end{aligned}
$$

The last line holds because the constraints for $g_{opt}$ and $g_{pes}$ include the one based on $(D, V)$ (Line 6), so the expectation of $g_{opt}$ and $g_{pes}$ on $D$ can only differ by the amount of the allowed slackness $2\phi_{h+1}$ and the violations of feasibility $2\epsilon_{\text{feas}}$. Therefore the condition in the if clause is satisfied and the algorithm does not call itself recursively. We here assumed that the constrained optimization problem has an approximately feasible solution but if that is not the case, the if condition is trivially satisfied.

Since the number of learned states per level is bounded by $M$, this means that within the first $T_{\max}$ calls to dfslearn, the algorithm can make recursive calls to the level below at most $M$ times. Further note that for any fixed level $h$ the total number of non-recursive calls to dfslearn is bounded by $MHn_{\text{exp}}$ since MetaAlg has at most $MH$ iterations and in each dfslearn is called $n_{\text{exp}}$ times at each level (but the first). Therefore, in event $\mathcal{E}'$, the total number of calls to dfslearn at any level $h$ is bounded by $MHn_{\text{exp}} + M \leq T_{\max}$ and the statement follows. □

## B.2 Bound on Oracle Calls

*Proof of Theorem 2.* Consider event $\mathcal{E}$ from Definition 13 which by Lemma 14 has probability at least $1 - \delta/2$. VALOR requires two types of nontrivial computations over the function classes. We show that they can be reduced to CSC on $\Pi$ and LP on $\mathcal{G}$ (recall Sec. 3.1), respectively, and hence VALOR is oracle-efficient.

First, Line 12 in dfslearn involves optimizing $V_D(\pi; \{V_a\})$ (Eq. (1)) over $\Pi$, which can be reduced to CSC as follows: We first form tuples $(x^{(i)}, a^{(i)}, y^{(i)})$ from $D$ and $\{V_a\}$ on which $V_D(\pi; \{V_a\})$ depends, where we bind $x_h$ to $x^{(i)}$, $a_h$ to $a^{(i)}$, and $r_h + V_{a_h}$ to $y^{(i)}$. From the tuples, we construct a CSC data set $(x^{(i)}, -[K\mathbf{1}\{a = a^{(i)}\}y^{(i)}]_{a \in \mathcal{A}})$, where the second argument is a $K$-dimensional vector with one non-zero. On this data set, the cost-sensitive risk of any policy (interpreted as a classifier) is exactly $-V_D(\pi; \{V_a\})$, so minimizing risk (which the oracle does) maximizes the original objective.[6]

Second, the optimization in Line 4 in polvalfun can be reduced to CSC with the very same argument, except that we now accumulate all CSC inputs for each data set in $\mathcal{D}_h$. Since $|\mathcal{D}_h| \leq T_{\max}$ is polynomial, the total input size is still polynomial.

Third, the state identity test in Line 6 in dfslearn requires solving the following problem over the function class $\mathcal{G}$:

$$
V_{opt} = \max_{g \in \mathcal{G}} \hat{\mathbf{E}}_{D'}[g(x_h)] \quad \text{(and min for } V_{pes}) \tag{10}
$$

$$
\text{s.t. } V - \phi_h \leq \hat{\mathbf{E}}_D[g(x_h)] \leq V + \phi_h, \forall (D, V) \in \mathcal{D}_h. \tag{11}
$$

The objective and the constraints are linear functionals of $\mathcal{G}$, all empirical expectations involve polynomially many samples, and the number of constraints is $|\mathcal{D}_h| \leq T_{\max}$ which remains polynomial throughout the execution of the algorithm. Therefore, the LP oracle can directly handle this optimization problem.

Altogether, we showed that all non-trivial computations can be reduced to oracle calls with inputs with polynomial description length. It remains to show that the number of calls is bounded. Since there are at most $T_{\max}$ calls to dfslearn at each level $h \in [H]$, the total number of calls to the LP

oracle is $T_{\max}HK$. Similarly, the number of CSC oracle calls from `dfslearn` is at most $T_{\max}H$. In addition, there at at most $MH$ calls to the CSC oracle in `polvalfun`. The statement follows with realizing that $T_{\max} = MHn_{\exp} + M = O\left(\frac{MH}{\epsilon}\ln\left(\frac{MH}{\delta}\right)\right)$.  $\square$

## B.3  Depth First Search and Estimated Values

In this section, we show that in the high-probability event $\mathcal{E}$ (Definition 13), `dfslearn` produces good estimates of optimal values on learned states. The next lemma first quantifies the error in the value estimate at level $h$ in terms of the estimation error of the values of the next time step $\{V_a\}_a$.

**Lemma 15** (Error propagation when learning a state). *Consider a call to* `dfslearn` *with input path $p$ of depth $h$. Assume that all values $\{V_a\}_{a\in\mathcal{A}}$ in Algorithm 3 satisfy $|V_a - V^\star(p\circ a)| \leq \beta$ for some $\beta > 0$. Then in event $\mathcal{E}$, $\tilde{V}$ returned in Line 14 satisfies $|\tilde{V} - V^\star(p)| \leq \epsilon_{stat} + \beta + \epsilon_{sub}$.*

*Proof.* The proof follows a standard analysis of empirical risk minimization (here we are maximizing). Let $\tilde{\pi}$ denote the empirical risk maximizer in Line 12 and let $\pi^\star$ denote the globally optimal policy (which is in our class due to realizability). Then

$$\tilde{V} \leq \hat{\mathbf{E}}_{\tilde{D}}[K\mathbf{1}\{\tilde{\pi}(x_h) = a_h\}(r_h + V_{a_h})] \leq \mathbf{E}_p[K\mathbf{1}\{\tilde{\pi}(x_h) = a_h\}(r_h + V_{a_h})] + \epsilon_{stat}$$
$$\leq \mathbf{E}_p[K\mathbf{1}\{\tilde{\pi}(x_h) = a_h\}(r_h + g^\star(x_{h+1}))] + \beta + \epsilon_{stat}$$
$$\leq \mathbf{E}_p[K\mathbf{1}\{\pi^\star(x_h) = a_h\}(r_h + g^\star(x_{h+1}))] + \beta + \epsilon_{stat} = V^\star(s) + \beta + \epsilon_{stat}.$$

The first inequality is the deviation bound, which holds in event $\mathcal{E}$. The second inequality is based on the precondition on $\{V_a\}_{a\in\mathcal{A}}$, linearity of expectation, and the realizability property of $g^\star_{h+1}$. The third inequality uses that $\pi^\star$ is the global and point-wise maximizer of the long-term expected reward, which is precisely $r_h + g^\star$.

Similarly, we can lower bound $\tilde{V}$ by

$$\tilde{V} = \hat{\mathbf{E}}_{\tilde{D}}[K\mathbf{1}\{\tilde{\pi}(x_h) = a_h\}(r_h + V_{a_h})] - \epsilon_{sub} \geq \hat{\mathbf{E}}_{\tilde{D}}[K\mathbf{1}\{\pi^\star(x_h) = a_h\}(r_h + V_{a_h})] - \epsilon_{sub}$$
$$\geq \mathbf{E}_p[K\mathbf{1}\{\pi^\star(x_h) = a_h\}(r_h + V_{a_h})] - \epsilon_{stat} - \epsilon_{sub}$$
$$\geq \mathbf{E}_p[K\mathbf{1}\{\pi^\star(x_h) = a_h\}(r_h + g^\star(x_{h+1}))] - \epsilon_{stat} - \beta - \epsilon_{sub} = V^\star(s) - \epsilon_{stat} - \beta - \epsilon_{sub}.$$

Here we first use $\tilde{V}$ is optimal up to $\epsilon_{sub}$ and then that $\tilde{\pi}$ is the empirical maximizer. Subsequently, we leveraged the deviation bounds of event $\mathcal{E}$ and finally used the assumption about the estimation accuracy from the level below. This proves the claim.  $\square$

The goal of the proof is to apply the above lemma inductively so that we can learn all of the values to reasonable accuracy. Before doing so, we need to quantify the estimation error when $V_a$ is set in Line 8 of the algorithm without a recursive call.

**Lemma 16** (Error when not recursing). *Consider a call to* `dfslearn` *with input path $p$ of depth $h$. If $g^\star$ is feasible for Line 6 of* `dfslearn` *and $V_a$ is set in Line 8 of Algorithm 3, then in event $\mathcal{E}$, the value $V_a = \frac{V_{opt} + V_{pes}}{2}$ satisfies $|V_a - V^\star(p\circ a)| \leq \phi_{h+1} + 3\epsilon_{stat} + \epsilon_{feas} + \epsilon_{sub}$.*

*Proof.* Recall that $D'_a$ is the data set sampled in Line 5 for the particular action $a$ in consideration. Since $g^\star_{h+1}$ is feasible for both $V_{opt}$ and $V_{pes}$, we have

$$V_{pes} - \epsilon_{sub} = \hat{\mathbf{E}}_{D'_a}[g_{pes}(x_{h+1})] - \epsilon_{sub} \leq \hat{\mathbf{E}}_{D'_a}[g^\star(x_{h+1})] \leq \hat{\mathbf{E}}_{D'_a}[g_{opt}(x_{h+1})] + \epsilon_{sub} = V_{opt} + \epsilon_{sub}.$$

Without loss of generality, we can assume that $V_{pes} \leq V_{opt}$, otherwise we can just exchange them. This implies that $0 \leq V_{opt} - V_a = V_a - V_{pes} = \frac{V_{opt} - V_{pes}}{2} \leq \phi_{h+1} + 2\epsilon_{stat} + \epsilon_{feas}$. Therefore,

$$\hat{\mathbf{E}}_{D'}[g^\star(x_{h+1})] - V_a \leq V_{opt} - V_a + \epsilon_{sub} = \frac{V_{opt} - V_{pes}}{2} + \epsilon_{sub} \leq \phi_{h+1} + 2\epsilon_{stat} + \epsilon_{feas} + \epsilon_{sub}.$$

$$V_a - \hat{\mathbf{E}}_{D'}[g^\star(x_{h+1})] \leq V_a - V_{pes} + \epsilon_{sub} = \frac{V_{opt} - V_{pes}}{2} + \epsilon_{sub} \leq \phi_{h+1} + 2\epsilon_{stat} + \epsilon_{feas} + \epsilon_{sub}.$$

By the triangle inequality

$$|V_a - V^\star(p\circ a)| \leq |\hat{\mathbf{E}}_{D'_a}[g^\star(x_{h+1})] - V_a| + |\hat{\mathbf{E}}_{D'_a}[g^\star(x_{h+1})] - V^\star(p\circ a)|$$
$$\leq \phi_{h+1} + 3\epsilon_{stat} + \epsilon_{feas} + \epsilon_{sub}.$$

The last inequality is the concentration statement, which holds in event $\mathcal{E}$.  $\square$

We now are able to apply Lemma 15 inductively in combination with Lemma 16 to obtain the main result of `dfslearn` in this section.

**Proposition 17** (Accuracy of learned values). *Assume the realizability condition $g^\star \in \mathcal{G}_h$. Set $\phi_h = (H + 1 - h)(6\epsilon_{stat} + 2\epsilon_{sub} + \epsilon_{feas})$ for all $h \in [H]$. Then under event $\mathcal{E}$, for any level $h \in [H]$ and any state $s \in \mathcal{S}_h$ all triplets $(D, V, \{V_a\}) \in \mathcal{D}_h$ associated with state $s$ (formally with paths $p$ that lead to $s$) satisfy*

$$|V - V^\star(s)| \le \phi_h - 2\epsilon_{stat}, \qquad |V_a - V^\star(s \circ a)| \le \phi_{h+1} + 3\epsilon_{stat} + \epsilon_{feas} + \epsilon_{sub}.$$

*Moreover, under event $\mathcal{E}$, we have $g^\star$ is feasible for Line 6 of `dfslearn` for all $h$, at all times.*

*Proof.* We prove this statement by induction over $h$. For $h = H+1$ the statement holds trivially since $\mathcal{G}_{H+1} = \{g^\star_{h+1}\}$ the constant $0$ function is the only function in $\mathcal{G}_{H+1}$ and therefore the algorithm always returns on Line 8 and never calls level $H + 1$ recursively.

Consider now some data set $(\tilde{D}, \tilde{V}, \{V_a\}) \in \mathcal{D}_h$ at level $h$ associated with state $s \in \mathcal{S}_h$. This data set was obtained by calling `dfslearn` at some path $p$ (pointing to state $s$). Since when we added this data set, we have not yet exhausted the budget of $T_{\max}$ calls to `dfslearn` (by the preconditions of the lemma), we have that the once we reach Line 11 the inductive hypothesis applies for all data sets at level $h + 1$ (which may have been added by recursive calls of this execution). Each of the $V_a$ values can be set in one of two ways.

1. The algorithm did not make a recursive call. Since by the inductive assumption $g^\star$ is feasible for Line 6 of `dfslearn`, we can apply Lemma 16 and get that

$$|V_a - V^\star(s \circ a)| \le \phi_{h+1} + 3\epsilon_{stat} + \epsilon_{feas} + \epsilon_{sub}.$$

2. The algorithm made a recursive call. Since the value returned was added as a data set at level $h + 1$, it satisfies the inductive assumption

$$|V_a - V^\star(s \circ a)| \le \phi_{h+1} - 2\epsilon_{stat}.$$

This demonstrates the second inequality in the inductive step. For the first, applying Lemma 15 with $\beta = \phi_{h+1} + 3\epsilon_{stat} + \epsilon_{feas} + \epsilon_{sub}$, we get that $|\tilde{V} - V^\star(s)| \le \phi_{h+1} + 4\epsilon_{stat} + \epsilon_{feas} + 2\epsilon_{sub} = \phi_h - 2\epsilon_{stat}$, by definition of $\phi_h$. Finally, this also implies that $|\tilde{V} - \hat{\mathbf{E}}_{\tilde{D}}[g^\star_h(x_h)]| \le |\tilde{V} - V^\star(s)| + |\hat{\mathbf{E}}_{\tilde{D}}[g^\star_h(x_h)] - V^\star(s)| \le \phi_h$ which means that $g^\star$ is still feasible. $\qquad\square$

## B.4 Policy Performance

In this section, we bound the quality of the policy returned by `polvalfun` in the good event $\mathcal{E}$ by using the fact that `dfslearn` produces accurate estimates of the optimal values (previous section). Before we state the main result of this section in Proposition 19, we prove the following helpful lemma. This Lemma is essentially Lemma 4.3 in Ross and Bagnell [20].

**Lemma 18.** *The suboptimality of a policy $\pi$ can be written as*

$$V^\star - V^\pi = \mathbf{E}\left[\sum_{h=1}^{H}(V^\star(s_h) - Q^\star(s_h, \pi_h)) \mid a_h \sim \pi_h\right].$$

*Proof.* The difference of values of a policy $\pi$ compared to the optimal policy in a certain state $s \in \mathcal{S}_h$ can be expressed as

$$
\begin{aligned}
V^\star(s) - V^\pi(s) &= V^\star(s) - \mathbf{E}_s[K\mathbf{1}\{\pi_h(x_h) = a_h\}(r_h + V^\pi(x_{h+1}))] \\
&= V^\star(s) - \mathbf{E}_s[K\mathbf{1}\{\pi_h(x_h) = a_h\}(r_h + V^\star(x_{h+1}) - V^\star(x_{h+1}) + V^\pi(x_{h+1}))] \\
&= V^\star(s) - Q^\star(s, \pi_h) + \mathbf{E}_s[K\mathbf{1}\{\pi_h(x_h) = a_h\}(V^\star(x_{h+1}) - V^\pi(x_{h+1}))] \\
&= V^\star(s) - Q^\star(s, \pi_h) + \mathbf{E}_s[V^\star(x_{h+1}) - V^\pi(x_{h+1}) \mid a_h \sim \pi_h].
\end{aligned}
$$

Therefore, by applying this equality recursively, the suboptimality of $\pi$ can be written as

$$V^\star(s) - V^\pi = \mathbf{E}\left[\sum_{h=1}^{H}(V^\star(s_h) - Q^\star(s_h, \pi_h)) \mid a_h \sim \hat{\pi}_h\right]. \qquad\square$$

Now we may bound the policy suboptimality.

**Proposition 19.** *Assume $g_h^\star \in \mathcal{G}_h$ and the we are in event $\mathcal{E}$. Recall the definition $\phi_h = (H + 1 - h)(6\epsilon_{stat} + 2\epsilon_{sub} + \epsilon_{feas})$ for all $h \in [H]$. Then the policy $\hat{\pi} = \hat{\pi}_{1:H}$ returned by* `polvalfun` *satisfies*

$$V^{\hat{\pi}} \geq V^\star - p_{ul}^{\hat{\pi}} - 2H^2 T_{\max}(7\epsilon_{stat} + 3\epsilon_{sub} + 2\epsilon_{feas})$$

*where $p_{ul}^{\hat{\pi}} = \mathbf{P}(\exists h \in [H] \,:\, s_h \notin \mathcal{S}^{learned} \mid a_{1:H} \sim \hat{\pi})$ is the probability of hitting an unlearned state when following $\hat{\pi}$.*

*Proof.* To bound the suboptimality of the learned policy, we bound the difference of how much following $\hat{\pi}_h$ for one time step can hurt per state using Proposition 17. For a state $s \in \mathcal{S}^{learned}$ at level $h$, we have

$$
\begin{aligned}
&V^\star(s) - Q^\star(s, \hat{\pi}_h) \\
&= \mathbf{E}_s[K(\mathbf{1}\{\pi_h^\star(x_h) = a_h\} - \mathbf{1}\{\hat{\pi}_h(x_h) = a_h\})(r_h + g_{h+1}^\star(x_{h+1})] \\
&\leq \sum_{s \in \mathcal{S}_h^{learned}} \mathbf{E}_s[K(\mathbf{1}\{\pi_h^\star(x_h) = a_h\} - \mathbf{1}\{\hat{\pi}_h(x_h) = a_h\})(r_h + g_{h+1}^\star(x_{h+1})] \\
&\leq \sum_{(s,\_,\{V_a\}) \in \mathcal{D}_h} \Big( \mathbf{E}_s[K(\mathbf{1}\{\pi_h^\star(x_h) = a_h\} - \mathbf{1}\{\hat{\pi}_h(x_h) = a_h\})(r_h + V_{a_h})] \\
&\qquad\qquad + 2\phi_{h+1} + 6\epsilon_{stat} + 2\epsilon_{feas} + 2\epsilon_{sub}\Big) \\
&\leq \sum_{(\tilde{D},\_,\{V_a\}) \in \mathcal{D}_h} \Big( \mathbf{E}_{\tilde{D}}[K(\mathbf{1}\{\pi_h^\star(x_h) = a_h\} - \mathbf{1}\{\hat{\pi}_h(x_h) = a_h\})(r_h + V_{a_h})] \\
&\qquad\qquad + 2\phi_{h+1} + 8\epsilon_{stat} + 2\epsilon_{feas} + 2\epsilon_{sub}\Big) \\
&\leq 2|\mathcal{D}_h|(\phi_{h+1} + 4\epsilon_{stat} + \epsilon_{feas} + 2\epsilon_{sub}).
\end{aligned}
$$

Here the first identity is based on expanding definitions. For the first inequality, we use that $s \in \mathcal{S}^{learned}$ and also that $\pi^\star$ simultaneously maximizes the long term reward from all states, so the terms we added in are all non-negative. In the second inequality, we introduce the notation $(s, \_, \{V_a\}) \in \mathcal{D}_h$ to denote a data set in $\mathcal{D}_h$ associated with state $s$ with successor values $\{V_a\}$. For this inequality we use Proposition 17 to control the deviation of the successor values. The third inequality uses the deviation bound that holds in event $\mathcal{E}$.

Since per `dfslearn` call, only one data set can be added to $\mathcal{D}_h$, the magnitude $|\mathcal{D}_h| \leq T_{\max}$ is bounded by the total number of calls to `dfslearn` at each level. Using Lemma 18, the suboptimality of $\hat{\pi}$ is therefore at most

$$
\begin{aligned}
V^\star - V^{\hat{\pi}} &\leq p_{ul}^{\hat{\pi}} + (1 - p_{ul}^{\hat{\pi}}) \sum_{h=1}^{H} 2|\mathcal{D}_h|(\phi_{h+1} + 4\epsilon_{stat} + 2\epsilon_{sub} + \epsilon_{feas}) \\
&\leq p_{ul}^{\hat{\pi}} + 2HT_{\max}(4\epsilon_{stat} + 2\epsilon_{sub} + \epsilon_{feas}) + 2T_{\max} \sum_{h=1}^{H} \phi_{h+1} \\
&\leq p_{ul}^{\hat{\pi}} + 2HT_{\max}(4\epsilon_{stat} + 2\epsilon_{sub} + \epsilon_{feas}) + 2T_{\max}(6\epsilon_{stat} + 2\epsilon_{sub} + \epsilon_{feas}) \sum_{h=1}^{H}(H - h) \\
&\leq p_{ul}^{\hat{\pi}} + 2HT_{\max}(4\epsilon_{stat} + 2\epsilon_{sub} + \epsilon_{feas}) + H^2 T_{\max}(6\epsilon_{stat} + 2\epsilon_{sub} + \epsilon_{feas}) \\
&\leq p_{ul}^{\hat{\pi}} + 14H^2 T_{\max}\epsilon_{stat} + 6H^2 T_{\max}\epsilon_{sub} + 3H^2 T_{\max}\epsilon_{feas}.
\end{aligned}
$$

This argument is similar to the proof of Lemma 8 in Krishnamurthy et al. [17]. Note that we introduce the dependency on $T_{\max}$ since we perform joint policy optimization, which will degrade the sample complexity. $\qquad\square$

## B.5  Meta-Algorithm Analysis

Now that we have the main guarantees for `dfslearn` and `polvalfun`, we may turn to the analysis of `MetaAlg`.

**Lemma 20.** *Consider running* `MetaAlg` *with* `dfslearn` *and* `polvalfun` *(Algorithm 1 + 2 + 3) with parameters*

$$n_{exp} \geq \frac{8}{\epsilon} \ln\left(\frac{4MH}{\delta}\right), \quad n_{eval} \geq \frac{32}{\epsilon^2} \ln\left(\frac{8MH}{\delta}\right), \quad \epsilon_{stat} = \epsilon_{sub} = \epsilon_{feas} = \frac{\epsilon}{2^6 7 H^2 T_{\max}}$$

*Then with probability at least $1 - \delta$, `MetaAlg` returns a policy that is at least $\epsilon$-optimal after at most $MK$ iterations.*

*Proof.* First apply Lemma 14 so that the good event $\mathcal{E}$ holds, except with probability $\delta/2$.

In the event $\mathcal{E}$, since before the first execution of `polvalfun`, we called `dfslearn`($\varnothing$), by Proposition 17, we know that $|\hat{V}^\star - V^\star| \leq \phi_1 - 2\epsilon_{\text{stat}}$ where $\hat{V}^\star$ is the value stored in the only dataset associated with the root. This value does not change for the remainder of the algorithm, and the choice of $\epsilon_{\text{stat}}, \phi$ ensure that

$$|\hat{V}^\star - V^\star| \leq \phi_1 - 2\epsilon_{\text{stat}} = 6H\epsilon_{\text{stat}} + 2H\epsilon_{\text{sub}} + H\epsilon_{\text{feas}} - 2\epsilon_{\text{stat}} \leq \epsilon/8.$$

This is true for all executions of `polvalfun` (formally all $\hat{V}^{(k)}$ values). Next, since we perform at most $MH$ iterations of the loop in `MetaAlg`, we consider at most $MH$ policies. Via a standard application of Hoeffding's inequality, with probability $1 - \delta/4$, we have that for all $k \in [MH]$

$$|\hat{V}^{\hat{\pi}_k} - V^{\hat{\pi}_k}| \leq \sqrt{\frac{\log(8MH/\delta)}{2n_{\text{eval}}}}.$$

The choice of $n_{\text{eval}}$ ensure that this is at most $\epsilon/8$. With these two bounds, if `MetaAlg` terminates, the termination condition implies that

$$V^\star - V^{\hat{\pi}^{(k)}} \leq \hat{V}^{(k)} - \hat{V}^{\hat{\pi}^{(k)}} + \frac{\epsilon}{4} \leq \frac{3}{4}\epsilon \leq \epsilon$$

and hence the returned policy is $\epsilon$-optimal.

On the other hand, if the algorithm does not terminate in iteration $k$, we have that $\hat{V}^{(k)} - \hat{V}^{\hat{\pi}^{(k)}} > \frac{\epsilon}{2}$ and therefore

$$V^\star - V^{\hat{\pi}^{(k)}} \geq \hat{V}^{(k)} - \hat{V}^{\hat{\pi}^{(k)}} - \frac{\epsilon}{4} \geq \frac{\epsilon}{4}.$$

We now use this fact with Proposition 19 to argue that the policy $\hat{\pi}^{(k)}$ must visit an unlearned state with sufficient probability. Under the conditions here, applying Proposition 19, we get that

$$\frac{\epsilon}{4} \leq V^\star - V^{\hat{\pi}^{(k)}} \leq p_{ul}^{\hat{\pi}^{(k)}} + 2T_{\max}H^2(7\epsilon_{\text{stat}} + 3\epsilon_{\text{sub}} + 2\epsilon_{\text{feas}}).$$

With the choice of $\epsilon_{\text{stat}}$, rearranging this inequality reveals that $p_{ul}^{\hat{\pi}^{(k)}} \geq \epsilon/8 > 0$. Hence, if the algorithm does not terminate there must be at least one unlearned state, i.e., $\mathcal{S} \setminus \mathcal{S}^{\text{learned}} \neq \emptyset$.

For the last step of the proof, we argue that since $p_{ul}^{\hat{\pi}^{(k)}}$ is large, the probability of reaching an unlearned state is high, and therefore the additional calls to `dfslearn` in Line 11 with high probability will visit a new state, which we will then learn. Specifically, we will prove that on every non-terminal iteration of `MetaAlg`, we learn at least one previously unlearned state. With this fact, since there are at most $MH$ states, the algorithm must terminate and return a near-optimal policy after at most $MH$ iterations.

In a non-terminal iteration $k$, the probability that we do not hit an unlearned state in Line 11 is

$$(1 - p_{ul}^{\hat{\pi}^{(k)}})^{n_{\exp}} \leq (1 - \epsilon/8)^{n_{\exp}} \leq \exp(-\epsilon n_{\exp}/8).$$

This follows from independence of the $n_{\exp}$ trajectories sampled from $\hat{\pi}^{(k)}$. $n_{\exp} \geq \frac{8}{\epsilon} \ln\left(\frac{4MH}{\delta}\right)$ ensures that the probability of not hitting unlearned states in any of the $MH$ iterations is at most $\delta/4$.

In total, except with probability $\delta/2 + \delta/4 + \delta/4$ (for the three events we considered above), on every iteration, either the algorithm finds a near optimal policy and returns it, or it visits a previously unlearned state, which subsequently becomes learned. Since there are at most $MH$ states, this proves that with probability at least $1 - \delta$, the algorithm returns a policy that is at most $\epsilon$-suboptimal. $\quad\square$

## B.6 Proof of Sample Complexity: Theorem 3

We now have all parts to complete the proof of Theorem 3. For the calculation, we instantiate all the parameters as

$$\epsilon_{\text{stat}} = \epsilon_{\text{sub}} = \epsilon_{\text{feas}} = \frac{\epsilon}{2^6 7 H^2 T_{\max}},$$

$$\phi_h = (H - h + 1)(6\epsilon_{\text{stat}} + 2\epsilon_{\text{sub}} + \epsilon_{\text{feas}}), \quad T_{\max} = MH n_{\exp} + M,$$

$$n_{\text{test}} = \frac{\log\left(12KHT_{\max}|\mathcal{G}|/\delta\right)}{2\epsilon_{\text{stat}}^2}, \quad n_{\text{train}} = \frac{16K \log(12HT_{\max}|\mathcal{G}||\Pi|/\delta)}{\epsilon_{\text{stat}}^2},$$

$$n_{\exp} = \frac{8\log(4MH/\delta)}{\epsilon}, \quad n_{\text{eval}} = \frac{32\log(8MH/\delta)}{\epsilon^2}.$$

These settings suffice to apply all of the above lemmas and therefore with these settings the algorithm outputs a policy that is at most $\epsilon$-suboptimal, except with probability $\delta$. For the sample complexity, since $T_{\max}$ is an upper bound on the number of data sets we collect (because $T_{\max}$ is an upper bound on the number of execution of dfslearn at any level), and we also $n_{\text{eval}}$ trajectories for each of the $MH$ iterations of MetaAlg, the total sample complexity is

$$HT_{\max}n_{\text{train}} + KHT_{\max}n_{\text{test}} + MH n_{\text{eval}}$$

$$= O\left(\frac{T_{\max}^3 KH^5}{\epsilon^2} \log\left(\frac{MKH}{\epsilon\delta}|\mathcal{G}||\Pi|\log(MH/\delta)\right) + \frac{MH}{\epsilon^2}\log(MH/\delta)\right)$$

$$= O\left(\frac{M^3 KH^8}{\epsilon^5} \log^3(MH/\delta) \log\left(\frac{MKH}{\epsilon\delta}|\mathcal{G}||\Pi|\log(MH/\delta)\right)\right).$$

This proves the theorem. $\square$

## B.7 Extension: VALOR with Constrained Policy Optimization

We note that Theorem 3 suffers relatively high sample complexity compared to the original LSVEE. The issue is that VALOR pools all the data sets together for policy optimization (Algorithm 2). This implicitly weights all data sets uniformly, and allows some undesired trade-off: the policy that maximizes the objective could sacrifice significant amount of value on one data set (for some hidden state) to gain slightly more value on many others, only to find out later that the sacrificed state is visited very often during execution. This is the well-known distribution mismatch issue of reinforcement learning.

To address this issue and attain better sample complexity results, Algorithm 4 shows an alternative to the policy optimization component of VALOR in Algorithm 2. Instead of using an unconstrained optimization problem, it finds the policy through a feasibility problem, and hence avoid the undesired trade-off mentioned above. The computation can be implemented by the multi data set classification oracle defined in Section A.

---

**Algorithm 4:** Constrained policy optimization with local values

1 **Function** polvalfun()
2 $\quad \hat{V}^\star \leftarrow V$ associated with only dataset in $\mathcal{D}_1$;
3 $\quad$ **for** $h = 1 : H$ **do**
4 $\quad\quad$ Pick $\hat{\pi}_h$ such that the following constraints are violated at most $\epsilon_{\text{feas}}$ for all
$\quad\quad (D, V, \{V_a\}_a) \in \mathcal{D}_h \ : \ \hat{\mathbf{E}}_D[K\mathbf{1}\{\pi(x_h) = a_h\}(r_h + V_{a_h})] \geq V - 2\phi_h + 4\epsilon_{\text{stat}} + \epsilon_{\text{sub}}$ ;
5 $\quad$ **return** $\hat{\pi}_{1:H}, \hat{V}^\star$;

---

Below, we prove a stronger version of Proposition 19 (which is for Algorithm 2) for this approach based on feasibility. First, we show that $\pi^\star$ is always a feasible choice in Line 4 in event $\mathcal{E}$.

**Lemma 21.** *Assume $g^\star \in \mathcal{G}_h$, $\pi^\star \in \Pi_h$ and $\phi_h = (H + 1 - h)(6\epsilon_{\text{stat}} + 2\epsilon_{\text{sub}} + \epsilon_{\text{feas}})$ for all $h \in [H]$. Then $\pi^\star$ is a valid choice in Line 4 of* polvalfun *in Algorithm 4 in event $\mathcal{E}$.*

*Proof.* Consider a single data set $(D, V, \{V_a\}_a) \in \mathcal{D}_h$ that is associated with state $s \in \mathcal{S}_h$. Using Proposition 17, we can bound the deviation of the optimal policy for each constraint as

$$V - \hat{\mathbf{E}}_D[K\mathbf{1}\{\pi^\star(x_h) = a_h\}(r_h + V_{a_h})]$$

$$\leq V^\star(s) + \phi_h - 2\epsilon_{\text{stat}} - \hat{\mathbf{E}}_D[K\mathbf{1}\{\pi^\star(x_h) = a_h\}(r_h + V_{a_h})]$$

$$\leq V^\star(s) + \phi_h - 2\epsilon_{\text{stat}} - \mathbf{E}_s[K\mathbf{1}\{\pi^\star(x_h) = a_h\}(r_h + V_{a_h})] + \epsilon_{\text{stat}}$$

$$\leq V^\star(s) + \phi_h + 2\epsilon_{\text{stat}} - \mathbf{E}_s[K\mathbf{1}\{\pi^\star(x_h) = a_h\}(r_h + V^\star(s \circ a_h))] + \phi_{h+1} + \epsilon_{\text{sub}} + \epsilon_{\text{feas}}$$

$$= \phi_h + 2\epsilon_{\text{stat}} + \phi_{h+1} + \epsilon_{\text{sub}} + \epsilon_{\text{feas}} = 2\phi_h - 4\epsilon_{\text{stat}} - \epsilon_{\text{sub}}.$$

Here we first used that $V$ is close to the optimal value $V^\star(s)$, the deviation bounds next and finally leveraged that $V_a$ is a good estimate. Since that inequality holds for all constraints, $\pi^\star$ is feasible. □

We now show that Algorithm 4 produces policies with a better guarantees than its unconstrained counterpart. The difference is that we eliminate the $T_{\max}$ term in the error bound.

**Proposition 22** (Improvement over Proposition 19). *Assume $g^\star \in \mathcal{G}_h$ and that we are in event $\mathcal{E}$. Recall the definition $\phi_h = (H + 1 - h)(6\epsilon_{stat} + 2\epsilon_{sub} + \epsilon_{feas})$ for all $h \in [H]$. Then the policy $\hat{\pi} = \hat{\pi}_{1:H}$ returned by* `polvalfun` *in Algorithm 4 satisfies*

$$V^{\hat{\pi}} \geq V^\star - p_{ul}^{\hat{\pi}} - 32H^2(\epsilon_{stat} + \epsilon_{feas} + \epsilon_{sub})$$

*where $p_{ul}^{\hat{\pi}} = \mathbf{P}(\exists h \in [H] \ : \ s_h \notin \mathcal{S}^{learned} \mid a_{1:H} \sim \hat{\pi})$ is the probability of hitting an unlearned state when following $\hat{\pi}$.*

*Proof.* We bound the difference of how much following $\hat{\pi}_h$ for one time step can hurt per state using Proposition 17. First note that by Lemma 21, the optimization problem always has a feasible solution in event $\mathcal{E}$, so $\hat{\pi}_h$ is well defined. For a state $s \in \mathcal{S}_h^{\text{learned}}$, we have

$$V^\star(s) - Q^\star(s, \hat{\pi}_h)$$

$$= \mathbf{E}_s[K(\mathbf{1}\{\pi_h^\star(x_h) = a_h\} - \mathbf{1}\{\hat{\pi}_h(x_h) = a_h\})(r_h + g_{h+1}^\star(x_{h+1})]$$

$$\leq \mathbf{E}_s[K(\mathbf{1}\{\pi_h^\star(x_h) = a_h\} - \mathbf{1}\{\hat{\pi}_h(x_h) = a_h\})(r_h + V_{a_h})] + 2\phi_{h+1} + 6\epsilon_{\text{stat}} + 2\epsilon_{\text{sub}} + 2\epsilon_{\text{feas}}$$

$$\leq \hat{\mathbf{E}}_D[K(\mathbf{1}\{\pi_h^\star(x_h) = a_h\} - \mathbf{1}\{\hat{\pi}_h(x_h) = a_h\})(r_h + V_{a_h})] + 2\phi_{h+1} + 8\epsilon_{\text{stat}} + 2\epsilon_{\text{sub}} + 2\epsilon_{\text{feas}}$$

$$\leq V + \epsilon_{\text{sub}} - V + 2\phi_h - 4\epsilon_{\text{stat}} - \epsilon_{\text{sub}} + \epsilon_{\text{feas}} + 2\phi_{h+1} + 8\epsilon_{\text{stat}} + 2\epsilon_{\text{sub}} + 2\epsilon_{\text{feas}}$$

$$= 4\phi_{h+1} + 16\epsilon_{\text{stat}} + 5\epsilon_{\text{feas}} + 6\epsilon_{\text{sub}} = 4\phi_h - 8\epsilon_{\text{stat}} - 2\epsilon_{\text{sub}} + \epsilon_{\text{feas}}.$$

Here $(D, V, \{V_a\})$ is one of the data sets in $\mathcal{D}_h$ that is associated with $s$, which has optimal policy value $V$ by construction. We first applied definitions and then used that $V_a$ are good value estimates. Subsequently we applied the deviation bounds and finally leveraged the definition of $V$ and the approximate feasibility of $\hat{\pi}_h$. Using Lemma 18, the suboptimality of $\hat{\pi}$ is therefore at most

$$V^\star - V^{\hat{\pi}} \leq p_{ul}^{\hat{\pi}} + (1 - p_{ul}^{\hat{\pi}}) \sum_{h=1}^{H}(4\phi_{h+1} + 16\epsilon_{\text{stat}} + 5\epsilon_{\text{feas}} + 6\epsilon_{\text{sub}})$$

$$\leq p_{ul}^{\hat{\pi}} + 16H\epsilon_{\text{stat}} + 6H\epsilon_{\text{sub}} + 5H\epsilon_{\text{feas}} + 4\sum_{h=1}^{H}\phi_{h+1}$$

$$\leq p_{ul}^{\hat{\pi}} + 16H\epsilon_{\text{stat}} + 6H\epsilon_{\text{sub}} + 5H\epsilon_{\text{feas}} + 4(6\epsilon_{\text{stat}} + 2\epsilon_{\text{sub}} + \epsilon_{\text{feas}})\sum_{h=1}^{H}(H - h)$$

$$\leq p_{ul}^{\hat{\pi}} + 16H\epsilon_{\text{stat}} + 6H\epsilon_{\text{sub}} + 5H\epsilon_{\text{feas}} + 2H^2(6\epsilon_{\text{stat}} + 2\epsilon_{\text{sub}} + \epsilon_{\text{feas}})$$

$$\leq p_{ul}^{\hat{\pi}} + 32H^2(\epsilon_{\text{stat}} + \epsilon_{\text{feas}} + \epsilon_{\text{sub}}). \qquad \square$$

Using this improved policy guarantee, we obtain a tighter analysis of `MetaAlg` that does not have a dependency on $T_{\max}$ in $\epsilon_{\text{stat}}$.

**Lemma 23.** *Consider running* `MetaAlg` *with* `dfslearn` *and* `polvalfun` *(Algorithm 1 + 4 + 3) with parameters*

$$n_{exp} \geq \frac{8}{\epsilon}\ln\left(\frac{4MH}{\delta}\right), \quad n_{eval} \geq \frac{32}{\epsilon^2}\ln\left(\frac{8MH}{\delta}\right), \quad \epsilon_{stat} = \epsilon_{feas} = \epsilon_{sub} = \frac{\epsilon}{2^{10}H^2}.$$

*Then with probability at least $1 - \delta$, $\mathtt{MetaAlg}$ returns a policy that is at least $\epsilon$-optimal after at most $MK$ iterations.*

*Proof.* The proof is identical to the proof of Lemma 20 except using Proposition 22 in place of Proposition 19, and using Lemma 21 to guarantee that the optimization problem in Line 4 is always feasible, in event $\mathcal{E}$. □

Finally, we are ready to assemble all statements to the following sample-complexity bound:

**Theorem 24.** *Consider a Markovian CDP with deterministic dynamics over $M$ hidden states, as described in Section 3. When $\pi^\star \in \Pi$ and $g^\star \in \mathcal{G}$ (Assumptions 7 and 8 hold), for any $\epsilon, \delta \in (0, 1)$, the local value algorithm with constrained policy optimization (Algorithm 1 + 4 + 3) returns a policy $\pi$ such that $V^\star - V^\pi \leq \epsilon$ with probability at least $1 - \delta$, after collecting at most $\tilde{O}\left(\frac{MKH^6}{\epsilon^3}\log(|\mathcal{G}||\Pi|/\delta)\log(1/\delta)\right)$ trajectories.*

*Proof.* We now have all parts to complete the proof of Theorem 3. For the calculation, we instantiate all the parameters as

$$\epsilon_{\text{stat}} = \epsilon_{\text{feas}} = \epsilon_{\text{sub}} = \frac{\epsilon}{2^{10}H^2}, \quad \phi_h = (H - h + 1)(6\epsilon_{\text{stat}} + 2\epsilon_{\text{sub}} + \epsilon_{\text{feas}}),$$

$$T_{\max} = MHn_{\exp} + M,$$

$$n_{\text{test}} = \frac{\log\left(12KHT_{\max}|\mathcal{G}|/\delta\right)}{2\epsilon_{\text{stat}}^2}, \quad n_{\text{train}} = \frac{16K\log(12HT_{\max}|\mathcal{G}||\Pi|/\delta)}{\epsilon_{\text{stat}}^2},$$

$$n_{\exp} = \frac{8\log(4MH/\delta)}{\epsilon}, \quad n_{\text{eval}} = \frac{32\log(8MH/\delta)}{\epsilon^2}.$$

These settings suffice to apply all of the above lemmas for these algorithms and therefore with these settings the algorithm outputs a policy that is at most $\epsilon$-suboptimal, except with probability $\delta$. For the sample complexity, since $T_{\max}$ is an upper bound on the number of data sets we collect (because $T_{\max}$ is an upper bound on the number of execution of $\mathtt{dfslearn}$ at any level), and we also $n_{\text{eval}}$ trajectories for each of the $MH$ iterations of $\mathtt{MetaAlg}$, the total sample complexity is

$$HT_{\max}n_{\text{train}} + KHT_{\max}n_{\text{test}} + MHn_{\text{eval}}$$

$$= O\left(\frac{T_{\max}KH^5}{\epsilon^2}\log\left(\frac{MKH}{\epsilon\delta}|\mathcal{G}||\Pi|\log(MH/\delta)\right) + \frac{MH}{\epsilon^2}\log(MH/\delta)\right)$$

$$= O\left(\frac{MKH^6}{\epsilon^3}\log(MH/\delta)\log\left(\frac{MKH}{\epsilon\delta}|\mathcal{G}||\Pi|\log(MH/\delta)\right)\right). \qquad \square$$

## C Alternative Algorithms

**Theorem 25** (Informal statement). *Under Assumption 9 or Assumptions 8+10, there exist oracle-efficient algorithms with polynomial sample complexity in CDPs (contextual decision processes) with deterministic dynamics over small hidden states. These algorithms do not store or use local values.*

### C.1 Algorithm with Two-Sample State-Identity Test

See Algorithm 1 + 5. The algorithm uses a novel state identity test which compares two distributions using a two-sample test [43] in Line 10 (recall that $\mathcal{G}_h = \mathcal{G}$ for $h \in [H]$ and $\mathcal{G}_{H+1} = \{x \mapsto 0\}$). Such an identity test mechanism is very different from the one used in the VALOR algorithm, and the two mechanisms have very different behavior. For example, if $\mathcal{G} = \{g^\star\}$, the local value algorithm will claim every state $s$ as "not new" because it knows the optimal value $V^\star(s)$, whereas the two-sample test may still declare a state $s$ to be new if $\mathbf{E}_s[g^\star(x)] \neq \mathbf{E}_{s'}[g^\star(x)]$ for any previously visited $s'$. On the other hand, the two-sample test algorithm may not have learned $V^\star(s)$ at all when it claims that a state $s$ is not new. Given the novelty of the mechanism, we believe analyzing the two-sample test algorithm and understanding its computational and statistical properties enriches our toolkit for dealing with the challenges addressed in this paper.

---

**Algorithm 5:** Algorithm with Two-Sample State-Identity Test

---

**1** **Function** polvalfun()

**2** &emsp; $\hat{g}_{H+1} \leftarrow 0$ ;

**3** &emsp; **for** $h = H : 1$ **do**

**4** &emsp;&emsp; $\hat{\pi}_h \leftarrow \text{argmax}_{\pi \in \Pi_h} \sum_{D \in \mathcal{D}_h^{\text{learned}}} \hat{\mathbf{E}}_D[K\mathbf{1}\{\pi(x_h) = a_h\}(r_h + \hat{g}_{h+1}(x_{h+1}))]$ ;

**5** &emsp;&emsp; $\hat{g}_h \leftarrow \text{argmin}_{g \in \mathcal{G}_h} \sum_{D \in \mathcal{D}_h^{\text{val}}} \hat{\mathbf{E}}_D[K\mathbf{1}\{\hat{\pi}_h(x_h) = a_h\}(g(x_h) - r_h - \hat{g}_{h+1}(x_{h+1}))^2]$ ;

**6** &emsp; $\hat{V}^\star \leftarrow \hat{\mathbf{E}}_D[\hat{g}_1(x_1)]$ where $D$ is the only distribution in $\mathcal{D}_1^{\text{val}}$;

**7** &emsp; **return** $\hat{\pi}_{1:H}, \hat{V}^\star$;

**8** **Function** dfslearn($a_{1:h-1}$)

**9** &emsp; $\tilde{D} \leftarrow$ sample $x_h \sim a_{1:h-1}, a_h \sim \text{Unif}(\mathcal{A}), r_h, x_{h+1}$ ;

**10** &emsp; $d_{\text{MMD}} \leftarrow \min_{D \in \mathcal{D}_h^{\text{val}}} \sup_{g \in \mathcal{G}_h} \left| \hat{\mathbf{E}}_D[g(x_h)] - \hat{\mathbf{E}}_{\tilde{D}}[g(x_h)] \right|$ ;

**11** &emsp; **if** $d_{MMD} \leq 2\tau$ **and** IS_RECURSIVE_CALL **then**

**12** &emsp;&emsp; **return**

**13** &emsp; **if** $d_{MMD} > 2\tau$ **then**

**14** &emsp;&emsp; Add $\tilde{D}$ to $\mathcal{D}_h^{\text{val}}$

**15** &emsp; Add $\tilde{D}$ to $\mathcal{D}_h^{\text{learned}}$;

**16** &emsp; **for** $a \in \mathcal{A}$ **do**

**17** &emsp;&emsp; dfslearn($a_{1:h-1} \circ a$) ;

---

### C.1.1 Computational considerations

The two-sample test algorithm requires three types nontrivial computation. Line 4 requires importance weighted policy optimization, which is simply a call to the CSC oracles. Line 5 performs squared-loss regression on $\mathcal{G}_h$, which is a call to a LS oracle.

The slightly unusual computation occurs on Line 10: we compute the (empirical) Maximum Mean Discrepancy (MMD) between $D$ and $\tilde{D}$ against the function class $\mathcal{G}_h$, and take the minimum over $D \in \mathcal{D}^{\text{val}}$. First, since $|\mathcal{D}_h^{\text{val}}|$ remains small over the execution of the algorithm, the minimization over $D \in \mathcal{D}_h^{\text{val}}$ can be done by enumeration. Then, for a fixed $D$, computing the MMD is a linear optimization problem over $\mathcal{G}_h$. In the special case where $\mathcal{G}_h$ is the unit ball in a Reproducing Kernel Hilbert Space (RKHS) [44], MMD can be computed in closed form by $O(n^2)$ kernel evaluations, where $n$ is the number of data points involved [43].

To unclutter the sample-complexity analysis, we assume that perfect oracles, i.e., $\epsilon_{\text{feas}} = \epsilon_{\text{sub}} = 0$.

### C.1.2 Sample complexity

**Theorem 26.** *Consider the same Markovian CDP setting as in Theorem 3 but we explicitly require here that the process is an MDP over $\mathcal{X}$. Under Assumption 9, for any $\epsilon, \delta \in (0, 1)$, the two-sample state-identity test algorithm (Algorithm 1+5) returns a policy $\pi$ such that $V^\star - V^\pi \leq \epsilon$ with probability at least $1 - \delta$, after collecting at most $\tilde{O}\left( \frac{M^2 K^2 H^6}{\epsilon^4} \log(|\mathcal{G}||\Pi|/\delta) \log^2(1/\delta) \right)$ trajectories.*

For this algorithm, we use the following notion of learned state:

**Definition 27** (Learned states). *Denote the sequence of states whose data sets are added to $\mathcal{D}_h^{\text{learned}}$ as $\mathcal{S}_h^{\text{learned}}$. States that are in $\mathcal{S}^{\text{learned}} = \bigcup_{h \in [H]} \mathcal{S}_h^{\text{learned}}$ are called learned. The sequence of states whose data sets are added to $\mathcal{D}_h^{\text{val}}$ are denoted by $\mathcal{S}_h^{\text{val}}$. Let $\mathcal{S}_h^{\text{check}}$ denote the set of all states that have been reached by any previous* dfslearn *call at level $h$.*

**Fact 28.** *We have $\mathcal{S}_h^{\text{val}} \subseteq \mathcal{S}_h^{\text{learned}} \subseteq \mathcal{S}_h^{\text{check}}$. Furthermore, $\forall s \in \mathcal{S}_h^{\text{learned}}$ and $a \in \mathcal{A}$, $s \circ a \in \mathcal{S}_{h+1}^{\text{check}}$.*

Define the following short-hand notations for the objective functions used in Algorithm 5:

$$V_D(\pi; g') := \hat{\mathbf{E}}_D[K\mathbf{1}\{\pi(x) = a\}(r + g'(x'))].$$

$$V_{\mathcal{D}_h^{\text{learned}}}(\pi; g') := \sum_{D \in \mathcal{D}_h^{\text{learned}}} V_D(\pi; g').$$

$$L_D(g; \pi, g') := \hat{\mathbf{E}}_D[K\mathbf{1}\{\pi(x) = a\}(g(x) - r - g'(x'))^2].$$

$$L_{\mathcal{D}_h^{\text{val}}}(g; \pi, g') := \sum_{D \in \mathcal{D}_h^{\text{val}}} L_D(g; \pi, g').$$

Also define $V_s$, $V_{\mathcal{S}_h^{\text{learned}}}$, $L_s$, $L_{\mathcal{S}_h^{\text{val}}}$ as the population version of $V_D$, $V_{\mathcal{D}_h^{\text{learned}}}$, $L_D$, $L_{\mathcal{D}_h^{\text{val}}}$, respectively.

**Concentration Results.** For our analysis we rely on the following concentration bounds that define the good event $\mathcal{E}$. This definition involves parameters $\tau, \tau_L, \tau_V$ whose values we will set later.

**Definition 29.** *Let $\mathcal{E}$ denote the event that for all $h \in [H]$ the total number of calls to* dfslearn$(p)$ *at level $h$ is at most $T_{\max} = M(K+1)(1 + Hn_{exp})$ during the execution of* MetaAlg *and that for all these calls to* dfslearn$(p)$ *the following deviation bounds hold for all $g \in \mathcal{G}_h, g' \in \mathcal{G}_{h+1}$ and $\pi \in \Pi_h$ (where $\tilde{D}$ is the data set of $n_{train}$ samples from Line 9 and $s$ is the state reached by $p$):*

$$|\hat{\mathbf{E}}_{\tilde{D}}[g(x)] - \mathbf{E}_s[g(x)]| \leq \tau \tag{12}$$

$$|V_{\tilde{D}}(\pi; g') - V_s(\pi; g')| \leq \tau_V \tag{13}$$

$$|L_{\tilde{D}}(g; \pi, g') - L_s(g; \pi, g')| \leq \tau_L. \tag{14}$$

We now show that this event has high probability.

**Lemma 30.** *Set $n_{train}$ so that*

$$n_{train} \geq \max\left\{ \frac{1}{2\tau^2} \ln\left(\frac{12HT_{\max}|\mathcal{G}|}{\delta}\right), \right.$$
$$\frac{16K}{\tau_V^2} \ln\left(\frac{12HT_{\max}|\mathcal{G}||\Pi|}{\delta}\right),$$
$$\left. \frac{32K}{\tau_L^2} \ln\left(\frac{12HT_{\max}|\mathcal{G}|^2|\Pi|}{\delta}\right) \right\}.$$

*Then $\mathbf{P}[\mathcal{E}] \geq 1 - \delta/2$ where $\mathcal{E}$ is defined in Definition 29. In addition, in event $\mathcal{E}$, during all calls the sequences are bounded as $|\mathcal{S}_h^{val}| \leq M$ and $|\mathcal{S}_h^{learned}| \leq T_{\max}$.*

*Proof.* Let us first focus on one call to dfslearn, say at path $p$ at level $h$. First, observe that the data set $\tilde{D}$ is a set of $n_{train}$ transitions sampled i.i.d. from the state $s$ that is reached by $p$. By Hoeffding's inequality and a union bound, with probability $1 - \delta'$, for all $g \in \mathcal{G}_h$

$$\left| \hat{\mathbf{E}}_{\tilde{D}}[g(x_h)] - \mathbf{E}_s[g(x_h)] \right| \leq \sqrt{\frac{\log(2|\mathcal{G}|/\delta')}{2n_{train}}}.$$

With $\delta' = \frac{\delta}{6HT_{\max}}$ the choice for $n_{train}$ let us bound the LHS by $\tau$.

For the random variable $K\mathbf{1}\{\pi(x_h) = a_h\}(r_h + g'(x_{h+1}))$, since $a_h$ is chosen uniformly at random, it is not hard to see that both the variance and the range are at most $2K$ (see for example Lemma 14 by Jiang et al. [1]). Applying Bernstein's inequality and a union bound, for all $\pi \in \Pi_h$ and $g \in \mathcal{G}_{h+1}$, we have

$$|V_{\tilde{D}}(\pi; g') - V_s(\pi; g')| \leq \sqrt{\frac{4K \log(2|\mathcal{G}||\Pi|/\delta')}{n_{train}}} + \frac{4K}{3n_{train}} \log(2|\mathcal{G}||\Pi|/\delta')$$

with probability $1 - \delta'$. As above, with $\delta' = \frac{\delta}{6HT_{\max}}$ our choice of $n_{train}$ ensures that this deviation is bound by $\tau_V$.

Similarly, we apply Bernstein's inequality to the random variable $K\mathbf{1}\{\pi(x_h) = a_h\}(g(x_h) - r_h - g'(x_{h+1}))^2$ which has range and variance at most $4K$. Combined with a union bound over all $g \in \mathcal{G}_h, g' \in \mathcal{G}_{h+1}, \pi \in \Pi_h$ we have that with probability $1 - \delta'$,

$$|L_{\tilde{D}}(g; \pi, g') - L_s(g; \pi, g')| \leq \sqrt{\frac{8K \log(2|\mathcal{G}|^2|\Pi|/\delta')}{n_{\text{train}}}} + \frac{2K}{n_{\text{train}}} \log(2|\mathcal{G}|^2|\Pi|/\delta') \leq \tau_L.$$

This last inequality is based on the choice for $n_{\text{train}}$ and $\delta' = \frac{\delta}{6HT_{\max}}$. For details on this concentration bound see for example Lemma 14 by Jiang et al. [1]. Using a union bound, the deviation bounds (12)–(14) hold for a single call to `dfslearn` with probability $1 - 3\delta'$.

Consider now the event $\mathcal{E}'$ that these bounds hold for the first $T_{\max}$ calls at each level $h$. Applying a union bound let us bound $\mathbf{P}(\mathcal{E}') \geq 1 - 3HT_{\max}\delta' = 1 - \frac{\delta}{2}$. It remains to show that $\mathcal{E}' \subseteq \mathcal{E}$.

First note that in event $\mathcal{E}'$ in the first $T_{\max}$ calls to `dfslearn` at level $h$, the algorithm does not call itself recursively during a recursive call if $p$ leads to a state $s \in \mathcal{S}_h^{\text{val}}$. To see this assume $p$ leads to a state $s \in \mathcal{S}_h^{\text{val}}$ and let $D \in \mathcal{D}_h^{\text{val}}$ be a data set sampled from this state. This means that $\tilde{D}$ and $D$ are sampled from the same distribution, and as such, we have for every $g \in \mathcal{G}_h$

$$|\hat{\mathbf{E}}_{\tilde{D}}[g(x)] - \hat{\mathbf{E}}_D[g(x)]| \leq |\hat{\mathbf{E}}_{\tilde{D}}[g(x)] - \mathbf{E}_s[g(x)]| + |\mathbf{E}_s[g(x)] - \hat{\mathbf{E}}_D[g(x)]| \leq 2\tau. \qquad (15)$$

Therefore $d_{MMD} \leq 2\tau$, the condition in the first clause is satisfied, and the algorithm does not recurse. If this condition is not satisfied, the algorithm adds $\tilde{D}$ to $\mathcal{D}_h^{\text{val}}$. Therefore, the initial call to `dfslearn` at the root can result in at most $MK$ recursive calls per level, since the identity tests must return true on identical states.

Further, for any fixed level, we issue at most $MHn_{\text{exp}}$ additional calls to `dfslearn`, since `MetaAlg` has at most $MH$ iterations and in each one, `dfslearn` is called $n_{\text{exp}}$ times per level. Any new state that we visit in this process was already counted by the $MK$ calls per level in the initial execution of `dfslearn`. On the other hand, these calls always descend to the children, so the number of calls to old states is at most $M(1 + K)Hn_{\text{exp}}$ per level. In total the number of calls to `dfslearn` per level is at most $M(1 + K)Hn_{\text{exp}} + MK \leq T_{\max}$, and $\mathbf{P}(\mathcal{E}) \leq \delta/2$ follows.

Further, the bound $|\mathcal{S}_h^{\text{learned}}| \leq T_{\max}$ follows from the fact that per call only one state can be added to $\mathcal{S}_h^{\text{learned}}$ and there are at most $T_{\max}$ calls. The bound $|\mathcal{S}_h^{\text{val}}| \leq M$ follows from the fact that in $\mathcal{E}$ no state can be added twice to $\mathcal{S}_h^{\text{val}}$ since as soon as it is in $\mathcal{S}_h^{\text{val}}$ once, $d_{MMD} \leq 2\tau$ holds (see Eq.(15)) and the current data set is not added to $\mathcal{D}_h^{\text{val}}$. $\qquad \square$

**Depth-first search and learning optimal values.** We now prove that `polvalfun` and `dfslearn` produce good value function estimates.

**Proposition 31.** *In event $\mathcal{E}$, consider an execution of `polvalfun` and let $\{\hat{g}_h, \hat{\pi}_h\}_{h \in [H]}$ denote the learned value functions and policies. Then every state $s$ in $\mathcal{S}_h^{\text{check}}$ satisfies*

$$|\mathbf{E}_s[\hat{g}_h(x_h)] - \mathbf{E}_s[g^\star(x_h)]| \leq (H + 1 - h)(2M\tau_V + \sqrt{4M^2\tau_V + 2T_{\max}\tau_L} + 8\tau), \qquad (16)$$

*and every learned state $s \in \mathcal{S}_h^{\text{learned}}$ satisfies*

$$V^\star(s) - Q^\star(s, \hat{\pi}_h) \leq 2M\tau_V + 2(H - h)(2M\tau_V + \sqrt{4M^2\tau_V + 2T_{\max}\tau_L} + 8\tau). \qquad (17)$$

*Proof.* We prove both inequalities simultaneously by induction over $h$. For convenience, we use the following short hand notations: $\epsilon_V = M\tau_V$ and $\epsilon_L = T_{\max}\tau_L$. Using this notation, in event $\mathcal{E}$, $|V_{\mathcal{D}_h^{\text{val}}}(\pi; g') - V_{\mathcal{S}_h^{\text{val}}}(\pi; g')| \leq \epsilon_V$ and $|L_{\mathcal{D}_h^{\text{learned}}}(g; \pi, g') - L_{\mathcal{S}_h^{\text{learned}}}(g; \pi, g')| \leq \epsilon_L$ hold for all $g, g'$ and $\pi$.

**Base case:** Both statement holds trivially for $h = H + 1$ since the LHS is 0 and the RHS is non-negative. In particular there are no actions, so Eq. (17) is trivial.

**Inductive case:** Assume that Eq. (16) holds on level $h + 1$. For any learned $s \in \mathcal{S}_h^{\text{learned}}$, we first show that $\hat{\pi}_h$ achieves high value compared to $\pi_{\hat{g}_{h+1}}^\star$ (recall its definition from Assumption 9) under $V_s(\cdot; \hat{g}_{h+1})$:

$$V_s(\pi_{\hat{g}_{h+1}}^\star; \hat{g}_{h+1}) - V_s(\hat{\pi}_h; \hat{g}_{h+1}) \leq \sum_{s \in \mathcal{S}_h^{\text{learned}}} V_s(\pi_{\hat{g}_{h+1}}^\star; \hat{g}_{h+1}) - V_s(\hat{\pi}_h; \hat{g}_{h+1})$$

$$(\pi_{\hat{g}_{h+1}}^\star \text{ is optimal w.r.t. } \hat{g}_{h+1} \text{ in all } s)$$

$$= V_{\mathcal{S}_h^{\text{learned}}}(\pi_{\hat{g}_{h+1}}^\star; \hat{g}_{h+1}) - V_{\mathcal{S}_h^{\text{learned}}}(\hat{\pi}_h; \hat{g}_{h+1})$$

$$\leq V_{\mathcal{D}_h^{\text{learned}}}(\pi_{\hat{g}_{h+1}}^\star; \hat{g}_{h+1}) - V_{\mathcal{D}_h^{\text{learned}}}(\hat{\pi}_h; \hat{g}_{h+1}) + 2\epsilon_V \leq 2\epsilon_V. \tag{18}$$

Eq. (17) follows as a corollary:

$$V^\star(s) - Q^\star(s, \hat{\pi}_h)$$
$$= V_s(\pi^\star; g^\star) - V_s(\hat{\pi}_h; g^\star) \qquad \text{(definition of } V_s\text{)}$$
$$\leq V_s(\pi^\star; \hat{g}_{h+1}) - V_s(\hat{\pi}_h; \hat{g}_{h+1}) + 2 \sup_{\substack{s' \text{ being child of } s}} |\mathbf{E}_{x' \sim s'}[\hat{g}_{h+1}(x_{h+1}) - g^\star(x_{h+1})]|$$

$$\leq V_s(\pi_{\hat{g}_{h+1}}^\star; \hat{g}_{h+1}) - V_s(\hat{\pi}_h; \hat{g}_{h+1}) + 2(H - h)(2\epsilon_V + \sqrt{4M\epsilon_V + 2\epsilon_L} + 8\tau)$$
$$(s \in \mathcal{S}_h^{\text{val}} \Rightarrow s' \in \mathcal{S}_{h+1}^{\text{check}} \text{ and induction})$$

$$\leq 2\epsilon_V + 2(H - h)(2\epsilon_V + \sqrt{4M\epsilon_V + 2\epsilon_L} + 8\tau). \qquad \text{(using Eq.(18))}$$

This proves Eq. (17) at level $h$. The rest of the proof proves Eq.(16). First we introduce and recall the definitions:

$$g_{\hat{\pi}_h, \hat{g}_{h+1}}(x) = \mathbf{E}[r + \hat{g}_{h+1}(x_{h+1}) \mid x_h = x, a_h = \hat{\pi}_h(x)],$$
$$g_{\star, \hat{g}_{h+1}}(x) = \mathbf{E}[r + \hat{g}_{h+1}(x_{h+1}) \mid x_h = x, a_h = \pi_{\hat{g}_{h+1}}^\star(x)].$$

Note that $g_{\hat{\pi}_h, \hat{g}_{h+1}} \notin \mathcal{G}_h$ in general, but it is the Bayes optimal predictor for the squared losses $L_s(\cdot; \hat{\pi}_h, \hat{g}_{h+1})$ for all $s$ simultaneously. On the other hand, Assumption 9 guarantees that $g_{\star, \hat{g}_{h+1}} \in \mathcal{G}_h$, for any $\hat{g}_{h+1}$.

The LHS of Eq.(16) can be bounded as

$$|\mathbf{E}_s[g^\star(x_h)] - \mathbf{E}_s[\hat{g}_h(x_h)]| \leq |\mathbf{E}_s[g^\star(x_h)] - \mathbf{E}_s[g_{\hat{\pi}_h, \hat{g}_{h+1}}(x_h)]| + |\mathbf{E}_s[g_{\hat{\pi}_h, \hat{g}_{h+1}}(x_h)] - \mathbf{E}_s[\hat{g}_h(x_h)]|. \tag{19}$$

To bound the first term in Eq.(19),

$$|\mathbf{E}_s[g^\star(x_h)] - \mathbf{E}_s[g_{\hat{\pi}_h, \hat{g}_{h+1}}(x_h)]| \leq |\mathbf{E}_s[g^\star(x_h)] - \mathbf{E}_s[g_{\star, \hat{g}_{h+1}}(x_h)]| + |\mathbf{E}_s[g_{\star, \hat{g}_{h+1}}(x_h)] - \mathbf{E}_s[g_{\hat{\pi}_h, \hat{g}_{h+1}}(x_h)]|$$

$$= |\mathbf{E}_s[g^\star(x_h) - g_{\star, \hat{g}_{h+1}}(x_h)]| + |V_s(\pi_{\hat{g}_{h+1}}^\star; \hat{g}_{h+1}) - V_s(\hat{\pi}_h; \hat{g}_{h+1})|$$

$$\leq |\mathbf{E}_s[g^\star(x_h) - g_{\star, \hat{g}_{h+1}}(x_h)]| + 2\epsilon_V. \qquad \text{(using Eq.(18))}$$

Now consider each individual context $x_h$ emitted in $s \in \mathcal{S}_h$:

$$g^\star(x_h) - g_{\star, \hat{g}_{h+1}}(x_h)$$
$$= \mathbf{E}_{r_h \sim R(x_h, \pi^\star(x_h))}[r_h] + \mathbf{E}_{s \circ \pi^\star(x_h)}[g^\star(x_{h+1})] - \max_{a \in \mathcal{A}} \left( \mathbf{E}_{r_h \sim R(x_h, a)}[r_h] + \mathbf{E}_{s \circ a}[\hat{g}_{h+1}(x_{h+1})] \right)$$

$$\leq \mathbf{E}_{r_h \sim R(x_h, \pi^\star(x_h))}[r_h] + \mathbf{E}_{s \circ \pi^\star(x_h)}[\hat{g}_{h+1}(x_h)]$$
$$- \max_{a \in \mathcal{A}} \left( \mathbf{E}_{r_h \sim R(x_h, a)}[r] + \mathbf{E}_{s \circ a}[\hat{g}_{h+1}(x_h)] \right) + |\mathbf{E}_{s \circ \pi^\star(x_h)}[\hat{g}_{h+1}(x_{h+1}) - g^\star(x_{h+1})]|$$

$$\leq |\mathbf{E}_{s \circ \pi^\star(x_h)}[\hat{g}_{h+1}(x_{h+1}) - g^\star(x_{h+1})]|$$
$$\leq (H - h)(2\epsilon_V + \sqrt{4M\epsilon_V + 2\epsilon_L} + 8\tau).$$

The second inequality is true since the second term optimizes over $a \in \mathcal{A}$ and the first term is the special case of $a = \pi^\star(x_h)$. The last inequality follows from the fact that if $s \in \mathcal{S}_h^{\text{learned}} \Rightarrow s \circ a \in \mathcal{S}_{h+1}^{\text{check}}$ and we can therefore apply the induction hypothesis. We can use the same argument to lower bound the above quantity. This gives

$$|\mathbf{E}_s[g^\star(x_h)] - \mathbf{E}_s[g_{\hat{\pi}_h, \hat{g}_{h+1}}(x_h)]| \leq (H - h)(2\epsilon_V + \sqrt{4M\epsilon_V + 2\epsilon_L} + 8\tau) + 2\epsilon_V.$$

Next, we work with the second term in Equation (19):

$$\left|\mathbf{E}_s[\hat{g}_h(x_h)] - \mathbf{E}_s[g_{\hat{\pi}_h,\hat{g}_{h+1}}(x_h)]\right|$$

$$\leq \sqrt{\mathbf{E}_s\left[\left(\hat{g}_h(x_h) - g_{\hat{\pi}_h,\hat{g}_{h+1}}(x_h)\right)^2\right]} \qquad \text{(Jensen's inequality)}$$

$$= \sqrt{L_s(\hat{g}_h;\hat{\pi}_h,\hat{g}_{h+1}) - L_s(g_{\hat{\pi}_h,\hat{g}_{h+1}};\hat{\pi}_h,\hat{g}_{h+1})} \quad (g_{\hat{\pi}_h,\hat{g}_{h+1}} \text{ is Bayes-optimal for } L_s(\cdot;\hat{\pi}_h,\hat{g}_{h+1}))$$

$$\leq \sqrt{L_{\mathcal{S}_h^{\text{val}}}(\hat{g}_h;\hat{\pi}_h,\hat{g}_{h+1}) - L_{\mathcal{S}_h^{\text{val}}}(g_{\hat{\pi}_h,\hat{g}_{h+1}};\hat{\pi}_h,\hat{g}_{h+1})} \qquad (\dots \text{for all } s)$$

$$\leq \sqrt{L_{\mathcal{D}_h^{\text{val}}}(\hat{g}_h;\hat{\pi}_h,\hat{g}_{h+1}) - L_{\mathcal{S}_h^{\text{val}}}(g_{\hat{\pi}_h,\hat{g}_{h+1}};\hat{\pi}_h,\hat{g}_{h+1}) + \epsilon_L}$$

$$\leq \sqrt{L_{\mathcal{D}_h^{\text{val}}}(g_{\star,\hat{g}_{h+1}};\hat{\pi}_h,\hat{g}_{h+1}) - L_{\mathcal{S}_h^{\text{val}}}(g_{\hat{\pi}_h,\hat{g}_{h+1}};\hat{\pi}_h,\hat{g}_{h+1}) + \epsilon_L}$$
$$\qquad (\hat{g}_h \text{ minimizes the first term over } \mathcal{G}_h, \text{ and } g_{\star,\hat{g}_{h+1}} \in \mathcal{G}_h \text{ from Assumption } 9)$$

$$\leq \sqrt{L_{\mathcal{S}_h^{\text{val}}}(g_{\star,\hat{g}_{h+1}};\hat{\pi}_h,\hat{g}_{h+1}) - L_{\mathcal{S}_h^{\text{val}}}(g_{\hat{\pi}_h,\hat{g}_{h+1}};\hat{\pi}_h,\hat{g}_{h+1}) + 2\epsilon_L}$$

$$= \sqrt{\sum_{s \in \mathcal{S}_h^{\text{val}}} \mathbf{E}_{x\sim s}[(g_{\star,\hat{g}_{h+1}}(x) - g_{\hat{\pi}_h,\hat{g}_{h+1}}(x))^2] + 2\epsilon_L}$$

$$\leq \sqrt{\sum_{s \in \mathcal{S}_h^{\text{val}}} \mathbf{E}_{x\sim s}[2|g_{\star,\hat{g}_{h+1}}(x) - g_{\hat{\pi}_h,\hat{g}_{h+1}}(x)|] + 2\epsilon_L} \qquad \text{(range of variables)}$$

$$= \sqrt{\sum_{s \in \mathcal{S}_h^{\text{val}}} 2\mathbf{E}_{x\sim s}[g_{\star,\hat{g}_{h+1}}(x) - g_{\hat{\pi}_h,\hat{g}_{h+1}}(x)] + 2\epsilon_L} \qquad (g_{\star,\hat{g}_{h+1}}(x) \geq g_{\hat{\pi}_h,\hat{g}_{h+1}}(x) \;\; \forall x)$$

$$= \sqrt{\sum_{s \in \mathcal{S}_h^{\text{val}}} 2(V_s(\pi_{\hat{g}_{h+1}}^{\star};\hat{g}_{h+1}) - V_s(\hat{\pi}_h;\hat{g}_{h+1})) + 2\epsilon_L}$$

$$\leq \sqrt{4M\epsilon_V + 2\epsilon_L}. \qquad (|\mathcal{S}_h^{\text{val}}| \leq M \text{ and Eq.}(18))$$

Put together, we get the desired result for states $s \in \mathcal{S}_h^{\text{val}}$:

$$|\mathbf{E}_s[g^\star(x_h)] - \mathbf{E}_s[\hat{g}_h(x_h)]| \leq (H-h)(2\epsilon_V + \sqrt{4M\epsilon_V + 2\epsilon_L} + 8\tau) + 2\epsilon_V + \sqrt{4M\epsilon_V + 2\epsilon_L}. \tag{20}$$

It remains to deal with states $s \in \mathcal{S}_h^{\text{check}} \setminus \mathcal{S}_h^{\text{val}}$. According to the algorithm, this only happens when the MMD test suggests that the data set $\tilde{D}$ drawn from $s$ looks very similar to a previous data set $D \in \mathcal{D}_h^{\text{val}}$, which corresponds to some $s' \in \mathcal{S}_h^{\text{val}}$. So,

$$|\mathbf{E}_s[\hat{g}_h(x_h)] - \mathbf{E}_s[g^\star(x_h)]|$$
$$\leq |\mathbf{E}_{s'}[\hat{g}_h(x_h)] - \mathbf{E}_{s'}[g^\star(x_h)]| + |\mathbf{E}_{s'}[\hat{g}_h(x_h)] - \mathbf{E}_s[\hat{g}_h(x_h)]| + |\mathbf{E}_s[g^\star(x_h)] - \mathbf{E}_{s'}[g^\star(x_h)]|$$
$$\leq (H-h)(2\epsilon_V + \sqrt{4M\epsilon_V + 2\epsilon_L} + 8\tau) + 2\epsilon_V + \sqrt{4M\epsilon_V + 2\epsilon_L} \qquad (s' \in \mathcal{S}_h^{\text{val}} \text{ \& Eq.}(20))$$
$$\quad + |\hat{\mathbf{E}}_D[\hat{g}_h(x_h)] - \hat{\mathbf{E}}_{\tilde{D}}[\hat{g}_h(x_h)]| + |\hat{\mathbf{E}}_D[g^\star(x_h)] - \mathbf{E}_{\tilde{D}}[g^\star(x_h)]| + 4\tau$$
$$\leq (H-h)(2\epsilon_V + \sqrt{4M\epsilon_V + 2\epsilon_L} + 8\tau) + 2\epsilon_V + \sqrt{4M\epsilon_V + 2\epsilon_L} + 2\tau + 2\tau + 4\tau$$
$$\qquad \text{(MMD test fires)}$$
$$= (H+1-h)(2\epsilon_V + \sqrt{4M\epsilon_V + 2\epsilon_L} + 8\tau). \qquad \square$$

**Quality of Learned Policies and Meta-Algorithm Analysis.** After quantifying the estimation error of the value function returned by `polvalfun`, it remains to translate that into a bound on the suboptimality of the returned policy:

**Proposition 32.** *Assume we are in event $\mathcal{E}$. Then the policy $\hat{\pi} = \hat{\pi}_{1:H}$ returned by* `polvalfun` *in Algorithm 5 satisfies*

$$V^{\hat{\pi}} \geq V^\star - p_{ul}^{\hat{\pi}} - 2HM\tau_V - H^2(2M\tau_V + \sqrt{4M^2\tau_V + 2T_{\max}\tau_L} + 8\tau)$$

*where $p_{ul}^{\hat{\pi}} = \mathbf{P}(\exists h \in [H] : s_h \notin \mathcal{S}^{learned} \mid a_{1:H} \sim \hat{\pi})$ is the probability of hitting an unlearned state when following $\hat{\pi}$.*

*Proof.* Proposition 31 states that for every learned state $s \in \mathcal{S}_h^{\text{learned}}$

$$V^\star(s) - Q^\star(s, \hat{\pi}_h) \leq 2M\tau_V + 2(H-h)(2M\tau_V + \sqrt{4M^2\tau_V + 2T_{\max}\tau_L} + 8\tau). \quad (21)$$

Using Lemma 18, we can show that $\hat{\pi}$ yields expected return that is optimal up to

$$V^\star - V^{\hat{\pi}} \leq p_{ul}^{\hat{\pi}} + (1 - p_{ul}^{\hat{\pi}}) \sum_{h=1}^{H} (2M\tau_V + 2(H-h)(2M\tau_V + \sqrt{4M^2\tau_V + 2T_{\max}\tau_L} + 8\tau))$$

$$\leq p_{ul}^{\hat{\pi}} + 2HM\tau_V + 2(2M\tau_V + \sqrt{4M^2\tau_V + 2T_{\max}\tau_L} + 8\tau) \sum_{h=1}^{H} (H-h)$$

$$\leq p_{ul}^{\hat{\pi}} + 2HM\tau_V + H^2(2M\tau_V + \sqrt{4M^2\tau_V + 2T_{\max}\tau_L} + 8\tau). \qquad \square$$

**Lemma 33.** *Consider running* MetaAlg *with* dfslearn *and* polvalfun *(Algorithm 1 + 5) with parameters*

$$n_{exp} \geq \frac{8}{\epsilon} \ln\left(\frac{4MH}{\delta}\right), \qquad n_{eval} \geq \frac{32}{\epsilon^2} \ln\left(\frac{8MH}{\delta}\right),$$

$$\tau = \frac{\epsilon}{2^6 3 H^2}, \quad \tau_V = \frac{\epsilon^2}{2^8 3^4 M^2 H^4}, \quad \tau_L = \frac{\epsilon^2}{2^7 3^2 H^4 T_{\max}}$$

*Then with probability at least $1 - \delta$,* MetaAlg *returns a policy that is at least $\epsilon$-optimal after at most $MK$ iterations.*

*Proof.* The proof is completely analogous to the proof of Lemma 20 except with using Proposition 32 instead of Proposition 19. We set the parameters $\tau$, $\tau_L$ and $\tau_V$ so that the policy guarantee of Proposition 32 is $V^{\hat{\pi}} \geq V^\star - p_{ul}^{\hat{\pi}} - \epsilon/8$. More specifically, we bound the guaranteed gap as

$$2HM\tau_V + H^2(2M\tau_V + \sqrt{4M^2\tau_V + 2T_{\max}\tau_L} + 8\tau)$$

$$\leq 2MH\tau_V + 2MH^2\tau_V + 2MH^2\sqrt{\tau_V} + H^2\sqrt{2T_{\max}\tau_L} + 8H^2\tau$$

$$\leq 6MH^2\sqrt{\tau_V} + H^2\sqrt{2T_{\max}\tau_L} + 8H^2\tau$$

and then set $\tau$, $\tau_L$ and $\tau_V$ so that each terms evaluates to $\epsilon/24$. $\qquad \square$

**Proof of Theorem 26.** We now have all parts to complete the proof of Theorem 26.

*Proof.* For the calculation, we instantiate all the parameters as

$$n_{\exp} = \frac{8}{\epsilon} \ln\left(\frac{4MH}{\delta}\right), \quad n_{\text{eval}} = \frac{32}{\epsilon^2} \ln\left(\frac{8MH}{\delta}\right), \quad n_{\text{train}} = 16K\left(\frac{2}{\tau_L^2} + \frac{1}{\tau_V^2}\right) \ln\left(\frac{12HT_{\max}|\mathcal{G}|^2|\Pi|}{\delta}\right).$$

$$\tau = \frac{\epsilon}{2^6 3 H^2}, \quad \tau_V = \frac{\epsilon^2}{2^8 3^4 M^2 H^4}, \quad \tau_L = \frac{\epsilon^2}{2^7 3^2 H^4 T_{\max}} \quad T_{\max} = M(K+1)(1 + Hn_{\exp}).$$

These settings suffice to apply all of the above lemmas for these algorithms and therefore with these settings the algorithm outputs a policy that is at most $\epsilon$-suboptimal, except with probability $\delta$. For the sample complexity, since $T_{\max}$ is an upper bound on the number of datasets we collect (because $T_{\max}$ is an upper bound on the number of execution of dfslearn at any level), and we also $n_{\text{eval}}$ trajectories for each of the $MH$ iterations of MetaAlg, the total sample complexity is

$$HT_{\max}n_{\text{train}} + MHn_{\text{eval}} = \tilde{O}\left(\frac{M^5 H^{12} K^4}{\epsilon^7} \log(|\mathcal{G}||\Pi|/\delta) \log^3(1/\delta)\right). \qquad \square$$

### C.2 Global Policy Algorithm

See Algorithm 6. As the other algorithms, this method learns states using depth-first search. The state identity test is similar to that of VALOR at a high level: for any new path $p$, we derive an upper bound and a lower bound on $V^\star(p)$, and prune the path if the gap is small. Unlike in VALOR where both bounds are derived using the value function class $\mathcal{G}$, here only the upper bound is from a value

function (see Line 11), and the lower bound comes from *Monte-Carlo roll-out* with a near-optimal policy, which avoids the need for on-demand exploration.

More specifically, the global policy algorithm does not store data sets but maintains a global policy, a set of learned paths, and a set of pruned paths, all of which are updated over time. We always guarantee that $\hat{\pi}_{h:H}$ is near-optimal for any learned state at level $h$, and leverage this property to conduct state-identity test: if a new path $p$ leads to the same state as a learned path $q$, then Eq.(22) yields a tight upper bound on $V^\star(p)$, which can be achieved by $\hat{\pi}_{h:H}$ up to some small error and we check by Monte-Carlo roll-outs. If the test succeeds, the path $p$ is added to the set PRUNED$(h)$. Otherwise, all successor states are learned (or pruned) in a recursive manner, after which the state itself becomes learned (i.e., $p$ added to LEARNED$(h)$). Then, the policy at level $h$ is updated to be near-optimal for the newly learned state in addition to the previous ones (Line 25). Once we change the global policy, however, all the pruned states need to be re-checked (Line 26), as their optimal values are only guaranteed to be realized by the previous global policy and not necessarily by the new policy.

### C.2.1 Computational efficiency

The algorithm contains three non-trivial computational components. In Eq.(22), a linear program is solved to determine the optimal value estimate of the current path given the value of one learned state (LP oracle). In Line 24, computing the value of each learned path can be reduced to multi-class cost-sensitive classification as in the other two algorithms (CSC oracle). Finally, fitting the global policy in Line (25) requires the same problem as the policy fitting procedure discussed in Section B.7 (multi data set classification oracle).

As with the previous algorithm, we assume no error in the oracles ($\epsilon_{\text{feas}} = \epsilon_{\text{sub}} = 0$) in the following to simplify the analysis.

### C.2.2 Sample complexity

**Theorem 34.** *Consider a Markovian contextual decision process with deterministic dynamics over $M$ hidden states, as described in Section 3. When Assumption 10 and 8 hold, for any $\epsilon, \delta \in (0, 1)$, the global policy algorithm (Algorithm 6) returns a policy $\pi$ such that $V^\star - V^\pi \leq \epsilon$ with probability at least $1 - \delta$, after collecting at most $\tilde{O}\left(\frac{M^3 H^3 K}{\epsilon^2} \log\left(|\Pi||\mathcal{G}|/\delta\right)\right)$ trajectories.*

In the following, we prove this statement but first introduce helpful notation:

**Definition 35** (Deviation Bounds). *We say the deviation bound holds for a data set of $n_{train}$ observations sampled from $q$ in Line 23 during a call to* dfslearn *if for all $\pi \in \Pi_h$*

$$|\hat{\mathbf{E}}_{D_q}[K\mathbf{1}\{a = \pi(x)\}\bar{r}] - \mathbf{E}_{q,\hat{\pi}_{h+1:H}}[K\mathbf{1}\{a_h = \pi(x_h)\}\bar{r}]| \leq \tau_{pol},$$

*where we use $\mathbf{E}_{q,\hat{\pi}_{h+1:H}}[\cdot]$ as shorthand for $\mathbf{E}[\cdot|s_h = s, a_h \sim Uniform(K), a_{h+1:H} \sim \hat{\pi}_{h+1:H}]$ with $s$ being the state reached by $p$ and $\hat{\pi}_{h+1:H}$ being the current policy when the data set was collected. We say the deviation bound holds for a data set of $n_{test}$ observations sampled in Line 8 during a call to* TestLearned *if for all $g \in \mathcal{G}_h$:*

$$|\hat{\mathbf{E}}_D[g(x_h)] - \mathbf{E}_p[g(x_h)]| \leq \tau_{val}, \qquad |\hat{\mathbf{E}}_D[\bar{r}] - V^{\hat{\pi}_{h+1:H}}(p)| \leq \tau_{val}.$$

*We say the deviation bound holds for a data set of $n_{test}$ observations sampled in Line 10 during a call to* TestLearned *if for all $g \in \mathcal{G}_h$:*

$$|\hat{\mathbf{E}}_{D_q'}[g(x_h)] - \mathbf{E}_q[g(x_h)]| \leq \tau_{val}, \qquad |\hat{\mathbf{E}}_{D_q'}[\bar{r}] - V^{\hat{\pi}_{h+1:H}}(q)| \leq \tau_{val}.$$

**Learning Values using Depth First Search.** We first show that if the current policy is close to optimal for all learned states, then the policy is also good on all states for which TestLearned returns true.

**Lemma 36** (Policy on Tested States). *Consider a call of* TestLearned *at path $p$ and level $h$ and assume the deviation bounds of Definition 35 hold for all data sets collected during this and all prior calls. Assume further that $\hat{\pi}_{h:H}$ satisfies $V^{\hat{\pi}_{h:H}}(q) \geq V^\star(q) - \phi_h$ for all $q \in$ LEARNED$(h)$. Then $g^\star$ is always feasible for the program in Equation (22) and if* TestLearned *returns true, then the current policy $\hat{\pi}_{h:H}$ is near optimal for $p$, that is $V^{\hat{\pi}_{h:H}}(p) \geq V^\star(p) - \phi_h - 8\tau_{val}.$*

**Algorithm 6:** Global Policy

---

**1 Function** main
**2**   Global LEARNED$(h)$, $h \in [H]$;
**3**   Global PRUNED$(h)$, $h \in [H]$;
**4**   Global $\{\hat{\pi}_h\}_{h \in [H]}$;
**5**   dfslearn $(\circ)$;
**6**   **return** $\{\hat{\pi}_h\}_{h \in [H]}$;

**7 Function** TestLearned $(p, h)$
**8**   Collect dataset $D = \{(x_h, \bar{r})\}$ of size $n_{\text{test}}$ where $x_h \sim p$, $a_{h:H} \sim \hat{\pi}_{h:H}$, $\bar{r} = \sum_{h'=h}^{H} r_{h'}$ ;
**9**   **for** $q \in$ LEARNED$(h)$ **do**
**10**      Collect dataset $D'_q = \{(x_h, \bar{r})\}$ of size $n_{\text{test}}$ where $x_h \sim q$, $a_{h:H} \sim \hat{\pi}_{h:H}$,
          $\bar{r} = \sum_{h'=h}^{H} r_{h'}$ ;
**11**      Solve

$$V_{opt} = \max_{g \in \mathcal{G}} \mathbf{E}_D[g(x_h)] \text{ s.t. } \mathbf{E}_{D'_q}[g(x_h) - \bar{r}] \le \phi_h + 2\tau_{val} \qquad (22)$$

         **if** $V_{opt} \le \mathbf{E}_{D'_q}[\bar{r}] + \phi_h + 4\tau_{val}$ *and* $\mathbf{E}_D[\bar{r}] \ge \mathbf{E}_{D'_q}[\bar{r}] - 2\tau_{val}$ **then**
**12**         **return** *true*;

**13**   **return** *false*;

**14 Function** dfslearn $(p)$
**15**   Let $h = |p| - 1$ the current level;
**16**   **if** *not_called_from_Line_28* *and* TestLearned $(p, h)$ **then**
**17**      Add $p$ to PRUNED$(h)$;
**18**      **return** ;
**19**   **for** $a \in \mathcal{A}$ **do**
**20**      dfslearn $(p \circ a)$ ;
**21**   Add $p$ to LEARNED$(h)$;
**22**   **for** $q \in$ LEARNED$(h)$ **do**
**23**      Collect dataset $D_q = \{(x_h, a_h, \bar{r})\}$ of size $n_{\text{train}}$ where $x_h \sim q$, $a_h \sim$ Unif,
          $a_{h+1:H} \sim \hat{\pi}_{h+1:H}$, $\bar{r} = \sum_{h'=h}^{H} r_{h'}$ ;
**24**      $\hat{V}(q) \leftarrow \max_{\pi \in \Pi} \mathbf{E}_{D_q}[K\mathbf{1}\{a_h = \pi(x_h)\}\bar{r}]$ ;
**25**   Update $\hat{\pi}_h$ to be any policy satisfying

$$\forall q \in \text{LEARNED}(h) \quad \hat{\mathbf{E}}_{D_q}[K\mathbf{1}\{a_h = \pi(x_h)\}\bar{r}] \ge \hat{V}(q) - 2\tau_{pol}$$

      **for** $q \in$ PRUNED$(h)$ **do**
**26**      **if** TestLearned$(q, h) = false$ **then**
**27**         remove $q$ from PRUNED$(h)$;
**28**         dfslearn $(q)$ ;

**29**   **return** ;

---

*Proof.* The optimal value function $g^\star$ is always feasible since

$$\hat{\mathbf{E}}_{D'}[g^\star(x) - \bar{r}] \leq V^\star(q) - V^{\hat{\pi}_{h:H}}(q) + 2\tau_{val} \leq \phi_h + 2\tau_{val}.$$

Here, we first used the deviation bounds and then the assumption about the performance of the current policy on learned states. Therefore, $V_{opt} \geq \hat{\mathbf{E}}_D[g^\star(x)] \geq V^\star(p) - \tau_{val}$ cannot underestimate the optimal value of $p$ by much. Consider finally the performance of the current policy on $p$ if `TestLearned` returns true:

$$V^{\hat{\pi}_{h:H}}(p) \geq \hat{\mathbf{E}}_D[\bar{r}] - \tau_{val} \geq \hat{\mathbf{E}}_{D'}[\bar{r}] - 3\tau_{val}$$
$$\geq V_{opt} - 3\tau_{val} - 4\tau_{val} - \phi_h \geq V^\star(p) - 8\tau_{val} - \phi_h.$$

Here, the first inequality follows from the deviation bounds, the second from the second condition of the if-clause in `TestLearned`, the third from the first condition of the if-clause and finally the fact that $V_{opt}$ is an accurate estimate of the optimal value of $p$. □

Thus, the `TestLearned` routine can identify paths where the current policy is close to optimal if this policy's performance on all learned states is good. Next, we prove that the policy has near-optimal performance on all the learned states.

**Lemma 37** (Global policy fitting). *Consider a call of* `dfslearn` *(p) at level $h$ and assume the deviation bounds hold for all data sets collected during this and all prior calls. Then the program in Line 25 is always feasible and after executing that line, we have $\forall q \in \text{LEARNED}(h)$,*

$$Q^{\hat{\pi}_{h+1:H}}(q, \hat{\pi}_h) \geq Q^{\hat{\pi}_{h+1:H}}(q, \star) - 3\tau_{pol}, \tag{23}$$

*where $\star$ is a shorthand for $\pi^\star_{\hat{\pi}_{h+1:H}}$, the policy defined in Assumption 10 w.r.t. the current policy $\hat{\pi}_{h+1:H}$. This implies that if all children nodes $q'$ of $q$ satisfy $V^{\hat{\pi}_{h+1:H}}(q') \geq V^\star(q') - \beta$ for some $\beta$, then $V^{\hat{\pi}_{h:H}}(q) \geq V^\star(q) - \beta - 3\tau_{pol}$.*

*Proof.* We prove feasibility by showing that $\pi^\star_{\hat{\pi}_{h+1:H}}$ is always feasible. For each $q \in \text{LEARNED}(h)$, let $\hat{\pi}^q_h$ denote the policy that achieves the maximum in computing $\hat{V}(q)$. Then

$$\hat{\mathbf{E}}_{D_q}[K\mathbf{1}\{a_h = \pi^\star_{\hat{\pi}_{h+1:H}}(x_h)\}\bar{r}] \geq Q^{\hat{\pi}_{h+1:H}}(q, \star) - \tau_{pol} \geq Q^{\hat{\pi}_{h+1:H}}(q, \hat{\pi}^q_h) - \tau_{pol} \geq \hat{V}(q) - 2\tau_{pol}.$$

The first and last inequality are due to the deviation bounds and the second inequality follows from definition of $\pi^\star_{\hat{\pi}_{h+1:H}}$. This proves the feasibility. Now, using this inequality along with $\hat{V}(q) = \max_{\pi \in \Pi} \mathbf{E}_{D_q}[K\mathbf{1}\{a_h = \pi(x_h)\}\bar{r}]$, we can relate $\hat{V}(q)$ and $Q^{\hat{\pi}_{h+1:H}}(q, \star)$:

$$\hat{V}(q) \geq \hat{\mathbf{E}}_{D_q}[K\mathbf{1}\{a_h = \pi^\star_{\hat{\pi}_{h+1:H}}(x_h)\}\bar{r}] \geq Q^{\hat{\pi}_{h+1:H}}(q, \star) - \tau_{pol}.$$

Finally, since $\hat{\pi}_h$ is feasible in Line 25,

$$V^{\hat{\pi}_{h:H}}(q) = Q^{\hat{\pi}_{h+1:H}}(q, \hat{\pi}_h) \geq \hat{V}(q) - 2\tau_{pol} \geq Q^{\hat{\pi}_{h+1:H}}(q, \star) - 3\tau_{pol}.$$

To prove the implication, consider the case where for $a \in \mathcal{A}$, all paths $q' = q \circ a$ satisfy $V^{\hat{\pi}_{h+1:H}}(q') \geq V^\star(q') - \beta$. Then

$$V^\star(q) - V^{\hat{\pi}_{h:H}}(q) \leq V^\star(q) - Q^{\hat{\pi}_{h+1:H}}(q, \star) + 3\tau_{pol} \leq V^\star(q) - Q^{\hat{\pi}_{h+1:H}}(q, \pi^\star) + 3\tau_{pol}$$
$$= \mathbf{E}_{q' \sim q \circ \pi^\star}[V^\star(q') - V^{\hat{\pi}_{h+1:H}}(q')] + 3\tau_{pol} \leq \beta + 3\tau_{pol},$$

where we first used the inequality from above and then the fact that $\pi^\star_{\hat{\pi}_{h+1:H}}$ is optimal given the fixed policy $\hat{\pi}_{h+1:H}$. The equality holds since both $V^\star(q) - Q^{\hat{\pi}_{h+1:H}}(q, \pi^\star)$ both are with respect to $a_h \sim \pi^\star_h$ and finally we apply the assumption. □

We are now ready to apply both lemmas above recursively to control the performance of the current policy on all learned and pruned paths:

**Lemma 38.** *Set $\phi_h = (H - h + 1)(8\tau_{val} + 3\tau_{pol})$ and consider a call to* `dfslearn`*(p) at level $h$. Assume the deviation bounds hold for all data sets collected until this call terminates. Then for all $p \in \text{LEARNED}(h)$, the current policy satisfies*

$$V^{\hat{\pi}_{h:H}}(p) \geq V^\star(p) - \phi_h$$

*at all times except between adding a new path and updating the policy. Further, for all $p \in$* PRUNED$(h)$ *the currently policy satisfies*

$$V^{\hat{\pi}_{h:H}}(p) \geq V^{\star}(p) - \phi_h - 8\tau_{val}$$

*whenever* dfslearn *returns from level $h$ to $h-1$.*

*Proof.* We prove the claim inductively. For $h = H + 1$ the statement is trivially true since there are no actions left to take and therefore the value of all policies is identical 0 by definition.

Assume now the statement holds for $h + 1$. We first study the learned states. To that end, consider a call to dfslearn$(p)$ at level $h$ that does not terminate in Line 18 and performs a policy update. Since dfslearn is called recursively for all $p \circ a$ with $a \in \mathcal{A}$ before $p$ is added to LEARNED$(h)$ and every path that dfslearn is called with either makes that path learned or pruned, all successor states of $p$ are in PRUNED$(h)$ or LEARNED$(h)$ when $p$ is added. Since the statement holds for $h + 1$, for all successor paths $p'$ we have $V^{\hat{\pi}_{h+1:H}}(p') \geq V^{\star}(p') - \phi_{h+1} - 8\tau_{val}$. We can apply Lemma 37 and obtain that after changing $\hat{\pi}_h$, it holds that for all $q \in$ LEARNED$(h)$ $V^{\hat{\pi}_{h:H}}(q) \geq V^{\star}(q) - \phi_{h+1} - 8\tau_{val} - 3\tau_{pol} = V^{\star}(q) - \phi_h$. Since that is the only place where the policy changes or a state is added to LEARNED$(h)$, this proves the first part of the statement for level $h$.

For the second part, we can apply Lemma 36 which claims that for all paths $q$ for which TestLearned$(q, h)$ returns true, it holds that $V^{\hat{\pi}_{h:H}}(q) \geq V^{\star}(q) - \phi_h - 8\tau_{val}$. It remains to show that whenever dfslearn returns to a higher level, for all paths $q \in$ PRUNED$(h)$, TestLearned$(q, h)$ evaluates to true. This condition can only be violated when we add a new state to PRUNED$(h)$ or change the policy $\hat{\pi}_{h:H}$.

For the later case, we explicitly check the condition in Lines 26-28 after we change the policy before returning. Therefore dfslearn can only return after Line 28 without further recursive calls to dfslearn if TestLearned evaluated to true for all $q \in$ PRUNED$(h)$. The statement is therefore true if the algorithm returns after Line 28. Further, a path can only be added to PRUNED$(h)$ after we explicitly checked that TestLearned evaluates true for it before we return in Line 18. Hence, the second part of the statement also holds for $h$ which completes the proof. $\square$

**Lemma 39** (Termination). *Assume the deviation bounds hold for all Data sets collected during the first $T_{\max} = 3M^2HK$ calls of* dfslearn *and* TestLearned. *The algorithm terminates during these calls and at all times for all $h \in [H]$ it holds $|$LEARNED$(h)| \leq M$. Moreover, the number of paths that have ever been added to* PRUNED$(h)$ *(that is, counting those removed in Line 26) is at most $KM$.*

*Proof.* Consider a call to TestLearned$(p, h)$ where $p$ leads to the same state as a $q \in$ LEARNED$(h)$. Assume the deviation bounds hold for all data sets collected during this call and before, and we can show that TestLearned must evaluate to true: Using Lemma 38 we get that on all learned paths $p$ it holds that $V^{\hat{\pi}_{h:H}}(p) \geq V^{\star}(p) - \phi_h$. Therefore, $g^{\star}$ is feasible in (22) since $\hat{\mathbf{E}}_{D'}[g^{\star}(x) - \bar{r}] \leq V^{\star}(q) - V^{\hat{\pi}_{h:H}}(q) + 2\tau_{val} \leq \phi_h + 2\tau_{val}$. This allows us to relate $V_{opt}$ to the optimal value as

$$V_{opt} \geq \hat{\mathbf{E}}_D[g^{\star}(x)] \geq V^{\star}(p) - \tau_{val}.$$

It further holds that

$$\hat{\mathbf{E}}_D[\bar{r}] \geq V^{\hat{\pi}_{h:H}}(p) - \tau_{val} = V^{\hat{\pi}_{h:H}}(q) - \tau_{val} \geq \hat{\mathbf{E}}_{D'}[\bar{r}] - 2\tau_{val}.$$

and so the second condition in the if-clause holds. For the first condition, let $\hat{g}$ be the function that achieves the maximum in the computation of $V_{opt}$. Then

$$V_{opt} = \hat{\mathbf{E}}_D[\hat{g}(x_h)] \leq \mathbf{E}_s[\hat{g}(x_h)] + \tau_{val} \leq \hat{\mathbf{E}}_{D'_q}[\hat{g}(x_h)] + 2\tau_{val}$$
$$\leq \hat{\mathbf{E}}_{D'_q}[\bar{r}] + \phi_h + 2\tau_{val} + 2\tau_{val} = \hat{\mathbf{E}}_{D'_q}[\bar{r}] + \phi_h + 4\tau_{val}.$$

Then the first condition is also true and TestLearned returns true. Therefore, TestLearned evaluates to true for all paths that reach the same state as a learned path. As a consequence, if dfslearn is called with such a path it returns in Line 18. Furthermore, as long as all deviation bounds hold, the number of learned paths per level is bounded by $|$LEARNED$(h)| \leq M$.

We next show that the number of paths that have ever appeared in $\textsc{Pruned}(h)$ is at most $KM$. This is true since there are at most $KM$ recursive calls to $\texttt{dfslearn}$ at level $h$ from level $h-1$ and only during those calls a path can be added to $\textsc{Pruned}(h)$ that has not been in $\textsc{Pruned}(h)$ before.

Assume the deviation bounds hold for all data sets collected during the first $T_{\max}$ calls of $\texttt{dfslearn}$. There can be at most $MH$ calls of $\texttt{dfslearn}$ in which a path is learned. Since the recursive call in Line 28 always learns a new state at the next level, the only way to grow $\textsc{Pruned}(h)$ is via the recursive call on Line 20, which occurs at most $MKH$ times. Therefore the algorithm terminates after at most $MH + MHK$ calls to $\texttt{dfslearn}$. Each of these calls can make at most 1 call to $\texttt{TestLearned}$ unless it learns a new state and calls $\texttt{TestLearned}$ up to $|\textsc{Pruned}(h)|+1 \leq MK+1$ times. Therefore, the total number of calls to $\texttt{TestLearned}$ is bounded by $MH(MK+1) + MHK$. The lemma follows by noticing that both numbers of calls are bounded by $T_{\max}$. $\qquad\square$

**Lemma 40.** *Let $\mathcal{E}$ be the event that the deviation bounds in Definition 35 hold for all data sets collected during Algorithm 6. Set $n_{train}$ and $n_{test}$ such that*

$$n_{train} \geq \frac{16K}{\tau_{pol}^2} \log\left( \frac{16T_{\max}M|\Pi||\mathcal{G}|}{\delta} \right)$$

$$n_{test} \geq \frac{1}{2\tau_{val}^2} \log\left( \frac{16T_{\max}M|\Pi||\mathcal{G}|}{\delta} \right)$$

*Then $\mathbf{P}(\bar{\mathcal{E}}) \leq \delta$.*

*Proof.* Consider a single data set $D_q$ collected in $\texttt{dfslearn}(p)$ at level $h$ where $p$ is learned for $q \in \textsc{Learned}(h)$. For the random variable $K\mathbf{1}\{\pi(x_h) = a_h\}\bar{r}$, since $a_h$ is chosen uniformly at random, it is not hard to see that both the variance and the range are upper-bounded by $2K$ (see for example Lemma 14 by Jiang et al. [1]). As such, Bernstein's inequality and a union bound over all $\pi \in \Pi_h$ gives that with probability $1 - \delta'$,

$$|\hat{\mathbf{E}}_{D_q}[K\mathbf{1}\{a = \pi(x)\}\bar{r}] - \mathbf{E}_{q,\hat{\pi}_{h+1}:H}[K\mathbf{1}\{a_h = \pi(x_h)\}\bar{r}]| \leq \sqrt{\frac{4K\log(2|\Pi|/\delta')}{n_{\text{train}}}} + \frac{4K}{3n_{\text{train}}}\log(2|\Pi|/\delta').$$

Consider a single data set $D$ collected in $\texttt{TestLearned}(p, h)$. By Hoeffding's inequality and a union bound, with probability $1 - \delta'$, for all $g \in \mathcal{G}_h$

$$|\hat{\mathbf{E}}_D[g(x_h)] - \mathbf{E}_p[g(x_h)]| \leq \sqrt{\frac{\log(2|\mathcal{G}|/\delta')}{2n_{\text{test}}}}$$

Analogously, for a data set $D'_q$ collected during $\texttt{TestLearned}(p, h)$ with $q \in \textsc{Learned}(q)$, we have with probability at least $1 - \delta'$ that

$$|\hat{\mathbf{E}}_{D'_q}[g(x_h)] - \mathbf{E}_q[g(x_h)]| \leq \sqrt{\frac{\log(2|\mathcal{G}|/\delta')}{2n_{\text{test}}}}$$

Further, again by Hoeffding's inequality and a union bound we get that for a single data set $D$ collected in $\texttt{TestLearned}(p, h)$ and a single data set $D'_q$ collected during $\texttt{TestLearned}(p, h)$ with $q \in \textsc{Learned}(q)$ with probability at least $1 - \delta'$ it holds

$$|\hat{\mathbf{E}}_{D'_q}[\bar{r}] - V^{\hat{\pi}_{h+1}:H}(q)| \leq \sqrt{\frac{\log(4/\delta')}{2n_{\text{test}}}} \quad \text{and}$$

$$|\hat{\mathbf{E}}_D[\bar{r}] - V^{\hat{\pi}_{h+1}:H}(p)| \leq \sqrt{\frac{\log(4/\delta')}{2n_{\text{test}}}}.$$

Combining all these bounds with a union bound and using $\delta' = \frac{\delta}{4MT_{\max}}$, we get that the deviation bounds hold for the first $MT_{\max}$ data sets of the form $D'_q$ and $D_q$ and $D$ with probability at least $1 - \delta$. Using Lemma 39, this is sufficient to show that $\mathbf{P}(\bar{\mathcal{E}}) \leq \delta$. $\qquad\square$

**Proof of Theorem 34.** We now have all parts to complete the proof of Theorem 3.

*Proof.* For the calculation, we instantiate all the parameters as

$$\tau_{pol} = \frac{\epsilon}{6H}, \quad \tau_{val} = \frac{\epsilon}{6H}, \quad \phi_h = (H - h + 1)(8\tau_{val} + 3\tau_{pol}), \quad T_{\max} = 3M^2 HK,$$

$$n_{\text{test}} = \frac{\log(16T_{\max}M|\Pi||\mathcal{G}|/\delta)}{2\tau_{val}^2}, \quad n_{\text{train}} = \frac{16K\log(16T_{\max}M|\Pi||\mathcal{G}|/\delta)}{\tau_{pol}^2}.$$

These settings suffice to apply all of the above lemmas for these algorithms and therefore with these settings the algorithm outputs a policy that is at most $\epsilon$-suboptimal, except with probability $\delta$. For the sample complexity, since $T_{\max}$ is an upper bound on the number of calls to `TestLearned` and at most $M$ states are learned per level $h \in [H]$, we collect a total of at most the following number of episodes:

$$(1 + M)T_{\max}n_{\text{test}} + M^2 H n_{\text{train}}$$
$$= \tilde{O}\left(\frac{T_{\max}MH^2}{\epsilon^2} \log(|\Pi||\mathcal{G}|/\delta) + \frac{M^2 H^3 K}{\epsilon^2} \log(|\Pi||\mathcal{G}|/\delta)\right)$$
$$= \tilde{O}\left(\frac{M^3 K H^3}{\epsilon^2} \log(|\Pi||\mathcal{G}|/\delta)\right). \qquad \square$$

## D  Oracle-Inefficiency of OLIVE

As explained in Section 5 Theorem 4 follows directly from Theorem 5 and Proposition 6 by proof by contradiction with $P \neq NP$. In the following two sections, we first prove Proposition 6 and then Theorem 5.

### D.1  Proof for Polynomial Time of Oracles

*Proof of Proposition 6.* We prove the claim for each oracle separately

1. **CSC-Oracle:** For tabular functions, the objective can be decomposed as

$$n^{-1} \sum_{i=1}^{n} c^{(i)}(\pi(x^{(i)})) = \sum_{x \in \mathcal{X}} n^{-1} \sum_{i=1}^{n} \mathbf{1}\{x = x^{(i)}\}c^{(i)}(\pi(x)). \qquad (24)$$

   Each of the $|\mathcal{X}|$ terms only depend on $\pi(x)$ but not on any action chosen for different observations. Hence, since $\Pi = (\mathcal{X} \to \mathcal{A}) \triangleq \mathcal{A}^{|\mathcal{X}|}$, the action chosen by $\hat{\pi} = n^{-1} \operatorname{argmin}_{\pi \in \Pi} \sum_{i=1}^{n} c^{(i)}(\pi(x^{(i)}))$ for $x \in \mathcal{X}$ is $\operatorname{argmin}_{a \in \mathcal{A}} \sum_{i=1}^{n} \mathbf{1}\{x = x^{(i)}\}c^{(i)}(\pi(x))$. To compute $\hat{\pi}$, we first compute for each $x$ the total cost vector $\sum_{i=1}^{n} \mathbf{1}\{x = x^{(i)}\}c^{(i)}(\pi(x))$ and then pick the smallest entry as the action for $\hat{\pi}(x)$. Per $x$, this takes $O(Kn)$ operations and therefore, the total runtime for this oracle is $O(nK|\mathcal{X}|)$.

2. **LS-Oracle:** Similarly to the CSC objective, the least-squares objective can be decomposed as

$$\sum_{i=1}^{n} (v^{(i)} - g(x^{(i)}))^2 = \sum_{x \in \mathcal{X}} \sum_{i=1}^{n} \mathbf{1}\{x = x^{(i)}\}(v^{(i)} - g(x))^2 \qquad (25)$$

   and therefore $\hat{g} = \operatorname{argmin} g \in \mathcal{G} \sum_{i=1}^{n} (v^{(i)} - g(x^{(i)}))^2$ can be computed for each observation separately. A minimizer per observation $x$ of $\sum_{i=1}^{n} \mathbf{1}\{x = x^{(i)}\}(v^{(i)} - g(x))^2$ is $\hat{g}(x) = \frac{\sum_{i=1}^{n} \mathbf{1}\{x = x^{(i)}\}v^{(i)}}{\sum_{i=1}^{n} \mathbf{1}\{x = x^{(i)}\}}$, where we set $\hat{g}(x)$ arbitrarily if $\sum_{i=1}^{n} \mathbf{1}\{x = x^{(i)}\} = 0$. This can be computed with $O(n)$ operations and therefore the total runtime of the LS-oracle is $O(|\mathcal{X}|n)$.

3. **LP-Oracle:** We parameterize $g \in \mathcal{G}$ by vectors $\theta \in \mathbb{R}^{|\mathcal{X}|}$ where each the value of $g$ for each $x \in \mathcal{X}$ is associated with a particular entry $\theta_x$ of $\theta$. Then the LP problem reduces to a

Figure 2: Family of MDPs that are determined up to terminal rewards $r_1, \ldots, r_n \in [0, 1]$. Finding the optimal value of the most optimistic MDP in this family solves the encoded 3-SAT instance. Solid arrows represent actions and dashed arrows represent random transitions.

standard linear program in $\mathbb{R}^{|\mathcal{X}|}$. Khachiyan [37], Grötschel et al. [45] have shown using the ellipsoid method, these problems can be solved approximately in polynomial time. Note that the initial ellipsoid can be set to any ellipsoid containing $[0, 1]^{|\mathcal{X}|}$ due to the normalization of rewards. Further, the volume of the smallest ellipsoid can be upper bounded by a polynomial in $\epsilon_{\text{feas}}$ using the fact that we only require a solution that is feasible up to $\epsilon_{\text{feas}}$ and applying the ellipsoid method to the extended polytope with all constraints relaxed by $\epsilon_{\text{feas}}$. □

### D.2  OLIVE is NP-hard in tabular MDPs

Instead of showing Theorem 5 directly, we first show the following simpler version:

**Theorem 41.** *Let $\mathcal{P}$ denote the family of problems of the form* (3)*, parameterized by $(\mathcal{X}, \mathcal{A}, D_0, \mathcal{D})$ with implicit $\mathcal{G} = (\mathcal{X} \to [0, 1])$ and $\Pi = (\mathcal{X} \to \mathcal{A})$ (i.e., the tabular value-function and policy classes) and with $\phi = 0$. $\mathcal{P}$ is NP-hard.*

Some remarks are in order about this statement

1. Our proof actually shows that it is NP-hard to find an $\epsilon$-approximate solution to these optimization problems, for polynomially small $\epsilon$.

2. The two theorems differ in whether the data sets ($D_i \in \mathcal{D}$) are chosen adversarially (Theorem 41), or induced naturally from an actual run of OLIVE (Theorem 5). Therefore, Theorem 5 is strictly stronger.

3. At a high level, these results imply that OLIVE in general must solve NP-hard optimization problems, presenting a barrier for computational tractability.

4. These results also hold with imperfect expectations and polynomially small $\phi$.

5. We use the $(\mathcal{G}, \Pi)$ representation here but similar results hold with $\mathcal{F}$ representation (i.e., approximating the $Q$-function; see Theorems 42 and 43).

For intuition we first sketch the proof of Theorem 41. The complete proof follows below.

*Proof Sketch of Theorem 41.* We reduce from 3-SAT. Let $\psi$ be a 3-SAT formula on $n$ variables $x_1, \ldots, x_n$ with $m$ clauses $c_1, \ldots, c_m$. We construct a family of MDPs as shown in Figure 2 that encodes the 3-SAT problem for this formula as follows: For each variable $x_i$ there are two terminal states $x_i^1$ and $x_i^0$ corresponding to the Boolean assignment to the variable. For each variable, the reward in either $x_i^1$ or $x_i^0$ is 1 and 0 in the other. The family of MDPs contains all possible combinations of such terminal rewards. There is also one state per clause $c_j$ and one start state $s_0$. From each clause, there are 7 actions, one for each binary string of length 3 except "000." These actions all receive zero instantaneous reward. For clause $c_\ell = x_i \vee \bar{x}_j \vee \bar{x}_k$, the action "$b_1 b_2 b_3$" transitions to states $x_i^{b_1}, x_j^{1-b_2}$, or $x_k^{1-b_3}$, each with probability $1/3$. The intuition is that the action describes which literals evaluate to true for this clause. From the start state, there are $n + m + 1$

actions. For each variable $x_i$, there is a [try $x_i$] action that transitions uniformly to $x_i^0, x_i^1$ and receives 0 instantaneous reward. For each clause $c_j$ there is a [try $c_j$] action that transitions deterministically to the state for clause $c_j$, but receives reward $-1/n$. And finally there is a [solve] action that transitions to a clause state uniformly at random.

For each $x_i$, we introduce a constraint into Problem (3) corresponding to the [try $x_i$] action. These constraints impose that the optimal $\hat{g} \in \mathcal{G}$ satisfies $\forall i \in [m] : \hat{g}(x_i^0) + \hat{g}(x_i^1) = 1$. We also introduce constraints for the [try $c_j$] actions and from $s_0$. Recall that values must be in $[0, 1]$.

With these constraints, if the 3-SAT formula has a satisfying assignment, then the optimal value from the start state is 1, and it is not hard to see that there exists function $\hat{g} \in \mathcal{G}$ that achieves this optimal value, while satisfying all constraints with a $\hat{\pi} \in \Pi$. Conversely, if the value of the start date is 1, we claim that the 3-SAT formula is satisfiable. In more detail, the policy must choose the [solve] action, and the value function must predict that each clause state has value 1, then the literal constraints enforce that exactly one of $x_i^0, x_i^1$ has value 1 for each $i$. Thus the optimistic value function encodes a satisfying assignment, completing the reduction. $\qquad\square$

### D.2.1 Proof of Theorem 41

In this section, we prove that the optimization problem solved by OLIVE is NP-hard. The proofs rely on the fact that OLIVE only adds a constraint for a single time step $h$ that has high average Bellman error. However, using an extended construction, one can show similar statements for a version of OLIVE that adds constraints for all time steps if there is high average Bellman error in any time step.

For notational simplicity, we do not prove Theorem 41 and Theorem 5 directly, but versions of these statements below with a tabular $Q$-function representation $\mathcal{F}$ instead of the $(\mathcal{G}, \Pi)$ version presented in the paper. For this formulation, OLIVE picks the policy for the next round as the greedy policy $\pi_{\hat{f}_k}$ of the $Q$-function that maximizes

$$\hat{f}_k = \underset{f \in \mathcal{F}}{\operatorname{argmax}} \hat{\mathbf{E}}_{D_0}[f(x, \pi_f(x))] \qquad (26)$$

$$\text{s.t.} \ \ \forall \ D_i \in \mathcal{D} :$$

$$|\hat{\mathbf{E}}_{D_i}[\mathbf{1}\{a = \pi_f(x)\}(f(x, a) - r - f(x', \pi_f(x')))]| \leq \phi.$$

This proof naturally extends to the $(\mathcal{G}, \Pi)$ representation: note that OLIVE runs in a completely equivalent way if it takes a set of $(g, \pi)$ pairs induced by $\mathcal{F}$ as inputs, i.e., $\{(x \mapsto f(x, \pi_f(x)), x \mapsto \pi_f(x)) : f \in \mathcal{F}\}$ [1, see Appendix A.2,]. When $\mathcal{F}$ is the tabular $Q$-function class, it is easy to verify that the induced set is the same as $\mathcal{G} \times \Pi$ where $\mathcal{G}$ and $\Pi$ are the tabular value-function / policy classes respectively. Therefore, the proof for Theorem 41 just requires a simple substitution where $f(x, \pi_f(x))$ is replaced by $g(x)$ and $\pi_f(x)$ is replaced by $\pi$.

We first prove the simpler NP-hardness claim.

**Theorem 42** ($\mathcal{F}$-Version of Theorem 41)**.** *Let $\mathcal{P}$ denote the family of problems of the form* (26)*, parameterized by $(\mathcal{X}, \mathcal{A}, D_0, \mathcal{D})$ with implicit $\mathcal{F} = (\mathcal{X} \times \mathcal{A} \to [0, 1])$ (i.e., the tabular $Q$-function class) and with $\phi = 0$. $\mathcal{P}$ is NP-hard.*

*Proof.* For the ease of presentation, we show the statement for $\mathcal{F} = (\mathcal{X} \times \mathcal{A} \to [-1, 1])$ and all values scaled to be in $[-1, 1]$. By linearly transforming all rewards accordingly, one obtains a proof for the statement with all values in $[0, 1]$.

We demonstrate a reduction from 3-SAT. Recall that an instance of 3-SAT is a Boolean formula $\psi$ on $n$ variables can be described by a list of clauses $C_1, \ldots C_m$ each containing at 3 literals (a variable $x_i$ or its negation $\bar{x}_i$), e.g. $C_1 = (\bar{x}_2 \vee x_3 \vee \bar{x}_5)$. As notation let $o_{j,i}^1$ for $i \in \{1, 2, 3\}$ denote the $i^{\text{th}}$ literal in the $j^{\text{th}}$ clause and $o_{j,i}^0$ its negation (e.g. $o_{1,3}^1 = \bar{x}_5$ and $o_{1,3}^0 = x_5$). Given a 3-SAT instance with $m$ clauses $C_{1:m}$ and $n$ variables $x_{1:n}$, we define a class of finite episodic MDPs $\mathcal{M}$. This class contains (among others) $2^n$ MDPs that correspond each to an assignment of Boolean values to $x_{1:n}$.

The proof proceeds as follows: First we describe the construction of this class of MDPs. Then we will demonstrate a set of constraints for the OLIVE program. Importantly, these constraints do not distinguish between the $2^n$ MDPs in the class $\mathcal{M}$ corresponding to the binary assignments to the variables $x_{1:n}$, so the optimistic planning step in OLIVE needs to reason about all possible

Figure 3: Family of MDPs $\mathcal{M}$ for a specific instance of a 3-SAT problem.

assignments. Finally, we show that with the function class $\mathcal{F} = (\mathcal{X} \times \mathcal{A}) \to [-1, 1]$, the solution to the optimization problem (26) determines whether $\psi$ is satisfiable or not.

For simplicity, the MDPs in $\mathcal{M}$ have different actions available in different states and rewards are in $[-1, 1]$ instead of the usual $[0, 1]$. We can however find equivalent MDPs that satisfy the formal requirements of OLIVE.

**MDP structure.** Let $\psi$ be the 3-SAT instance with variables $x_{1:n}$ and clauses $C_{1:m}$. The state space for MDPs in $\mathcal{M}$ consists of $m + 2n + 1$ states, two for each variable, one for each clause, and one additional starting state. For each variable $x_i$, there are two states $x_i^0, x_i^1$ corresponding to the variable and its negation. Each clause $C_j$ has a state $C_j$, and the starting state is denoted $s_0$.

The transitions are as follows: The states $x_i^0, x_i^1$ corresponding to the literals are terminal, with just a single action. The class $\mathcal{M}$ differs only in how it assigns rewards to these terminal states. Specifically let $y \in \{0, 1\}^n$ be a binary vector, then there is an MDP $M_y \in \mathcal{M}$ where for all $i \in [n]$ the reward for literal $x_i^{y_i} = 1$ and $x_i^{1-y_i} = 0$. Specifically, all MDPs in $\mathcal{M}$ have values that satisfy $V(x_i^1) + V(x_i^0) = 1$ for all $i \in [n]$.

Each clause state $C_j$ has 7 actions, indexed by $b \in \{0, 1\}^3 \setminus \{\text{"000"}\}$, each corresponding to an assignment of the variables that would satisfy the clause. Taking an action $b$ transitions the agent to three literal states with equal probability $1/3$ and the agent receives no immediate reward. Which literals is determined by the clause. Assume the clause consists of $C_t = (\bar{x}_i \vee x_j \vee \bar{x}_k)$. Then

$$\mathbf{P}(x_i^1|c_t, b) = \frac{1}{3}\mathbf{1}\{b_1 = 0\}, \quad \mathbf{P}(x_i^0|c_t, b) = \frac{1}{3}\mathbf{1}\{b_1 = 1\}$$

$$\mathbf{P}(x_j^1|c_t, b) = \frac{1}{3}\mathbf{1}\{b_2 = 1\}, \quad \mathbf{P}(x_j^0|c_t, b) = \frac{1}{3}\mathbf{1}\{b_2 = 0\}$$

$$\mathbf{P}(x_k^1|c_t, b) = \frac{1}{3}\mathbf{1}\{b_3 = 0\}, \quad \mathbf{P}(x_k^0|c_t, b) = \frac{1}{3}\mathbf{1}\{b_3 = 1\}.$$

For example, taking action 011 in clause state $C_1 = (\bar{x}_2 \vee x_3 \vee \bar{x}_5)$ transitions with equal probability to $x_2^1$ (since the first component of the action is 0), $x_3^1$ (second component is 1) and $x_5^0$ (last component is 1).

The initial state has $n + m + 1$ actions. The first set of actions are labeled [try $x_i$], for $i \in [n]$. They receive zero instantaneous reward and transition uniform to $x_i^1, x_i^0$. The second set of actions are labeled [try $C_j$] (for $j \in [m]$), which receives $1/m$ instantaneous reward and transitions deterministically to $c_j$. Finally there is a [solve] action that transitions uniformly to the $\{C_j\}_{j=1}^m$ states and receives zero instantaneous reward.

**OLIVE Constraints.** We introduce constraints at the start state $s_0$, all of the constraint states $C_j$, and the distributions induced when taking the [try $x_i$] action. Since the literal states $x_i^1, x_i^0$ have no actions, we omit the second argument from the $Q$-functions $f$. We list these constraints in the

following writing out the constraints for each optimal action that are implied by the indicator of the original constraints in Problem (26): From initial state:

$$f(s_0, \texttt{[try } c_j\texttt{]}) = \max_b f(c_1, b) - 1/m \qquad \text{if } \pi_f(s_0) = \texttt{[try } c_j\texttt{]} \qquad (27)$$

$$f(s_0, \texttt{[solve]}) = \frac{1}{m} \sum_{i=1}^{m} \max_b f(C_j, b) \qquad \text{if } \pi_f(s_0) = \texttt{[solve]} \qquad (28)$$

$$f(s_0, \texttt{[try } x_i\texttt{]}) = \frac{f(x_i^0) + f(x_i^1)}{2} \qquad \text{if } \pi_f(s_0) = \texttt{[try } x_i\texttt{]} \qquad (29)$$

From clause $j$ after [try $C_j$]:

$$f(C_j, b) = \frac{f(o_{j,1}^{b(1)}) + f(o_{j,2}^{b(2)}) + f(o_{j,3}^{b(3)})}{3} \qquad \text{if } \pi_f(C_j) = b \qquad (30)$$

From variable $i$ after [try $x_i$]:

$$\frac{f(x_i^1) + f(x_i^0)}{2} = \frac{1}{2} \qquad (31)$$

Note that all appearances of $f$ on the LHS could be replaced by $f(\cdot, \pi_f(\cdot))$. There are other types of constraints involving literal states that could be imposed, specifically constraints of the form

$$\sum_{i=1}^{m} w_{2i-1} f(x_i^1) + w_{2i} f(x_i^0) = V \qquad (32)$$

for some $V$ and $w \in \Delta([2m])$, which appears by first applying [solve] or [try $C_j$] and then various actions at the clause states to arrive at a distribution over the literal states. It is important here that constraints of this type are *not* included in the optimization problem, since it distinguishes elements of the family $\mathcal{M}$.

**The Optimal Value.**  Consider the OLIVE optimization problem (26) on the family of MDPs $\mathcal{M}$ with constraints described above. Note that all MDPs in the family generate identical constraints, so formulating the optimization problem does not require determining whether $\psi$ has a satisfying assignment or not.

Now, if $\psi$ has a satisfying assignment, say $y^\star \in \{0, 1\}^n$, then the MDP $M_{y^\star} \in \mathcal{M}$ has optimal value 1. Moreover since the function class $\mathcal{F}$ is entirely unconstrained, this function class can achieve this value, which is the solution to Problem (26). To see why $M_{y^\star}$ has optimal value 1, consider the policy that chooses the [solve] action and from each clause chooses the 3-bit string that transitions to the literal states that have value 1. Importantly, since $\psi$ has a satisfying assignment, this must be true for one of the 7 actions.

Conversely, suppose that Problem (26), with all the constraints described above, has value 1. We argue that this implies $\psi$ has a satisfying assignment. Let $\hat{f}, \hat{\pi}$ correspond to the $Q$-value and policy that achieve the optimal value in the program. First, due to the constraints on the [try $x_i$] distributions and the immediate negative rewards for taking [try $C_j$] actions, we must have $\hat{\pi}(s_0) = \texttt{[solve]}$ and $\hat{f}(s_0, \texttt{[solve]}) = 1$. The constraints on $\hat{f}$ now imply that for each clause $C_j$ there exists a action $b_j \in \{0, 1\}^3 \setminus \{000\}$ such that $\hat{f}(C_j, b_j) = 1$. Proceeding one level further, if $b_j$ satisfies $\hat{f}(C_j, b_j) = 1$ then we must have that $\hat{f}(o_{j,k}^{b_j(k)}) = 1$ for all $k \in \{1, 2, 3\}$. And due to the boundedness conditions on $\hat{f}$ along with the constraint that $\hat{f}(x_i^0) + \hat{f}(x_i^0) = 1$, one of these values must be 1, while the other is zero. Therefore, for any variable that appears in some clause the corresponding literal states must have predicted value that is binary. Since the constraints corresponding to the clauses are all satisfied (or else we could not have value 1 at $s_0$), the predicted values at the literal states encodes a satisfying assignment to $\psi$. $\qquad \square$

### D.2.2  Proof of Theorem 5

After showing that Problem (26) is NP-hard when constraints are chosen adversarially, we extend this result to the class of problems encountered by running OLIVE. Again, we prove a version of the statement with $\mathcal{F}$ representation but the proof for Theorem 5 is completely analogous.

**Theorem 43** ($\mathcal{F}$ *Version of Theorem* 5)**.** *Let* $\mathcal{P}_{\text{OLIVE}}$ *denote the family of problems of the form* (26)*, parameterized by* $(\mathcal{X}, \mathcal{A}, \text{Env}, t)$*, which describes the optimization problem induced by running* OLIVE *in the MDP environment Env (with states* $\mathcal{X}$*, actions* $\mathcal{A}$ *and perfect evaluation of expectations) for* $t$ *iterations with* $\mathcal{F} = (\mathcal{X} \times \mathcal{A} \to [0,1])$ *and with* $\phi = 0$*.* $\mathcal{P}_{\text{OLIVE}}$ *is NP-hard.*

*Proof.* The proof uses the same family of MDPs $\mathcal{M}$ and set of constraints as the proof of Theorem 42 above. As mentioned there, it is crucial that constraints in Equations (27)-(31) are added for all clauses and literals but none of the possible constraints of the form in Equation (32) that arise from distributions over literal states after taking actions [try $C_j$] or [Solve]. To prove that OLIVE can encounter NP-hard problems, it therefore remains to show that running OLIVE on any MDP in $\mathcal{M}$ can generate the exact set of constraints in Equations (27)-(31).

The specification of OLIVE by Jiang et al. [1] only prescribes that a constraint for one time step $h$ among all that have sufficiently large average Bellman error is added. It however leaves open how exactly $h$ is chosen and which $f \in \mathcal{F}$ is chosen among all that maximize Problem (26). Since this component of the algorithm is under-specified, we choose $h$ and $f \in \mathcal{F}$ in an adversarial manner within the specification, which amounts to adversarial tie breaking in the optimization.

We now provide a run of OLIVE on an arbitrary MDP in $\mathcal{M}$ that generates exactly the set of constraints in Equations (27)-(31):

- For the first $t \in [m]$ iterations, OLIVE picks any Q-function $f_t \in \mathcal{F}$ with $f_t(s_0, b) = \mathbf{1}\{b = $ [try $C_t$]$\}$ and $f_t(C_t, b) = 1$ and $f_t(x_i^0, \pi_{f_t}(x_i^0)) = f_t(x_i^1, \pi_{f_t}(x_i^1)) = 0$ for all actions $b$ and $i \in [n]$ and chooses to add constraints for $h = 2$. Since the context distributions is a different $C_t$ for every iteration $t$, this is a valid choice and generates constraints

$$f(C_t, b) = \frac{f(o_{t,1}^{b(1)}) + f(o_{t,2}^{b(2)}) + f(o_{t,3}^{b(3)})}{3} \qquad \text{if } \pi_f(C_t) = b$$

  for all $b$.

- For the next $n$ iterations $t = m+1, m+2, \ldots m+n$, OLIVE picks any Q-function $f_t \in \mathcal{F}$ with $f_t(s_0, b) = \mathbf{1}\{b = $ [try $x_{t-m}$]$\}$ and $f_t(x_{t-m}^0, \pi_{f_t}(x_{t-m}^0)) = f_t(x_{t-m}^1, \pi_{f_t}(x_{t-m}^1)) = 1$ for all $b$. The only positive average Bellman error occurs in the mixture over literal states at $h = 2$ and therefore constraints

$$\frac{f(x_{t-m}^1, \pi_f(x_{t-m}^1)) + f(x_{t-m}^0, \pi_f(x_{t-m}^0))}{2} = \frac{1}{2}$$

  are added.

- Finally, in iteration $t = m+n+1$, OLIVE picks any $f_t \in \mathcal{F}$ with $f_t(s_0, b) = \mathbf{1}\{b = $ [try $x_1$]$\}$ and $f_t(x_1^0, \pi_{f_t}(x_1^0)) = f_t(x_1^1, \pi_{f_t}(x_1^1)) = 1/2$ for all actions $b$. Now there is positive average Bellman error in the initial state $s_0$ and with $h_t = 1$ the following constraints are added

$$f(s_0, [\text{try } c_j]) = \max_b f(C_1, b) - 1/m \qquad \text{if } \pi_f(s_0) = [\text{try } c_j]$$

$$f(s_0, [\text{solve}]) = \frac{1}{m} \sum_{i=1}^{m} \max_b f(C_j, b) \qquad \text{if } \pi_f(s_0) = [\text{solve}]$$

$$f(s_0, [\text{try } x_i]) = \frac{f(x_i^0) + f(x_i^1)}{2} \qquad \text{if } \pi_f(s_0) = [\text{try } x_i]$$

  for all $i \in [n]$ and $j \in [m]$.

Since at iteration $t = m+n+2$, the set of constraints matches exactly the one in the proof of Theorem 42, OLIVE solves exactly the problem instance described there which solves the given 3-SAT instance. $\qquad\square$

# E   Additional Barriers

In this section, we describe several further barriers that we must resolve in order to obtain tractable algorithms in the stochastic setting.

Figure 4: Further barriers to tractable algorithms. Circles denote states, while rectangles denote observations. Solid lines denote deterministic transitions. Dashed lines denote stochastic transitions (middle) or context emissions (right). Left: construction for Theorem 45, where $\Delta := g_{\text{bad}} - g^\star$ reflects the amount that $g_{\text{bad}}$ over-predicts in each state. On the upper chain, statistical fluctuations can favor $g_{\text{bad}}$ over $g^\star$, which leads to a policy choosing the wrong action at the start. Center: construction for Proposition 46, where most policies induce a uniform distribution over states at level two and an average constraint cannot drive the agent to the top state. Right: construction for Proposition 47 where an $\epsilon$ loss in roll-out policy converts into a $\sqrt{\epsilon}$ prediction error in value function.

## E.1 Challenges with Credit Assignment

We start with the learning step, ignoring the challenges with exploration, and focus on a family of algorithms that we call *Bellman backup* algorithms.

**Definition 44.** *A* Bellman backup *algorithm collects $n$ samples from every state and iterates the policy/value updates*

$$\hat{\pi}_h = \underset{\pi \in \Pi_h}{\operatorname{argmax}} \sum_{s \in \mathcal{S}_h} \mathbf{E}_s[r + \hat{g}_{h+1}(x') | a = \pi(x)]$$

$$\hat{g}_h = \underset{g \in \mathcal{G}_h}{\operatorname{argmin}} \sum_{s \in \mathcal{S}_h} \mathbf{E}_s[(g(x) - r - \hat{g}_{h+1}(x'))^2 | a = \hat{\pi}_h(x)].$$

This algorithm family differs only in the exploration component, which we are ignoring for now, but otherwise is quite natural. In fact, these algorithms can be viewed as a variants of Fitted Value Iteration (FVI)[7] [41, 47] adapted to the $(g, \pi)$ representation. Unfortunately, such algorithms cannot avoid exponential sample complexity, even ignoring exploration challenges.

**Theorem 45.** *For any $H \geq 1$, $\epsilon \in (0, 1)$, there exists a layered tabular MDP with $H$ levels, 2 states per level, and constant-sized $\mathcal{G}$ and $\Pi$ satisfying Assumptions 7 and 8, such that when $n < 4^H/(32\epsilon^2)$, with probability at least $1/4$, the bellman backup algorithm outputs a policy $\hat{\pi}$ such that $V^{\hat{\pi}} \leq V^\star - \epsilon$.*

A sketch of the construction is displayed in the left panel of Figure 4. The intuition is that statistical fluctuations at the final state can cause bad predictions, which can exponentiate as we perform the backup. Ultimately this can lead to choosing a exponentially bad action at the starting state. The full proof follows:

*Proof.* Consider an MDP with $H + 1$ levels with deterministic transitions and with one start state $x_0$ and two states per level $\{x_{h,a}, x_{h,b}\}_{1 \leq h \leq H}$. From the start state there are two actions $a, b$ where $a$ transitions to $x_{1,a}$ and $b$ transitions to $x_{1,b}$. From then on, there is just one action which transitions from $x_{h,z}$ to $x_{h+1,z}$ $z \in \{a, b\}$. The reward from $x_{H,a}$ is $\text{Ber}(1/2 + \epsilon)$ and the reward from $x_{H,b}$ is $\text{Ber}(1/2)$. Both value functions in the class have $g(x_{h,a}) = 1/2 + \epsilon$. $g^\star$ is in the class and it has $g^\star(x_0) = 1/2 + \epsilon$, $g^\star(x_{h,b}) = 1/2$. There is also a bad function $g_{bad}(x_0) = 1/2 + \epsilon$, $g_{bad}(x_{h,b}) = 1/2 + \epsilon/2^{h-1}$.

Since in our construction there are only two policies, and they only differ at $x_0$, for the majority of the proof we can focus on policy evaluation. The first step is to show that with non-trivial probability, we will select $g_{bad}$ in the first square loss problem. Since all functions make the same predictions on

$x_{H,a}$ we focus on $x_{H,b}$. Our goal is to show that $g_{bad}$ will be chosen by the algorithm with substantial probability.

**A lower bound on the binomial tail.** The rewards from $x_{H,b}$ are drawn from $\text{Ber}(1/2)$. Call this values $r_1, \ldots, r_n$ with average $\bar{r}$. We select $g_{bad}$ if $\bar{r} \geq 1/2 + \epsilon/2^H$. By Slud's lemma, the probability is

$$\mathbb{P}[\bar{r} \geq 1/2 + \epsilon/2^H] \geq 1 - \Phi\left(\frac{n\epsilon/2^H}{\sqrt{n/4}}\right)$$

where $\Phi$ is the standard Gaussian CDF, which can be upper bounded by

$$\Phi(x) = \frac{1}{\sqrt{2\pi}} \int_{-\infty}^{x} \exp(-u^2/2)du = \frac{1}{2} + \frac{1}{\sqrt{2\pi}} \int_{0}^{x} \exp(-u^2/2)du \leq \frac{1}{2} + \frac{x}{\sqrt{2\pi}}.$$

Thus the probability is at least

$$\geq \frac{1}{2} - \frac{1}{\sqrt{2\pi}} \frac{n\epsilon/2^H}{\sqrt{n/4}} \geq \frac{1}{2} - \sqrt{\frac{8}{2\pi}}\epsilon/2^H\sqrt{n} \geq 1/4$$

where the final inequality holds since $n \leq \frac{4^H}{32\epsilon^2} < \frac{\pi 4^H}{64\epsilon^2}$.

Thus with probability at least $1/4$ the average reward from state $x_{H,b}$ is at least $1/2 + \epsilon/2^H$, in which case $g_{bad}$ is the square loss minimizer. From this point on, at every subsequent level $1 < h < H$, both $g$ and $g_{bad}$ have the same square loss and tie-breaking can cause $g_{bad}$ to always be chosen. Thus the final policy optimization step uses $g_{bad}$ to approximate the future but $g_{bad}(x_{1,a}) = g_{bad}(x_{1,b}) = 1/2 + \epsilon$. Thus the final policy can select action $b$, which leads to a loss of $\epsilon$. $\qquad\square$

Actually, the policy optimization step is inconsequential in the construction. As such, the theorem shows that FVI style learning rules cannot avoid bias that propagates exponentially without further assumptions, leading to an exponential sample complexity requirement. We emphasize that the result focuses exclusively on the learning rule and applies *even* with small observation spaces and *regardless* of exploration component of the algorithm, with similar conclusions holding for variations including $Q$-representations and different loss functions. Theorem 45 provides concrete motivation for stronger realizability conditions such as Assumption 9, variants of which are also used in prior analysis of FVI-type methods [41].

### E.2 Challenges with Exploration

We now turn to challenges with exploration that arise when factoring the $Q$-function class into the $(g, \pi)$ pairs, which works well in the deterministic setting, as in Section 4. However, the stochastic setting presents further challenges. Our first construction shows that a decoupled approach using OLIVE's average Bellman error in the learning rule can completely fail to learn in the stochastic setting.

Consider an algorithm that uses an optimistic estimate for $\hat{g}_{h+1}$ to find a policy $\hat{\pi}_h$ that drives further exploration. Specifically, suppose that we find an estimate $\hat{g}_{h+1}$ such that for all previously visited distributions $D \in \mathcal{D}_{h+1}$ at level $h+1$

$$\hat{\mathbf{E}}_D[\hat{g}_{h+1}(x)] = \hat{\mathbf{E}}_D[g^\star(x)], \tag{33}$$

where we assume that all expectations are exact. We may further encourage $\hat{g}_{h+1}$ to be optimistic over some distribution that provides good coverage over the states at the next level. Then, we use $\hat{\pi}_h = \text{argmax}_\pi \hat{\mathbf{E}}_{D_h}[r + \hat{g}_{h+1}(x')|a = \pi(x)]$ as the next exploration policy. Intuitively, optimism in $\hat{g}_{h+1}$ should encourage $\hat{\pi}_h$ to visit a new distribution, which will drive the learning process. Unfortunately, the next proposition shows that this policy $\hat{\pi}$ may be highly suboptimal and also fail to visit a new distribution.

**Proposition 46.** *There exists a problem, in which the algorithm above stops exploring new distributions when the best policy it finds is worse than the optimal policy by constant value.*

We sketch the construction in the center panel of Figure 4. We create a two level problem where most policies lead to a uniform mixture over two subsequent states, one good and one bad. Constraint (33) on this distribution favors a value function that predicts $1/2$ on both states, and with this function, the optimistic policy leads us back to the uniform distribution. Thus no further exploration occurs!

*Proof of Proposition 46.* Consider a two-layer process with one initial state $s_0$ with optimal value $V(s_0) = 1$ and two future states $s_1, s_2$ where $V(s_1) = 1$, $V(s_2) = 0$. From $s_0$ action $a$ deterministically transitions to $s_1$ and action $b$ deterministically transitions to $s_2$, from $s_1$ and $s_2$ all actions deterministically receive the corresponding reward. No rewards are received upon making the first transition. There are $m$ contexts that are equally likely from $s_0$ and the policy class consists of one policy, $\pi^\star$ that always chooses action $a$, and $\Omega(2^m)$ bad policies that choose action $a, b$ with equal probability. These policies each have value $1/2$.

If we perform a roll-in with a bad policy, we generate a constraint of the form $|g(s_1)/2 + g(s_2)/2 - 1/2| \leq \epsilon$ at level two. Hence, with this constraint we might pick $\hat{g}_{h+1}$ such that $\hat{g}_{h+1}(s_1) = \hat{g}_{h+1}(s_2) = 1/2$, since it satisfies all the constraints and also has maximal average value on any existing roll-in. However, using this future-value function in the optimization

$$\mathbb{E}_{x_h \sim s_0}[r_h + \hat{g}_{h+1}(s')|a_h = \pi(x_h)], \tag{34}$$

we see that all policies, including $\pi^\star$ have the same objective value. When we choose any one of them but $\pi^\star$, the "optimistic" value computed by maximizing Eq.(34) will be achieved by the chosen policy and the algorithm stops exploration with a suboptimal policy. $\qquad\square$

The main point is that by using the average value constraints (33), we lose information about the "shape" of $g^\star$, which can be useful for exploration. In fact, the proposition does not rule out approaches that learn the shape of the state-value function, for example with square loss constraints that capture higher order information. However square loss constraints are less natural for value based reinforcement learning, as we show in the next proposition. We specifically focus on measuring a value function by its square loss to a near optimal roll-out.

**Proposition 47.** *In the environment in the right panel of Figure 4, an $\epsilon$-suboptimal policy $\hat{\pi}$ achieves reward $0$, and the square loss of $g^\star$ w.r.t. the roll-out reward is $\mathbb{E}_{x \sim s}[(g^\star(x) - r)^2 \mid a \sim \hat{\pi}] = \epsilon$. This square loss is also achieved by a bad value function $g_{bad}$ such that $\mathbb{E}_{x \sim s}[g_{bad}(x) - g^\star(x)] = O(\sqrt{\epsilon})$.*

The claim here is weaker than the previous two barriers, but it does demonstrate some difficulty with using square loss in an approach that decouples value function and policy optimization. The essence is that a roll-out policy $\hat{\pi}$ that is slightly suboptimal on average may have significantly lower variance than $\pi^\star$. Since the square loss captures variance information, this means that $g^\star$ may have significantly larger square loss to $\hat{\pi}$'s rewards, which either forces elimination of $g^\star$ or prevents us from eliminating other bad functions, like $g_{bad}$ in the example.

*Proof of Proposition 47.* Consider a process with $H = 1$ and just one state with two observations: $x_1$ and $x_2$, both with two actions. For $x_1$, both actions receive $0$ reward, while for $x_2$ action $a$ receives reward $1$ while action $b$ receives reward $0$. However, observation $x_2$ appears only with probability $\epsilon$. As such, an $\epsilon$-optimal policy from this state may choose action $b$ on both contexts, receiving zero reward. Let $\hat{\pi}$ denote this near-optimal policy.

The value function class $\mathcal{G}$ provided to the algorithm has many functions, but three important ones are (1) $g_0$ which always predicts zero and is the correct value function for $\hat{\pi}$ above, (2) $g^\star$ which is the optimal value function, and (3) $g_{bad}$, which we now define. These latter two function have

$$g^\star(x_1) \triangleq 0, \quad g^\star(x_2) \triangleq 1$$
$$g_{bad}(x_1) \triangleq \sqrt{\epsilon}, \quad g_{bad}(x_2) \triangleq \sqrt{\epsilon}$$

Now, let us calculate the square loss of these three value functions to the roll-out achieved by $\hat{\pi}$.

$$\text{sqloss}(g_0, b_H, \hat{\pi}) = (1 - \epsilon)(0 - 0)^2 + \epsilon(0 - 0)^2 = 0$$
$$\text{sqloss}(g^\star, b_H, \hat{\pi}) = (1 - \epsilon)(0 - 0)^2 + \epsilon(1 - 0)^2 = \epsilon$$
$$\text{sqloss}(g_{bad}, b_H, \hat{\pi}) = (1 - \epsilon)(\sqrt{\epsilon} - 0)^2 + \epsilon(\sqrt{\epsilon} - 0)^2 = \epsilon$$

We see that $g_{\text{bad}}$ and $g^\star$ have identical square loss on this single state, which proves the claim.

Intuitively, this is bad because if we use constraints defined in terms of square loss, we risk eliminating $g^\star$ from the feasible set, or we need the constraint threshold to be so high that bad functions like $g_{\text{bad}}$ remain. These bad function can cause exploration to stagnate or introduce substantial bias depending on the learning rule. $\qquad\square$

To summarize, in this section we argue for the necessity of completeness type conditions for FVI-type learning procedures, and demonstrate barriers for exploration with decoupled optimization approaches, both with expectation and square loss constraints. We believe overcoming these barriers is crucial to the development of a computationally efficient algorithm.