[Reviews · NeurIPS 2018]

Reviewer 1



After author response: thanks for your response to my comments. I still think the paper should be accepted. ==================== Reinforcement learning (RL) has started gaining huge popularity in recent years due to impressive empirical success (e.g., AI defeated the world champion in Go). Unfortunately, this success is still unexplainable theoretically. Specifically, there is no mathematical explanation to RL in the case that there are many observations, as in real life RL problems. The paper presents an algorithm that is proven to be successful in the rich (i.e., many) observations setting, under some assumptions. Their algorithm, VALOR, is based on an algorithm, LSVEE, from a preceding paper [17]. They prove, under some assumptions (like realizability, a small number of hidden states and other assumptions that will be discussed later) that VALOR is PAC-RL (see e.g., Reinforcement Learning in Finite MDPs: PAC Analysis). This paper is very well written. The paper addresses the important mission of gaining a theoretical understanding of RL with rich observations and the paper's perspective is an interesting one. On the downside, there are many assumptions that the paper is making that somewhat weaken their result. Some of the assumptions are highlighted (e.g, the transitions between the hidden states are deterministic), and some are more subtle but important nonetheless and should be explained further: 1. For Theorem 3 to be meaningful, the value function class and the policy class should not be too large. 2. In line 98, it is assumed that the observations obey the Markovian assumption. It will be interesting to see experiments proving the empirical success of this algorithm.

Reviewer 2



This paper studies episodic reactive reinforcement learning with deterministic transitions where we have access to computational oracles (the method proposed uses a linear programming solver and a cost-sensitive classifier). Under the assumptions that the value function and optimal policy are within provided function approximation classes, and transitions are deterministic, it shows that an optimal policy is PAC-learnable using a number of oracle calls and number of episodes that is polynomial in the horizon, action space size, and number of states per level. The number of episodes also has a log dependence on the size of the value function approximation and policy approximation classes (so in the worst case this is polynomial in the number of states per level and the size of the action space). It improves over previous work by considering computational complexity rather than just statistical complexity. 1. Role of the size of the function classes: Could the authors please comment on the role of the size of the function classes G and Pi in Theorem 2? My understanding is that we assume access to an oracle LP solver that can optimize over G in a single oracle call. Solving an LP is polynomial in the input size, but if the class G is exponentially large can I still solve the LP in polynomial time? This is relevant of course because (1) the paper assumes that the actual value function lies in G; (2) typically one does not know the form of the value function, and so one would like to include all functions; (3) the set of all functions from X onto [0,1] is equivalent to the set of real vectors with dimension |X|, which is exponential in |X|; Adding to the pain, if I use a history dependent state (because I have a POMDP and want to use the method in this paper), then |X| is exponential in H. I agree that the example with a linear value function with a feature vector of not-too-large-dimension is appealing, but in this case we are unlikely to meet the assumption that the true value function is in the class of linear functions. 2. POMDP vs. MDP: Although this paper positions itself as studying POMDPs in the introduction, it assumes on lines 99-100 that the observation process (x_t) is Markov. Moreover, the reward depends only on x_t and the action, not the state S_t. They then correctly state (again, lines 99-100) that this makes the problem an MDP over X. It argues "The hidden states serve to introduce structure to the MDP and enable tractable learning." I don't understand why this is the case. Suppose I simply set S=X, s_t=x_t, and defined Gamma(s,a) = Gamma(x,a). Then my rewards, the observable stochastic process, the value function over X, and the set of optimal policies would all be unchanged. Moreover, the paper would be easier to understand, no less general, and I believe would better represent what the method described in the paper is capable of. Can the authors please explain more precisely why the hidden states enable tractable learning? 3. Somehow full POMDPs are very different from the MDP considered in this paper. It was very useful for me to consider the following example: Suppose we have binary actions and that x_h = a_1:h records the history of these actions. Also suppose that R(x,a) = 0 when x has length less than H. When x has length H, it is equal to 1 if x is equal to a particular binary sequence b, and 0 otherwise. b is fixed but unknown. The set G that I use is the set of value functions that I can have in this problem, enumerating over binary sequences b of length H. It has size 2^H. In particular, the g in G are of the form g(x) = 1(length(x)=H and x=b') for some b' in {0,1}^H. My intuition says that the best algorithm for this problem (both from a computational efficiency and oracle efficiency perspective) is to sample paths at random until I find the one that gives me a final reward of 1. The number of paths I need to sample will have expectation 2^H / 2. When I look at Theorem 2 and 3, at first glance they seem to be in contradiction. However, my computation is very simple, and the size of my state space at each level M is exponential in H. Thus, we actually have exponential dependence on H in both theorems. Moreover, if we did not have the reactivity assumption, I could have had a much smaller state space. I could have S = {0,1} \times [H], where the second component tracks time, and the first component tracks whether my actions have specified all of the bits in b correctly thus far. Then my M would be polynomial in H. Could the authors please discuss this in the paper? I believe it's an important aspect of the significance of the work. 4. Assumptions: The authors point this out, but it bears repeating that the assumptions are extremely strong. As described above, the paper is really about MDPs in my view. Moreover, the assumptions that the value function and policy are in some parametric class we are optimizing over is very strong. This combined with the lack of numerical experiments make this a paper that has a very good theoretical contribution, but not yet relevant to practice. Quality: high, though I would like more clarification along the lines of the comments above Originality: the paper is original, though it borrows many ideas from [1] Clarity: I found the paper hard to read. In part, this is because the material is dense, but partly it is because the authors did not elucidate along the lines of comments #1-3 above. Signifance: I think this paper is a significant theoretical result. #### UPDATE AFTER AUTHOR RESPONSE ##### Thanks for the responses to my major comments. I continue to feel that this paper deserves to be accepted.

Reviewer 3



The paper presents novel reinforcement learning algorithms for environments with rich observations. The paper lacks experiments but has some theoretical justifications. Impact of the submission is thus unclear to me.